# NEON: NEGATIVE EXTRAPOLATION FROM SELF-TRAINING IMPROVES IMAGE GENERATION

**Sina Alemohammad**[†]**, Zhangyang Wang**[†]**, Richard G. Baraniuk**[*]
ECE Department, The University of Texas at Austin; [*]ECE Department, Rice University;

## ABSTRACT

Scaling generative AI models is bottlenecked by the scarcity of high-quality training data. The ease of synthesizing from a generative model suggests using (unverified) synthetic data to augment a limited corpus of real data for the purpose of fine-tuning in the hope of improving performance. Unfortunately, however, the resulting positive feedback loop leads to model autophagy disorder (MAD, aka model collapse) that results in a rapid degradation in sample quality and/or diversity. In this paper, we introduce Neon (for Negative Extrapolation frOm self-traiNing), a new learning method that turns the degradation from self-training into a powerful signal for self-improvement. Given a base model, Neon first fine-tunes it on its own self-synthesized data but then, counterintuitively, reverses its gradient updates to extrapolate away from the degraded weights. We prove that Neon works because typical inference samplers that favor high-probability regions create a predictable anti-alignment between the synthetic and real data population gradients, which negative extrapolation corrects to better align the model with the true data distribution. Neon is remarkably easy to implement via a simple post-hoc merge that requires no new real data, works effectively with as few as 1k synthetic samples, and typically uses less than 1% additional training compute. We demonstrate Neon's universality across a range of architectures (diffusion, flow matching, autoregressive, and inductive moment matching models) and datasets (ImageNet, CIFAR-10, and FFHQ). In particular, on ImageNet 256x256, Neon elevates the xAR-L model to a new state-of-the-art FID of 1.02 with only 0.36% additional training compute. Code is available at https://github.com/VITA-Group/Neon

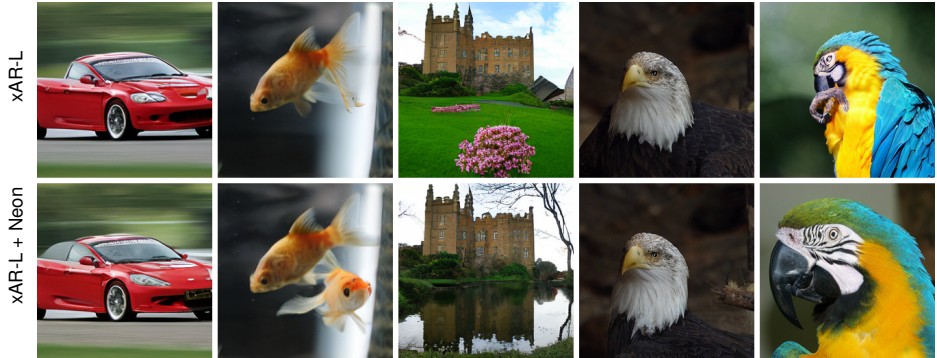

Figure 1: **Good to great: Neon's state-of-the-art performance on ImageNet-256.** Neon elevates a powerful baseline generative model (xAR-L, top row) to a new level of sharpness and realism (bottom row) with a simple post-hoc merge. This leap in quality, improving the Fréchet inception distance (FID) from 1.28 to a record-breaking 1.02, is accomplished with only 0.36% extra training compute.

Corresponding author: sina.alemohammad@austin.utexas.edu

## 1 INTRODUCTION

Modern generative models for images have achieved remarkable photorealism through continuous advances in architectures, training methods, and scale. Diffusion models (Ho et al., 2020; Song et al., 2021), flow matching approaches (Lipman et al., 2023; Liu et al., 2023), autoregressive architectures (Ding et al., 2021; Yu et al., 2022b), and few-step generators (Song et al., 2023; Zhou et al., 2025a) now form the backbone of large-scale image generation systems. Despite these advances, the most reliable path to state-of-the-art performance remains scaling: ever more parameters, ever larger datasets, and ever increasing compute (Henighan et al., 2020; Kaplan et al., 2020).

Important energy sustainability issues aside, this scaling paradigm faces a ***fundamental bottleneck***: ***high-quality training data***. Curating diverse, rights-cleared image datasets is expensive and time-consuming, with diminishing returns as existing sources are exhausted (Muennighoff et al., 2023; Villalobos et al., 2022). As the gap between model capacity and available training data widens, the field must explore alternative paths to model improvement that do not rely on ever-larger real datasets.

The ease of synthesizing data from generative models has inspired a range of model improvement approaches to augment a limited real data set. At the simplistic end, one can fine-tune a model on its own generated outputs. However, such naïve self-training has been shown to lead to "model autophagy disorder" (MAD) (Alemohammad et al., 2024a) or model collapse (Shumailov et al., 2024), where diversity and/or quality degrades. At the complicated end, researchers have avoided collapse through sophisticated workarounds like external verifiers for synthetic data quality (Feng et al., 2024), auxiliary discriminator networks (Kim et al., 2023), negative guidance during inference (Alemohammad et al., 2024b), and likelihood-based discrimination between distributions (Zheng et al., 2025). While effective, these approaches add significant computational overhead, are restricted to specific architectures, or require complex iterative training.

**Neon.** In this paper, ***we show that there is hidden promise in directly fine-tuning a model on its own generated data***. Our key insight is that the degradation due to self-training is not random noise but rather a power signal that is anti-aligned with the real-data population gradient. *Neon* (Negative Extrapolation from self-traiNing) exploits this anti-alignment through a simple parameter merge. Given a base model with parameters $\theta_r$ trained on real data, we first apply the naïve self-training approach: we generate synthetic samples and briefly fine-tune to obtain the parameters $\theta_s$ that exhibit degraded performance. Then, rather than using $\theta_s$ directly, we perform ***negative extrapolation***:

$$\theta_{\text{Neon}} = \theta_r - w(\theta_s - \theta_r) = (1+w)\theta_r - w\theta_s, \qquad w > 0, \tag{1}$$

where $w$ controls the extrapolation strength. The vector $\theta_s - \theta_r$ corresponds to the synthetic gradient direction; because this direction is anti-aligned with the (infinite real data) population gradient, reversing it reduces the true data risk and redistributes probability mass to under-represented modes.

**Contributions.** [C1] We introduce ***Neon***, a deceptively simple post-processing method that improves generative models by reversing their degradation on self-generated data (Section 3). In contrast to existing methods for synthetic data augmentation, Neon requires no additional real training data, no access to the original training data, no auxiliary models, no likelihood computation, and no inference modifications. [C2] We prove rigorously that mode-seeking inference samplers create a predictable anti-alignment between the synthetic and population gradients that guarantees the effectiveness of negative extrapolation (Section 3.1). [C3] We demonstrate Neon's universality across diffusion, flow matching (Section 4.1), autoregressive (Section 4.2), and few-step (Section 4.3) models on CIFAR-10, FFHQ, and ImageNet with $< 1\%$ additional compute and as few as 1k synthetic samples. For example, on ImageNet-256, Neon elevates xAR-L from an FID of 1.28 to the state-of-the-art 1.02 using only 0.36% additional compute. [C4] We show that Neon's improvement mechanism operates through a precision-recall trade-off that redistributes probability mass from over- to under-represented modes (Section 4.1). [C5] We demonstrate that the Neon degradation signal is transferable, which enables synthetic data from one model architecture to improve another (Section 4.4).

## 2 BACKGROUND

**Notation and definitions.** Let $\mathcal{D}$ be a training data set drawn from $p_{\text{data}}$. A training algorithm produces the generative model $G_\theta$, whose output is a score, velocity, or logit depending on the model family. The training budget $\mathcal{B}$ is the cumulative number of images seen (in millions):

$\mathcal{B}$ = (global steps) $\times$ (global batch size). An inference routine $\mathcal{I}$ with hyperparameters $\kappa$ induces a sampling distribution $q_{\theta,\kappa}$. Denote the idealized distribution without inference-time modifications (e.g., guidance) by $p_\theta := q_{\theta,\varnothing}$. We use $\mathrm{dist}(\cdot,\cdot)$ for a generic divergence, $|\cdot|$ for set cardinality, and the shorthand

$$\|x\|_M := \|M^{1/2}x\|_2, \quad \langle x,y\rangle_M := x^\top M y, \quad \|A\|_{\mathrm{op},M} := \|M^{1/2}AM^{-1/2}\|_{\mathrm{op}},$$

for any positive-definite matrix $M$, where $\|\cdot\|_2$, $\langle\cdot,\cdot\rangle$, and $\|\cdot\|_{\mathrm{op}}$ are the standard Euclidean norm, inner product, and operator norm. Let k denote $10^3$.

**Visual generative models.** Many image generators trace a path from noise to data via an affine interpolation $x_t = \alpha(t)x_0 + \sigma(t)\epsilon$ for $t \in [0,1]$, with $x_0 \sim p_{\mathrm{data}}$, $\epsilon \sim \mathcal{N}(0,I)$, and boundary conditions $\alpha(0) = 1$, $\sigma(0) = 0$, $\alpha(1) = 0$, $\sigma(1) = 1$, inducing $p_0 = p_{\mathrm{data}}$ and $p_1 = \mathcal{N}(0,I)$ (Lipman et al., 2023; Song et al., 2021).

**Diffusion models** (Ho et al., 2020; Song et al., 2021) train $G_\theta(x,t)$ to approximate the score $\nabla_x \log p_t(x)$ (or equivalently, predict noise). At inference, the learned score drives the reverse-time SDE or probability-flow ODE.

**Flow matching** (Lipman et al., 2023; Tong et al., 2024) learns the conditional velocity $v^\star(x_0,\epsilon,t) = \alpha'(t)x_0 + \sigma'(t)\epsilon$ by regressing $G_\theta(x_t,t)$ with squared error; sampling integrates $\dot{x}_t = G_\theta(x_t,t)$ from $t = 1$ to $t = 0$.

**Few-step generators** reduce sampling cost by collapsing many steps. Consistency models (Song et al., 2023) predict $x_0$ directly from $(x_t,t)$; IMM (Zhou et al., 2025a) learns direct transitions $x_s = G_\theta(x_t, t \to s)$ with moment-matching, enabling quality with $T \approx 1$–$8$ steps.

**Autoregressive models** (Ren et al., 2025; Tian et al., 2024) factorize images into tokens $y_{1:N} = \mathcal{T}(x)$ and model $p(y_{1:N}) = \prod_{i=1}^N p(y_{\pi(i)} \mid y_{\pi(<i)})$, where $G_\theta(y_{<i})$ outputs next-token logits trained via cross-entropy. The ordering $\pi$ and decoding choices (temperature, top-$k$) form part of inference hyperparameters $\kappa$.

**Self-training and collapse.** When models iteratively train on their own synthetic outputs, they exhibit what has been termed MADness or model collapse: $\mathbb{E}[\mathrm{dist}(p_{\mathrm{data}}, p_{\theta_t})]$ grows over time (Alemohammad et al., 2024a; Dohmatob et al., 2024; Shumailov et al., 2024). Pure self-training diverges, while mixing real and synthetic data converges to degraded equilibria (Bertrand et al., 2023; Gerstgrasser et al., 2024). While external signals beyond the training data can prevent collapse (Alemohammad et al., 2024b; Feng et al., 2024), these methods require additional resources such as verifiers or fresh data.

**Related work on synthetic data training.** Several recent methods successfully leverage synthetic data for model improvement, but require significant architectural constraints or computational overhead. Discriminator Guidance (Kim et al., 2023) trains a post-hoc discriminator on real versus generated samples across diffusion timesteps, using its gradients to correct the score function during sampling. While effective, it adds inference overhead and remains diffusion-specific. SIMS (Alemohammad et al., 2024b) employs self-generated data as negative guidance to steer diffusion trajectories away from degraded manifolds, but similarly requires inference-time modifications and is limited to diffusion models. Direct Discriminative Optimization (DDO) (Zheng et al., 2025) reformulates likelihood-based models as implicit discriminators via log-likelihood ratios between target and reference models, enabling strong improvements for diffusion (via ELBO) and autoregressive models, but fundamentally cannot apply to likelihood-free architectures like flow matching (Lipman et al., 2023) or inductive moment matching (Zhou et al., 2025a). Self-Play Fine-Tuning (Yuan et al., 2024) iteratively pits models against earlier checkpoints, surpassing RLHF methods on human preference benchmarks but requiring multiple training rounds and substantial computational overhead. In contrast to these methods, Neon requires no auxiliary models, no inference modifications, no likelihood computations, and works across all architectures with a simple post-hoc parameter merge.

# 3 NEON: NEGATIVE EXTRAPOLATION FROM SELF-TRAINING

When models train on synthetic samples produced by their inference procedure $\mathcal{I}$ (what we call "self-training"), they predictably degrade. Neon exploits this: by reversing the degradation direction, we can improve a model without additional real data. Starting from a base generator $G_{\theta_r}$ (typically trained on real data), we: (i) generate the synthetic dataset $\mathcal{S}$ once using test-time inference $\mathcal{I}(G_{\theta_r};\kappa)$,

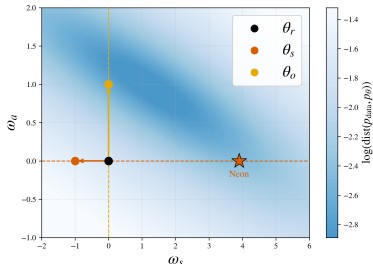 Figure 2: **Neon's key idea: synthetic degradation and real-data improvement point in opposite directions.** This toy 2D Gaussian example plots as a heat map the log Wasserstein distance to the true data distribution $p_{\text{data}}$ from the generative model $G_{\theta(w_s, w_o)}$. We see that updating the model's parameters in the reverse of the direction they would be updated by fine-tuning on self-synthesized data (increasing $w_s$) achieves similar improvements to fine-tuning the base model with $4\times$ more real data (increasing $w_o$).

(ii) briefly (e.g., using $< 1\%$ of the original training budget) fine-tune the generator on $\mathcal{S}$ to obtain the degraded $G_{\theta_s}$, and (iii) negatively extrapolate via the parameter merge:

$$\theta_{\text{Neon}} := \theta_r - w(\theta_s - \theta_r) = (1+w)\theta_r - w\theta_s, \tag{2}$$

where $w > 0$ controls the extrapolation strength. Algorithm 1 provides the full details.

---

**Algorithm 1** Neon: Negative Extrapolation from Self-Training

---

**Require:** Base model $G_{\theta_r}$, inference routine $\mathcal{I}$ with hyperparameters $\kappa$
    **Hyperparameters:** Synthetic dataset size $n_s = |\mathcal{S}|$, extrapolation strength $w$, training budget $\mathcal{B}$
1: $\mathcal{S} \leftarrow \{x_i\}_{i=1}^{n_s}$ where $x_i \sim q_{\theta_r, \kappa}$ induced by $\mathcal{I}(G_{\theta_r}; \kappa)$      ▷ sample using test-time inference
2: $G_{\theta_s} \leftarrow \text{FineTune}(G_{\theta_r}, \mathcal{S}, \mathcal{B})$      ▷ briefly fine-tune on synthetic data
3: $\theta_{\text{Neon}} \leftarrow (1+w)\theta_r - w\theta_s$      ▷ reverse the degradation
**Output:** Final generator $G_{\theta_{\text{Neon}}}$

---

## 3.1 WHY NEON WORKS

**Geometric intuition via a toy study.** To visualize why negative extrapolation from degradation succeeds, consider a 2D Gaussian example where $p_{\text{data}} = \mathcal{N}(\mu_{\text{true}}, \Sigma_{\text{true}})$. We train a base model $G_{\theta_r}$ on 1k real samples and then define two directions in parameter space: the ***degradation direction*** from fine-tuning the base model on $10^5$ synthetic samples from $G_{\theta_r}$ to obtain $G_{\theta_s}$, and an oracle ***improvement direction*** from fine-tuning on 5k real samples (the original 1k real data points plus 4k new ones) to obtain $G_{\theta_o}$. We evaluate models in the 2D span of these directions:

$$\theta(w_s, w_o) \;=\; \theta_r \;+\; w_s \underbrace{(\theta_r - \theta_s)}_{-\text{ degradation direction (Neon)}} \;+\; w_o \underbrace{(\theta_o - \theta_r)}_{\text{oracle improvement direction}} \tag{3}$$

where $w_s$ controls the amount of negative extrapolation (Neon) and $w_o$ adds real-data improvement (oracle baseline). Figure 2 visualizes our key finding: moving backwards along the Neon direction alone ($w_o = 0$) yields substantial improvement, indicating that the opposite of degradation direction and additional real-data improvement direction both point towards a better approximation of the true data distribution.

**Theoretical analysis.** We now formalize the intuition provided by the toy example. We prove that typical inference samplers cause the synthetic and real data gradients to point in opposite directions, enabling negative extrapolation to reduce the true data risk.

**Set-up.** Let $\ell_\theta(x)$ be differentiable loss function and $\mathcal{R}_{\text{data}}(\theta) := \mathbb{E}_{p_{\text{data}}}[\ell_\theta(X)]$ the corresponding risk. Let $\theta^* \in \arg\min_\theta \mathcal{R}_{\text{data}}(\theta)$ and write $\theta_r = \theta^* + \varepsilon$ with $\|\varepsilon\|_{H_d}^2 = \varepsilon^\top H_d \varepsilon$. Let $q_{\theta_r, \kappa}$ denote the fixed sampler constructed once at $\theta_r$. Define

$$\phi_\theta(x) := \nabla_\theta \ell_\theta(x), \qquad H_d := \nabla^2 \mathcal{R}_{\text{data}}(\theta^*) = \mathbb{E}_{p_{\text{data}}}\big[\partial_\theta \phi_\theta(X)\big]_{\theta = \theta^*},$$

$$\mathcal{R}_{\text{syn}}(\theta) := \mathbb{E}_{x \sim q_{\theta_r, \kappa}}[\ell_\theta(x)], \quad r_d := \nabla_\theta \mathcal{R}_{\text{data}}(\theta)\big|_{\theta_r}, \qquad r_s := \nabla_\theta \mathcal{R}_{\text{syn}}(\theta)\big|_{\theta_r}.$$

Let $P \succ 0$ be a preconditioner and set $K := H_d^{1/2} P H_d^{1/2}$ with $mI \preceq K \preceq MI$.

We say the synthetic and real data gradients are ***anti-aligned*** at $\theta_r$ if their preconditioned inner product is negative

$$s := \langle r_d, P r_s \rangle < 0.$$

**Neon improves under anti-alignment.** Short synthetic fine-tuning yields $\theta_s = \theta_r - \alpha\, P\, r_s + O(\alpha^2)$, which Neon reverses: $\theta_{\text{Neon}} = \theta_r + w\alpha\, P r_s + O(w\alpha^2)$. A Taylor expansion of the risk yields

$$\mathcal{R}_{\text{data}}(\theta_{\text{Neon}}) = \mathcal{R}_{\text{data}}(\theta_r) + w\alpha\, s + \frac{(w\alpha)^2}{2}\, r_s^\top P^\top \nabla^2 \mathcal{R}_{\text{data}}(\theta_r)\, P\, r_s + O\big((w\alpha)^3\big). \quad (4)$$

When $s < 0$, the negative linear term dominates for small $w > 0$, ensuring that $\mathcal{R}_{\text{data}}(\theta_{\text{Neon}}) < \mathcal{R}_{\text{data}}(\theta_r)$. When $\mathcal{R}_{\text{data}}$ is locally convex at $\theta_r$ (i.e., $\nabla^2 \mathcal{R}_{\text{data}}(\theta_r) \succeq 0$), the optimal $w^* = -s/(\alpha z) > 0$, where $z := r_s^\top P^\top \nabla^2 \mathcal{R}_{\text{data}}(\theta_r) P r_s$.[1] See Appendix B.4 for the proof.

**Sampler-induced anti-alignment.** Let

$$b := \mathbb{E}_{q_{\theta_r,\kappa}}\big[\phi_{\theta^*}(X)\big], \quad \Delta := \mathbb{E}_{q_{\theta_r,\kappa}}\big[J_{\theta^*}(X)\big] - \mathbb{E}_{p_{\text{data}}}\big[J_{\theta^*}(X)\big], \quad J_{\theta^*}(x) := \partial_\theta \phi_\theta(x)\big|_{\theta^*}, \quad (5)$$

and measure their sizes in the $H_d$–geometry by

$$\eta_0 := \|b\|_{H_d^{-1}}, \qquad \eta_1 := \|\Delta\|_{\text{op}, H_d^{-1}}.$$

Define the angle between the model error $\varepsilon$ and the sampler bias $b$ in the $H_d$–geometry by

$$\cos\varphi := \frac{\langle \varepsilon,\, H_d^{-1} b\rangle_{H_d}}{\|\varepsilon\|_{H_d} \|H_d^{-1} b\|_{H_d}} \in [-1, 1]. \quad (6)$$

Intuitively, $\cos\varphi < 0$ means that the sampler's bias points is in a direction opposing the current error, favoring anti-alignment.

**Theorem 1** (Anti-alignment under inference mismatch). *Let $K := H_d^{1/2} P H_d^{1/2}$ with spectral bounds $mI \preceq K \preceq MI$. Then the alignment $s = \langle r_d, P r_s\rangle$ obeys*

$$s \leq M(1 + \eta_1)\|\varepsilon\|_{H_d}^2 - m\,\eta_0\,\|\varepsilon\|_{H_d}[-\cos\varphi]_+ + O(\|\varepsilon\|_{H_d}^3).$$

*Consequently, a* sufficient *condition for $s < 0$ is that the leading two terms on the right-hand side be negative. In particular, for $\cos\varphi < 0$ and sufficiently small $\|\varepsilon\|_{H_d}$,*

$$\boxed{\|\varepsilon\|_{H_d} < \frac{m\,\eta_0}{M(1 + \eta_1)}\,(-\cos\varphi)} \implies s < 0.$$

See Appendices B.4–B.5 for the proof.

**Mode-seeking samplers enable Neon.** Many deployed inference routines can be written as a monotone reweighting of the reference model,

$$q(x) \propto f\big(\log p_{\theta_r}(x)\big) p_{\theta_r}(x), \quad \text{with } f \text{ nondecreasing and not a.e. constant.}$$

Such *mode-seeking* samplers emphasize high-density regions. The following theorem shows that this structure guarantees anti-alignment.

**Theorem 2** (Mode-seeking samplers induce $\cos\varphi < 0$). *If the sampler $q$ is mode-seeking (monotone reweighting with $f$ nondecreasing), then $\cos\varphi < 0$ to first order in $\|\varepsilon\|_{H_d}$, guaranteeing anti-alignment near good models.*

See Appendices B.6–B.7 for the proof.

*Concrete instances.* Combining Theorems 1 and 2, we obtain that Neon reduces risk for the following standard configurations:

(i) **Autoregressive models**: Temperature $\tau < 1$, top-$k$, and top-$p$ sampling all produce nondecreasing reweighting of $\log p_{\theta_r}$ (App. B.6).

(ii) **Diffusion and flow models**: Finite-step ODE solvers (including classifier-free guidance) induce monotone terminal reweighting to first order in step size (App. B.7).[2]

---

[1] Local convexity is sufficient but not necessary. The result holds under the weaker condition of directional smoothness along the step direction $d = P r_s$. See Appendix B.4 for details.

[2] Assumes curvature–density coupling (A-MONO): the conditional expectation $\mathbb{E}[\sum_k \|\nabla_x f(X_{t_k}, t_k)\|_{\text{Fr}}^2 | X_0 = x_0]$ increases with $\log p_{\theta_r}(x_0)$. See App. B.7.

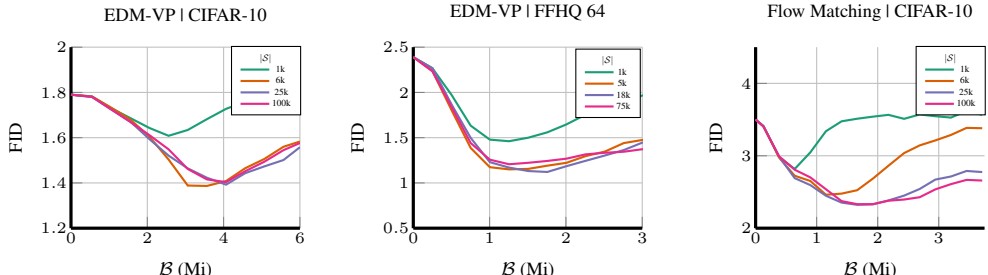

Figure 3: **Neon consistently improves FID with minimal self-training overhead.** Minimum FID (optimized over extrapolation strength $w$) vs. self-training budget $\mathcal{B}$ (millions of images seen during fine-tuning on $\mathcal{S}$) for varying synthetic dataset sizes $|\mathcal{S}|$, on EDM-VP (CIFAR-10/FFHQ-64) and flow matching (CIFAR-10). Optimal gains use $\mathcal{B} \leq 3\text{Mi}$ ($< 2\%$ of base model training compute for EDM; $< 3\%$ for flow), confirming Neon's efficiency. At $\mathcal{B} = 0$, FID reflects the base model (no Neon).

**When interpolation (not extrapolation) helps**. The theory also identifies a complementary regime: *diversity-seeking* samplers ($f$ nonincreasing) reverse the angle to $\cos\varphi \geq 0$, making $s > 0$ and thus favoring interpolation ($-1 < w < 0$) over Neon's negative extrapolation ($w > 0$). However, such samplers are rare in practice: autoregressive models would require high temperature $\tau > 1$ or anti-mode truncations; diffusion/flow models would need samplers that decrease contraction near modes, which is atypical. In these cases, the Neon formula $\theta_{\text{Neon}} = (1 + w)\theta_r - w\theta_s$ still applies, but the optimal merge weight satisfies $-1 < w < 0$ (moving toward $\theta_s$) rather than $w > 0$ (moving away from $\theta_s$). See App. B.9 for formal analysis.

**Finite $|\mathcal{S}|$ effects.** Our analysis assumes that the population synthetic gradients $r_s(\theta_r)$, but in practice we use finite $\mathcal{S}$ with brief fine-tuning from $\theta_r$. For checkpoint $\theta_s$ after $T$ steps with step size $\alpha$, the displacement $d_T := (\theta_s - \theta_r)/(\alpha T)$ concentrates on $-Pr_s^{(\mathcal{S})}(\theta_r)$ when $T$ is sufficiently large while $\alpha T$ remains small, yielding stable, low-variance Neon directions despite limited $|\mathcal{S}|$. This produces a U-shaped performance in $|\mathcal{S}|$: very small sets are variance-limited, very large sets amplify curvature effects (inflating the quadratic term in our Taylor expansion), while moderate sizes optimally balance these competing factors. See Appendix B.10 for formal bounds and parameter selection guidance.

## 4 EXPERIMENTS

We evaluate Neon across four model families — diffusion (EDM (Karras et al., 2022)), flow matching (Tong et al., 2023; 2024), autoregressive (VAR (Tian et al., 2024), xAR (Ren et al., 2025)), and few-step (IMM (Zhou et al., 2025a)) — on ImageNet (Deng et al., 2009), CIFAR-10 (Krizhevsky & Hinton, 2009), and FFHQ (Karras et al., 2019).

For each model, starting from a public checkpoint $G_{\theta_r}$, we generate synthetic datasets $\mathcal{S}$ using the FID-optimal inference settings $\kappa$ from each paper. We fine-tune on $\mathcal{S}$ with the original training recipe at reduced learning rate (see Appendix C for details). We report FID as our primary metric using 10k/50k samples for hyperparameter search/final evaluation (Heusel et al., 2017), with Precision/Recall (Kynkäänniemi et al., 2019) at $k = 5$ nearest neighbors. For a comprehensive comparison of Neon against state-of-the-art generative models across all benchmarks, please see Table A.1.

### 4.1 DIFFUSION AND FLOW MATCHING MODELS

We evaluate Neon with the EDM-VP (Karras et al., 2022) (CIFAR-10 conditional, FFHQ-64 unconditional) and flow matching (Tong et al., 2023; 2024) (CIFAR-10 unconditional) models using public checkpoints. The synthetic datasets $\mathcal{S}$ were generated with default inference settings.

**Results.** Figure 3 plots the FID vs. the fine-tuning budget $\mathcal{B}$ for various $|\mathcal{S}|$. Neon achieves substantial gains with minimal overhead: Neon+EDM-VP trained on CIFAR-10 improves the FID from 1.78 to **1.38** using only 6k synthetic samples and $1.75\%$ extra compute compared to training the base model. Neon+EDM-VP trained on FFHQ-64 improves the FID from 2.39 to **1.12** using only 18k samples and $0.85\%$ additional compute. Neon+Flow matching on CIFAR-10 improves the FID from 3.5 to **2.32** using only 25k samples and $3.2\%$ additional compute. Neon's performance shows a non-monotonic relationship with the synthetic dataset size $|\mathcal{S}|$, with optimal performance in the range

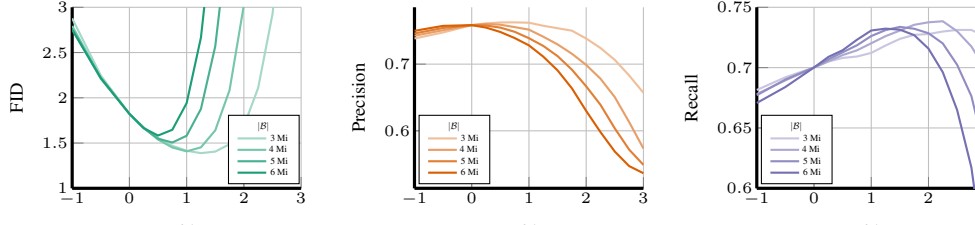

Figure 4: **Neon trades precision for recall, yielding net FID improvement.** For the EDM-VP model trained on CIFAR-10, we plot the FID, precision, and recall vs. negative extrapolation strength $w$ for various training budgets $\mathcal{B}$. $w = -1$ corresponds to the model directly trained on synthetic data, i.e., $\theta_{\text{Neon}} = \theta_s$. $w = 0$ corresponds to the base model, i.e., $\theta_{\text{Neon}} = \theta_r$. $w > 0$ corresponds to the negative extrapolation regime where Neon demonstrates its improvement capability. In each case, $|\mathcal{S}| = 6k$

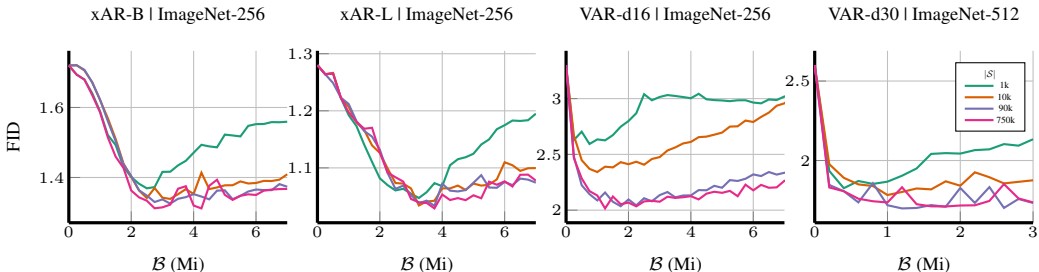

Figure 5: **Neon consistently improves autoregressive models across architectures and resolutions.** We plot the minimum FID (optimized over merge weight $w$ and CFG scale $\gamma$) versus the fine-tuning budget $\mathcal{B}$ for various synthetic dataset sizes $|\mathcal{S}|$. From left: xAR-B and xAR-L on ImageNet-256 (with xAR-L achieving a state-of-the-art 1.02 FID), VAR-d16 on ImageNet-256, and VAR-d30 on ImageNet-512.

6k–25k samples. Smaller $|\mathcal{S}|$ require more precise $w$ tuning but converge rapidly; larger $|\mathcal{S}|$ support a wider range of $w$'s but slower convergence.

Figure 4 dissects Neon's effect on EDM-VP trained on CIFAR-10 using precision-recall metrics with $|\mathcal{S}| = 6k$. The FID vs. weight relationship (left panel) exhibits the unimodal shape predicted by our Taylor series analysis. As fine-tuning progresses, the optimal $w^*$ decreases, which is consistent with $w^* \approx -s/(\alpha z)$, where $\alpha$ increases with training steps. The precision-recall trade-off (middle/right panels) reveals Neon's mechanism: precision monotonically decreases with $w$, while recall follows an inverted-U peaking near the FID-optimal weight. This aligns with our analysis: fine-tuning on synthetic data concentrates probability mass on well-captured modes, degrading coverage. By reversing this direction, Neon redistributes mass from over-represented to under-represented regions, trading precision for improved recall and yielding net FID improvement. These dynamics intensify with longer fine-tuning, with later checkpoints showing sharper recall peaks and steeper precision drops. (See Appendix D for all models.)

## 4.2 AUTOREGRESSIVE MODELS

We evaluate Neon's impact on xAR-B and xAR-L (Ren et al., 2025) (ImageNet-256), VAR-d16 (Tian et al., 2024) (ImageNet-256), and VAR-d30 (ImageNet-512). Both model families use CFG, with VAR adding top-$k$/top-$p$ sampling; these are mode-seeking samplers, and so our theory predicts Neon benefits. At evaluation, we jointly optimize both the merge weight $w$ and CFG scale $\gamma$. Co-optimization is crucial to reaching the best FID: $w$ increases recall at precision's expense, while $\gamma$ does the opposite.

**Results.** Figure 5 depicts the best FID after $(\gamma, w)$ grid search versus fine-tuning budget $\mathcal{B}$, testing up to $|\mathcal{S}| = 750k$ synthetic samples. The xAR family FID improves monotonically: xAR-B from 1.72 to **1.31** (750k synthetic samples, $0.41\%$ additional compute); xAR-L from 1.28 to the state-of-the-art FID **1.02** (750k samples, $0.36\%$ additional compute), surpassing UCGM's 1.06 (Sun et al., 2025). Even with just 1k samples, the xAR models achieve near-optimal performance (xAR-L: 1.05, xAR-B: 1.36), indicating that the degradation direction stabilizes quickly and requires minimal synthetic data to identify. VAR-d16 improves from 3.30 to **2.01** (750k samples, $0.64\%$ additional compute) but

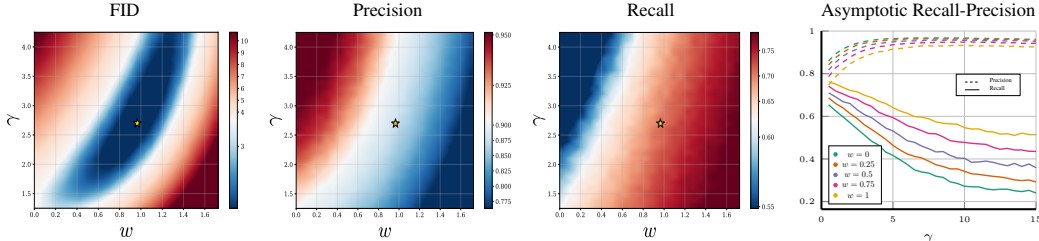

Figure 6: **Optimal precision-recall trade-offs for VAR-d16 as a function of $w$ and $\gamma$.** Left: Heatmaps for FID, precision, and recall on ImageNet-256 ($|\mathcal{S}|$=750k, $\mathcal{B}$=1.25Mi) from a grid search over $w$ and $\gamma$. The star marks the best FID ($w^*\approx 1.0$, $\gamma^*\approx 2.7$) achieving FID 2.01, unreachable by either parameter alone. Right: Asymptotic precision-recall curves showing expanded behavioral range through joint tuning.

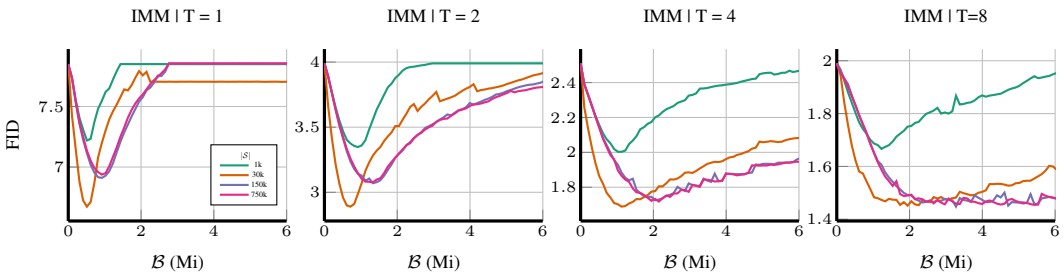

Figure 7: **Neon dramatically improves few-step inference for IMM on ImageNet-256.** Minimum FID (optimized over $w$ and $\gamma$) vs. fine-tuning budget $\mathcal{B}$ for different $|\mathcal{S}|$. Synthetic data were generated using $T$=8, $\gamma$=1.5. From left: $T$=1, 2, 4, 8 inference steps. Neon achieves substantial FID reductions with near-zero additional compute ($< 0.005\%$ of IMM's training), with Neon improved model with 4-step nearly matching base model with 8-step generation quality.

requires larger synthetic datasets—performance degrades with $|\mathcal{S}| < 90$k. VAR-d30 achieves its best FID of **1.69** with just 90k samples; adding more synthetic data provides no further meaningful improvement, suggesting the model has reached its capacity for Neon-based enhancement at this checkpoint.

Figure 6 visualizes the $(w, \gamma)$ interaction for VAR-d16. The FID landscape's diagonal valley with optimum ($w^*\approx 1.0$, $\gamma^*\approx 2.7$) yields FID **2.01**. Independent optimization ($\gamma$=1.25) yields FID 3.01 — far worse. Joint tuning enables precision-recall trade-offs unreachable by either parameter alone: at the optimum, precision drops to $\sim$0.87 while recall rises to $\sim$0.63. The rightmost panel reveals the asymptotic behavior: as $\gamma$ increases, the models converge to high precision ($> 0.95$) but severely degraded recall ($< 0.45$), leading to mode collapse. Higher $w$ values provide partial protection — at $w = 2$, the low-recall limit rises to $\sim$0.55 vs. $\sim$0.40 at $w = 0$, demonstrating how negative extrapolation counteracts CFG's mode-seeking tendency even at extreme guidance scales.

### 4.3 FEW-STEP GENERATORS

We investigate Neon paired with Inductive Moment Matching (IMM) (Zhou et al., 2025a) on ImageNet-256. We generated $\mathcal{S}$ using $T$=8 steps with CFG scale $\gamma$=1.5. At evaluation, we tested the models across inference steps $T\in\{1, 2, 4, 8\}$ and jointly searched over $(w, \gamma)$.

**Results.** Figure 7 plots the FID vs. the fine-tuning budget $\mathcal{B}$. Neon delivers dramatic improvements across all step counts with minimal overhead relative to IMM's 40,960Mi training budget. Performance scales inversely with the number of inference steps. Neon improves $T$=1 (single-step) inference to an FID of **6.67**. $T$=2 reaches **2.89**; $T$=4 reaches **1.69**; and $T$=8 reaches **1.46**. Remarkably, 4-step inference nearly matches base model with 8-step quality (1.69 vs. 1.98), effectively halving the inference cost. Unlike IMM's tens of thousands of million-image steps, Neon achieves optimal performance within 2Mi in all experiments for different $|\mathcal{S}|$, demonstrating rapid degradation direction stabilization for few-step models. The 30k sample sweet spot across all $T$ suggests that

few-step generators are particularly well-suited for Neon, as their training already distills multi-step dynamics into compact transitions, making the synthetic degradation signal especially informative.

## 4.4 ABLATION STUDIES

**Neon is transferable across different architectures.** A key advantage of Neon is that the degradation signal is transferable across different model architectures. We confirm this empirically in Figure 8, by improving a baseline unconditional EDM-VP model (FID = 1.97) using synthetic data from different sources. While data from the model itself yields the strongest improvement (FID = 1.38), cross-architecture transfer is highly effective. Data from a flow matching model achieves an FID of 1.59, and from an IMM model reaches 1.80. The theory expounded in Appendix B.8 formalizes why Neon is transferable. Consider models A and B that minimize the same objective with Hessians $H_d^{(A)}$ and $H_d^{(B)}$. If these Hessians are spectrally close (equivalent norms up to constants $c, C$) and the architectures induce similar sampler biases (small mismatch

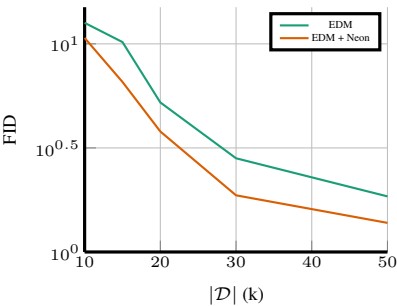

Figure 8: **Neon supports cross-architecture synthetic data transfer.** We illustrate by using synthetic data from an IMM and a Flow model to improve EDM-VP on CIFAR-10.

$\zeta$ in the terms $b, \Delta$ defined in (5), then anti-alignment transfers from one model to the other. That is, when model (A) satisfies $s^{(A)} \leq -\mu < 0$, any nearby model (B) inherits $s^{(B)} \leq -\mu/2 < 0$. Intuitively, models learning similar representations exhibit similar overconfidence patterns, and so one model's degradation direction corrects another's biases. This makes Neon practical when generating samples from the target model is costly.

To test if any out-of-distribution dataset provides a useful signal, we replaced the synthetic data with CIFAR-10C (Hendrycks & Dietterich, 2019), a dataset of corrupted real images. Neon resulted in no FID improvement. This null result confirms that Neon specifically leverages the anti-alignment from a model overemphasizing its own modes — a bias absent in structured corruptions like CIFAR-10C.

**How good must the base model be?** A key question is whether Neon's benefits are limited to nearly optimal models, since our theory guarantees anti-alignment only when the model error $\|\varepsilon\|_F$ is small. To test this condition's robustness, we applied Neon to a spectrum of EDM-VP base models trained on CIFAR-10 subsets of varying sizes. Figure 9 shows that Neon offers substantial improvements

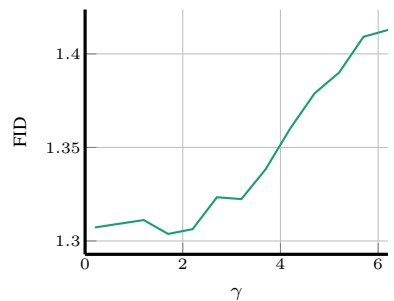

Figure 9: **Neon does not require a near-optimal base model to succeed.**

across the entire quality spectrum. Strikingly, a model trained on only 30k real samples (FID 1.87) and improved with Neon nearly matches the baseline model trained on the full 50k dataset (FID 1.85). This demonstrates that ***Neon can compensate for a 40% reduction in real training data***, confirming the anti-alignment condition ($s < 0$) is not fragile but holds across a wide range of model qualities. This bodes well for data-scarce applications. In all of the experiments, we used $|\mathcal{S}| = 6k$.

**Sensitivity to synthetic data quality.** Our main experiments generated synthetic datasets using optimal inference settings for FID (e.g., $\gamma = 2.7$ for xAR-B). To test the sensitivity to the quality of $\mathcal{S}$, we trained Neon+xAR-B on ImageNet-256 with $|\mathcal{S}| = 90k$ and varied the CFG scale used during generation. We generated synthetic datasets with $\gamma \in [0, 6.2]$, fine-tuned on each $\mathcal{S}$, and then optimized the final Neon model. Figure 10 demonstrates Neon's remarkable robustness: despite training on synthetic data of varying quality, the final FID remains near-optimal (1.30–1.31) for any $\gamma \in [1, 3]$. Even suboptimal synthetic datasets yield performance within 3% of optimal. This suggests that Neon captures the fundamental mode-seeking

Figure 10: **Neon does not require high-quality synthetic data to succeed.**

bias rather than requiring precisely tuned synthetic data. Only at extreme values (e.g., $\gamma \geq 6$) does performance degrade significantly, likely due to excessive mode collapse in $\mathcal{S}$.

## 5   CONCLUSIONS

We have introduced Neon, a simple and efficient post-processing method that improves generative models by inverting the degradation caused by self-training. Neon is grounded in a key insight: common mode-seeking inference samplers induce a predictable anti-alignment between gradients from synthetic and population data, explaining both the failure of naïve self-training and Neon's success. By extrapolating away from this degradation direction, Neon corrects the sampler's inherent bias, redistributing probability mass from over-represented modes to under-represented ones, thereby enhancing recall and overall generation fidelity. Neon's effectiveness across diverse model architectures and training datasets suggests that we can reframe model degradation not as a failure, but as a structured, harnessable signal for improvement in an increasingly data-scarce field. Our work also positions inference samplers as valuable diagnostic tools for uncovering and remedying a model's distributional flaws.

Neon opens several promising avenues for future research. Can we identify diversity-promoting samplers that foster positive alignment between synthetic and real data, thereby enabling direct self-improvement without inversion? Can we actively synthesize optimal "bad" datasets that elicit a degradation direction that maximizes the corrective signal?

As the demand for more capable models outpaces the availability of high-quality data, methods that extract more value from the models themselves will be crucial. Neon's success establishes a practical principle for self-correction: by identifying and inverting the predictable biases introduced by a model's own processes, we can achieve substantial performance gains. This work shows that even seemingly harmful signals can be harnessed for improvement, a finding that shows sometimes to move forward, we have to go backward.

## ACKNOWLEDGMENTS

This work was supported in part by NSF Awards 2145346 (CAREER), 02133861 (DMS), 2113904 (CCSS), and the NSF AI Institute for Foundations of Machine Learning (IFML); ONR N00014-23-1-2714; ONR MURI N00014-20-1-2787; DOE DE-SC0020345; and DOI 140D0423C0076. Thanks to Predrag Neskovic for pushing us down the path towards understanding negative extrapolation and to Ahmed Imtiaz Humayun for early discussions and for suggesting Algorithm 1 for model self-improvement with synthetic data.

## ETHICS STATEMENT

This research is a methodological contribution evaluated on standard public datasets (ImageNet, CIFAR-10, FFHQ) with no private or identifying data. We do not anticipate direct harms from our experiments; however, as with any image-synthesis method, potential misuse exists.

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

# A  STATE OF THE ART COMPARISON

Table A.1: Comprehensive comparison of generative models across four standard benchmarks. Best results are highlighted in  blue .

(a) Results on CIFAR-10.

| Type | Model | NFE | Uncond | Cond |
|------|-------|-----|--------|------|
| GAN | StyleGAN2-ADA (Karras et al., 2020) | 1 | 2.92 | 2.42 |
| | StyleGAN-XL (Sauer et al., 2022) | 1 | – | 1.85 |
| | SAN (Takida et al., 2024) | 1 | 1.85 | 1.36 |
| | CAF (Park et al., 2024) | 1 | 1.48 | 1.39 |
| Diff. & Flow | DDPM (Ho et al., 2020) | 1000 | 3.17 | – |
| | iDDPM (Nichol & Dhariwal, 2021) | 4000 | 2.90 | – |
| | NCSN++ (Song & Ermon, 2020) | 2000 | 2.20 | – |
| | DPM-Solver (Lu et al., 2022) | 10 | 4.70 | – |
| | LSGM (Vahdat et al., 2021) | 138 | 2.10 | – |
| | EDM-VP (Karras et al., 2024b) | 35 | 1.97 | 1.79 |
| | GMem-XL (Tang et al., 2024) | 35 | – | 1.22 |
| | Flow Matching (Lipman et al., 2023) | 100 | 3.50 | – |
| | Rectified Flow (Liu et al., 2023) | 127 | 2.58 | – |
| Few-step | CTM (Kim et al., 2024) | 2 | 1.87 | – |
| | sCT (Song et al., 2023) | 2 | 2.06 | – |
| | IMM (Zhou et al., 2025a) | 1 | 3.20 | – |
| Post-hoc | EDM + DG (Kynkäänniemi et al., 2024) | 53 | 1.77 | 1.64 |
| | EDM + DDO (Zheng et al., 2025) | 35 | 1.38 | 1.30 |
| | EDM + SIMS (Alemohammad et al., 2024b) | 70 | 1.33 | – |
| | EDM + SiD$^2$A (Zhou et al., 2025b) | 1 | 1.49 | 1.39 |
| Ours | EDM + **Neon** | 35 | 1.38 | 1.38 |
| | Flow + **Neon** | 100 | 2.32 | – |

(b) Results on FFHQ-64×64.

| Type | Model | NFE | FID |
|------|-------|-----|-----|
| GAN | R3GAN (Huang et al., 2024) | 1 | 1.95 |
| | Anycost GAN (Lin et al., 2021) | 1 | 2.52 |
| | MSG-GAN (Karnewar & Wang, 2020) | 1 | 2.70 |
| | StyleGAN2 (Karras et al., 2019) | 1 | 3.32 |
| Diffusion | EDM-G++ (Karras et al., 2024b) | 71 | 1.98 |
| | EDM-VE (Karras et al., 2024b) | 79 | 2.53 |
| | EDM-VP (Karras et al., 2024b) | 79 | 2.39 |
| Post-hoc. | SiD$^2$A (Zhou et al., 2025b) | 1 | 1.04 |
| | EDM + SIMS (Alemohammad et al., 2024b) | 158 | 1.04 |
| | EDM + D2O (Zheng & Yang, 2025) | 1 | 1.08 |
| | EDM + D2O-F (Zheng & Yang, 2025) | 1 | 0.85 |
| Ours | EDM + **Neon** | 79 | 1.12 |

(c) Results on ImageNet-256×256.

| Type | Model | NFE | FID |
|------|-------|-----|-----|
| GAN | GigaGAN (Kang et al., 2023) | 1 | 3.45 |
| | StyleGAN-XL (Sauer et al., 2022) | 1 | 2.30 |
| Diffusion | ADM (Dhariwal & Nichol, 2021) | 250 | 10.94 |
| | LDM-4 (Rombach et al., 2022) | 250 | 10.56 |
| | DiT-XL/2 (Peebles & Xie, 2023) | 250 | 9.62 |
| | U-ViT (Bao et al., 2023) | 50 | 2.29 |
| | MDT (Gao et al., 2023) | 250 | 6.23 |
| | REPA-UCGM (Sun et al., 2025) | 80 | 1.06 |
| Masked | MaskGIT (Chang et al., 2022) | 8 | 6.18 |
| | MAR (Li et al., 2024) | 100 | 1.98 |
| | MaskBit (Weber et al., 2024) | 256 | 1.52 |
| AR | VQGAN (Yu et al., 2022a) | 256 | 15.78 |
| | VAR-d16 (Tian et al., 2024) | 10 | 3.30 |
| | VAR-d30 (Tian et al., 2024) | 10 | 1.92 |
| | xAR-B (Ren et al., 2025) | 40 | 1.72 |
| | xAR-L (Ren et al., 2025) | 50 | 1.28 |
| Few-step | Shortcut (Frans et al., 2025) | 1 | 10.60 |
| | IMM (T=1) (Zhou et al., 2025a) | 1 | 7.77 |
| | IMM (T=8) (Zhou et al., 2025a) | 8 | 1.99 |
| Post-hoc | VAR-d16 + DDO (Zheng et al., 2025) | 10 | 2.54 |
| | VAR-d30 + DDO (Zheng et al., 2025) | 10 | 1.79 |
| Ours | VAR-d16 + **Neon** | 10 | 2.01 |
| | xAR-B + **Neon** | 40 | 1.31 |
| | xAR-L + **Neon** | 50 | 1.02 |
| | IMM (T=8) + **Neon** | 8 | 1.46 |
| | IMM (T=4) + **Neon** | 4 | 1.68 |
| | IMM (T=2) + **Neon** | 2 | 2.88 |
| | IMM (T=1) + **Neon** | 1 | 6.67 |

(d) Results on ImageNet-512×512.

| Type | Model | NFE | FID |
|------|-------|-----|-----|
| GAN | BigGAN-deep (Brock et al., 2019) | 1 | 8.43 |
| | StyleGAN-XL (Sauer et al., 2022) | 1 | 2.41 |
| | SiD$^2$A (Zhou et al., 2025b) | 1 | 1.37 |
| Diffusion | ADM (Dhariwal & Nichol, 2021) | 250 | 23.24 |
| | ADM-U (Dhariwal & Nichol, 2021) | 500 | 9.96 |
| | DiT-XL/2 (Peebles & Xie, 2023) | 250 | 12.03 |
| | SiT-XL (Ma et al., 2024) | 250 | 8.30 |
| | RiN (Jabri et al., 2023) | 1000 | 3.95 |
| | U-ViT-L (Bao et al., 2023) | 512 | 3.54 |
| | VDM++ (Kingma & Gao, 2023) | 512 | 2.99 |
| | EDM2-S (Karras et al., 2024b) | 63 | 1.73 |
| | EDM2-XXL (Karras et al., 2024b) | 63 | 1.91 |
| Masked | MAGVIT-v2 (Yu et al., 2024) | 64 | 3.07 |
| | MAR-L (Li et al., 2024) | 1024 | 2.74 |
| AR | VAR-d36-s (Tian et al., 2024) | 10 | 2.63 |
| | xAR-L (Ren et al., 2025) | 50 | 1.70 |
| Post-hoc | EDM2-S + SIMS (Alemohammad et al., 2024b) | 63 | 1.73 |
| | EDM2-L + DDO (Zheng et al., 2025) | 63 | 1.21 |
| | EDM2 + AG (Karras et al., 2024a) | 63 | 1.25 |
| | EDM2 + SiD$^2$A (Zhou et al., 2025b) | 1 | 1.37 |
| Ours | VAR-d30-s + **Neon** | 10 | 1.70 |

We summarize our results and provide a comprehensive comparison with state-of-the-art generative models in Table A.1. The following section discusses Neon's performance on each benchmark in more detail, highlighting its standing relative to top-performing models and other post-hoc methods.

**CIFAR-10**     On both conditional and unconditional CIFAR-10, Neon improves the EDM-VP baseline to a **1.38 FID** while maintaining its 35 NFE (Karras et al., 2024b). In the conditional setting, this is competitive with DDO, which achieves a 1.30 FID from the same base model but requires significantly more training compute ( 12% extra vs. Neon's 1.75%) (Zheng et al., 2025). In the unconditional setting, Neon's 1.38 FID is identical to DDO's and close to the SOTA held by SIMS at 1.33 FID (Alemohammad et al., 2024b). Notably, SIMS requires doubling the NFE to 70, making Neon a more sampling-efficient alternative. Neon also demonstrates versatility by improving a Flow Matching model to a 2.32 FID (Lipman et al., 2023).

**FFHQ-64x64**     On FFHQ, Neon significantly enhances the unconditional EDM-VP model, lowering its FID from 2.39 to **1.12** with 79 NFE. While the state-of-the-art is held by the one-step D2O-F at 0.85 FID (Zheng & Yang, 2025), Neon's performance is highly competitive. It stands against other post-hoc methods like SIMS (1.04 FID, 158 NFE) (Alemohammad et al., 2024b) and the one-step distilled SiD$^2$A (1.04 FID, 1 NFE) (Zhou et al., 2025b). Neon achieves its strong result with a simple parameter merge that preserves the base sampler's structure, offering a distinct trade-off between FID and NFE.

**ImageNet-256x256**     On ImageNet-256, Neon sets a new **state-of-the-art**, improving the xAR-L model from an already strong 1.28 FID to **1.02 FID** (Ren et al., 2025). This surpasses the previous best result of 1.06 FID from REPA-UCGM (Sun et al., 2025). Neon also demonstrates its superiority over DDO on this benchmark; when applied to the same VAR-d16 base model (Tian et al., 2024), Neon achieves a 2.01 FID, which is a significant improvement over DDO's 2.54 FID (Zheng et al., 2025). Furthermore, Neon consistently improves other architectures, including xAR-B (1.31 FID) and IMM (1.46 FID).

**ImageNet-512x512**     On ImageNet-512, Neon improves the VAR-d30 model to a **1.70 FID** with 10 NFE (Tian et al., 2024). While the state-of-the-art belongs to EDM2-L+DDO at 1.21 FID (Zheng et al., 2025), Neon's result is competitive with other post-hoc methods applied to different base models, such as EDM2-S+SIMS (1.73 FID) (Alemohammad et al., 2024b). It showcases Neon's ability to enhance autoregressive models at higher resolutions with its characteristic low compute overhead.

**Summary**     Across all benchmarks, Neon proves to be a simple, efficient, and broadly applicable post-hoc method for improving generative models. It achieves a new state-of-the-art on ImageNet-256 and delivers highly competitive results elsewhere, often with superior sampling efficiency compared to other post-hoc techniques. A key finding is that Neon's effectiveness corresponds directly to the quality of the base model it enhances; applying it to a stronger foundation like xAR-L yields a greater improvement and the best overall performance. This positions Neon as a reliable tool for adding a final layer of polish to strong, pre-existing generative models with minimal computational effort. Crucially, since Neon improves the base diffusion model itself, its benefits are potentially orthogonal to distillation methods; one could apply SiD$^2$A or D2O-F to the Neon-enhanced model for further gains.

## B PROOFS AND DETAILED EXPLANATIONS

This appendix provides the complete theoretical foundation for Neon. We begin with a simple Gaussian warmup example that demonstrates Neon's core mechanism with minimal mathematical machinery. We then present a roadmap outlining the logical structure of our proofs, followed by the detailed technical derivations.

### B.1 WARMUP: GAUSSIAN ESTIMATION WITH FINITE DATA

This subsection gives a self-contained, concrete instance of Neon in the simplest setting: estimating the mean and covariance of a Gaussian distribution from limited data. The goal is to show—with minimal machinery—why a *tiny* Neon step reduces the *population* (test) loss in expectation. The calculation also clarifies the role of mode-seeking synthesis and the sign appearing in the main theorems.

**Population objective and finite-sample MLE.** Let the ground-truth law be $p_*(x) = \mathcal{N}(\mu_*, \Sigma_*)$ in $\mathbb{R}^D$. We fit the model family $q_{\mu,\Sigma}(x) = \mathcal{N}(x \mid \mu, \Sigma)$ by minimizing the population negative log-likelihood

$$
\begin{aligned}
L_{\text{pop}}(\mu, \Sigma) &= \mathbb{E}_{x \sim p_*}\big[-\log q_{\mu,\Sigma}(x)\big] \\
&= \tfrac{1}{2}\big(\text{logdet}\,\Sigma + \text{tr}\big(\Sigma^{-1}(\Sigma_* + (\mu - \mu_*)(\mu - \mu_*)^\top)\big) + D\log(2\pi)\big),
\end{aligned}
$$

which is minimized at $(\mu, \Sigma) = (\mu_*, \Sigma_*)$.

In practice we observe $n$ samples $x_1, \ldots, x_n \sim p_*$ and use the *finite-sample MLE*

$$
\hat{\mu} = \tfrac{1}{n}\sum_{i=1}^{n} x_i, \qquad \hat{\Sigma} = \tfrac{1}{n}\sum_{i=1}^{n}(x_i - \hat{\mu})(x_i - \hat{\mu})^\top.
$$

For Gaussian data, $\hat{\mu}$ and $\hat{\Sigma}$ are independent; moreover $n\hat{\Sigma} \sim \text{Wishart}(\Sigma_*, \nu = n - 1)$.

**Synthetic objective.** To model the "mode-seeking" behavior of common samplers in a controlled way, define the synthetic law

$$
p_{\text{syn}}(x \mid \hat{\mu}, \hat{\Sigma}, \tau) := \mathcal{N}\big(x \mid \hat{\mu}, \tau^2 \hat{\Sigma}\big), \qquad \tau > 0,
$$

so $0 < \tau < 1$ shrinks variability (mode-seeking) and $\tau > 1$ expands variability (diversity-seeking). Given $(\mu, \Sigma)$, the synthetic negative log-likelihood is

$$
\begin{aligned}
L_{\text{syn}}(\mu, \Sigma; \mu_r, \Sigma_r, \tau) &= \mathbb{E}_{x \sim p_{\text{syn}}}\big[-\log q_{\mu,\Sigma}(x)\big] \\
&= \tfrac{1}{2}\big((\mu - \mu_r)^\top \Sigma^{-1}(\mu - \mu_r) + \text{tr}\big(\Sigma^{-1}\tau^2\Sigma_r\big) + \text{logdet}\,\Sigma + D\log(2\pi)\big).
\end{aligned}
$$

Equivalently, with $C_{\text{syn}} = \tau^2\hat{\Sigma} + (\mu - \hat{\mu})(\mu - \hat{\mu})^\top$,

$$
L_{\text{syn}}(\mu, \Sigma; \hat{\mu}, \hat{\Sigma}, \tau) = \tfrac{1}{2}\big(\text{logdet}\,\Sigma + \text{tr}\big(\Sigma^{-1}C_{\text{syn}}\big) + D\log(2\pi)\big).
$$

Taking derivatives gives the standard Gaussian gradients

$$
\nabla_\mu L_{\text{syn}}(\mu, \Sigma) = \Sigma^{-1}(\mu - \hat{\mu}), \qquad \nabla_\Sigma L_{\text{syn}}(\mu, \Sigma) = \tfrac{1}{2}\big(\Sigma^{-1} - \Sigma^{-1}C_{\text{syn}}\Sigma^{-1}\big).
$$

Evaluated at the MLE $(\hat{\mu}, \hat{\Sigma})$,

$$
\nabla_\mu L_{\text{syn}}(\hat{\mu}, \hat{\Sigma}) = 0, \qquad \nabla_\Sigma L_{\text{syn}}(\hat{\mu}, \hat{\Sigma}) = \tfrac{1}{2}\big(\hat{\Sigma}^{-1} - \hat{\Sigma}^{-1}(\tau^2\hat{\Sigma})\hat{\Sigma}^{-1}\big) = \tfrac{1}{2}(1 - \tau^2)\hat{\Sigma}^{-1}. \quad \text{(B.1)}
$$

Thus the synthetic step and Neon leave the mean unchanged to first order: $\mu_s = \hat{\mu}$, $\mu_{\text{Neon}} = \hat{\mu}$ (up to $o(\eta)$).

**A tiny synthetic step and the Neon flip.** Starting from $(\hat{\mu}, \hat{\Sigma})$, take an infinitesimal gradient step on $L_{\text{syn}}$ with rate $\eta$:

$$\Sigma_s = \hat{\Sigma} - \eta\,\nabla_\Sigma L_{\text{syn}}(\hat{\mu}, \hat{\Sigma}) + o(\eta) \;=\; \hat{\Sigma} - \tfrac{\eta}{2}(1 - \tau^2)\,\hat{\Sigma}^{-1} + o(\eta),$$

i.e., the synthetic self-train moves *inward* when $0 < \tau < 1$. Neon then *flips that tiny move* via a linear merge with weight $w > 0$:

$$\Sigma_{\text{Neon}} = (1 + w)\hat{\Sigma} - w\Sigma_s = \hat{\Sigma} + \underbrace{\tfrac{1}{2}(1 - \tau^2)w\eta}_{\alpha > 0}\,\hat{\Sigma}^{-1} + o(\eta) = \hat{\Sigma} + \alpha\hat{\Sigma}^{-1} + o(\eta)$$

To first order, Neon applies $\Delta\Sigma_{\text{Neon}} = \alpha\,\hat{\Sigma}^{-1}$ with small $\alpha > 0 \propto (1 - \tau^2)w\eta$.

**First-order change in population loss.** Compare $L_{\text{pop}}$ at $(\hat{\mu}, \hat{\Sigma})$ and $(\hat{\mu}, \Sigma_{\text{Neon}})$. A first-order expansion yields

$$\Delta L_{\text{pop}} := L_{\text{pop}}(\hat{\mu}, \Sigma_{\text{Neon}}) - L_{\text{pop}}(\hat{\mu}, \hat{\Sigma}) \;\approx\; \langle\nabla_\Sigma L_{\text{pop}}(\hat{\mu}, \hat{\Sigma}), \Delta\Sigma_{\text{Neon}}\rangle, \qquad \langle A, B\rangle = \text{tr}(A^\top B).$$

We now average over the randomness of $(\hat{\mu}, \hat{\Sigma})$ from finite sampling. Two classical facts for Gaussian data (valid for $n > D + 4$) are sufficient:
(i) $\hat{\mu}$ and $\hat{\Sigma}$ are independent with $\mathbb{E}[(\hat{\mu} - \mu_*)(\hat{\mu} - \mu_*)^\top] = \Sigma_*/n$;
(ii) $n\hat{\Sigma}$ is Wishart with scale $\Sigma_*$ and $\nu = n - 1$, so

$$\mathbb{E}[\hat{\Sigma}^{-1}] = \frac{n}{n - D - 2}\,\Sigma_*^{-1}, \qquad \mathbb{E}[\hat{\Sigma}^{-1}\Sigma_*\hat{\Sigma}^{-1}] = \frac{n^2}{(n - D - 2)(n - D - 4)}\,\Sigma_*^{-1}.$$

Using $\nabla_\Sigma L_{\text{pop}} = \tfrac{1}{2}(\Sigma^{-1} - \Sigma^{-1}C_*\Sigma^{-1})$ with $C_* = \Sigma_* + (\hat{\mu} - \mu_*)(\hat{\mu} - \mu_*)^\top$, independence gives

$$\mathbb{E}\big[\nabla_\Sigma L_{\text{pop}}(\hat{\mu}, \hat{\Sigma})\big] = \tfrac{1}{2}\Big(\mathbb{E}[\hat{\Sigma}^{-1}] - \Big(1 + \tfrac{1}{n}\Big)\mathbb{E}[\hat{\Sigma}^{-1}\Sigma_*\hat{\Sigma}^{-1}]\Big) = -\gamma_{\text{MLE}}\,\Sigma_*^{-1},$$

with

$$\gamma_{\text{MLE}} = \frac{n(D + 5)}{2\,(n - D - 2)(n - D - 4)} \;>\; 0.$$

Combining with $\Delta\Sigma_{\text{Neon}} = \alpha\,\hat{\Sigma}^{-1} + o(\eta)$,

$$\mathbb{E}[\Delta L_{\text{pop}}] \;\approx\; \big\langle -\gamma_{\text{MLE}}\,\Sigma_*^{-1},\, \alpha\,\mathbb{E}[\hat{\Sigma}^{-1}]\big\rangle = -\alpha\,\gamma_{\text{MLE}}\,\frac{n}{n - D - 2}\,\text{tr}(\Sigma_*^{-2}) \;+\; o(\alpha) \;<\; 0.$$

> **Conclusion:** for sufficiently small $\alpha > 0$, $\;\mathbb{E}\Big[L_{\text{pop}}(\hat{\mu}, \Sigma_{\text{Neon}}) - L_{\text{pop}}(\hat{\mu}, \hat{\Sigma})\Big] \;<\; 0.$

**Geometric intuition.** The synthetic step systematically identifies the local "tightening" direction (because $0 < \tau < 1$ shrinks variability); Neon flips that direction by a tiny amount. On average, the population gradient at $(\hat{\mu}, \hat{\Sigma})$ prefers a small *de-tightening*, so the inner product is negative and the population loss drops. In this toy case, the mean does not move: $\mu_{\text{Neon}} = \hat{\mu}$ to first order.

**Remarks on $\tau > 1$ and on not optimizing $L_{\text{syn}}$ to convergence.** If the synthetic law is *diversity-seeking* ($\tau > 1$), the sign in Eq. equation B.1 reverses and the first-order Neon update becomes $\Delta\Sigma_{\text{Neon}} = \alpha\,\hat{\Sigma}^{-1}$ with $\alpha \propto (1 - \tau^2)w\eta < 0$ for $w, \eta > 0$. In practice, one simply tunes the *signed* merge strength $\alpha$ (equivalently, the sign of $w$) so that $\alpha > 0$; after this tuning, the same conclusion holds, and the correction induced by diversity-seeking synthesis coincides with the mode-seeking case at the level of the final Neon update.

Finally, we are *not* interested in minimizing the synthetic objective $L_{\text{syn}}$ to its exact optimum. In this Gaussian toy, the minimizer is $(\mu, \Sigma) = (\hat{\mu}, \tau^2\hat{\Sigma})$. If one trained to that point and then merged, the result would be a pure scalar rescaling

$$\Sigma_{\text{Neon}} = (1 + w)\hat{\Sigma} - w(\tau^2\hat{\Sigma}) = \big(1 + w(1 - \tau^2)\big)\,\hat{\Sigma},$$

which carries no additional directional information beyond $\hat{\Sigma}$. What Neon needs is only a *robust estimate of the local gradient direction*; hence we use a very small synthetic training budget so the direction is measured in the local regime and the negative extrapolation is reliable.

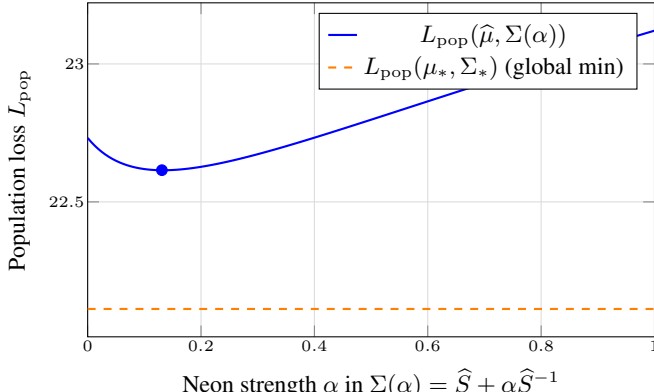

Figure B.1: Population loss $L_{\text{pop}}(\widehat{\mu}, \Sigma(\alpha))$ along the Neon path for a Gaussian with $D = 10$, $n = 100$. The dashed line shows the global minimum (requires true parameters). The optimal $\alpha$ (blue dot) substantially improves over the baseline empirical estimator.

Finally, we validate our theoretical framework on a controlled synthetic task where ground truth is known. We generate $n = 100$ samples from a $D = 10$ dimensional Gaussian $\mathcal{N}(\mu_*, \Sigma_*)$ and evaluate the population loss $L_{\text{pop}}(\widehat{\mu}, \Sigma(\alpha))$ along the Neon path $\Sigma(\alpha) = \widehat{S} + \alpha \widehat{S}^{-1}$ for $\alpha \in [0, 1]$. Figure B.1 shows that Neon improves over the empirical baseline ($\alpha = 0$), with the optimal $\alpha$ achieving reduction in population loss. The dashed line indicates the global minimum $L_{\text{pop}}(\mu_*, \Sigma_*)$, unattainable without knowledge of true parameters. The gap between the optimal Neon loss and this theoretical minimum reflects the fundamental constraint of finite-sample estimation within our parameter family, consistent with our theoretical predictions.

## B.2    ROADMAP: LOGICAL STRUCTURE OF THE PROOFS

The Gaussian warmup illustrates Neon's mechanism in a transparent setting. We now outline how the general theory extends these ideas to arbitrary generative models. The proofs are organized in five conceptual stages:

**Stage 1: Framework and first-order analysis (Sections B.3–B.4).**    We establish notation, define the alignment scalar $s = \langle r_d, P r_s \rangle$ (where $r_d, r_s$ are the real-data and synthetic gradients), and prove that Neon reduces population risk when $s < 0$.

**Key result:** Theorem B.1 shows

$$\mathcal{R}_{\text{data}}(\theta_{\text{Neon}}) = \mathcal{R}_{\text{data}}(\theta_r) + w\alpha\, s + \frac{(w\alpha)^2}{2} r_s^\top P^\top \widehat{H}_d P r_s + O((w\alpha)^3),$$

establishing that the sign of $s$ determines whether Neon helps ($s < 0$) or harms ($s > 0$).

**Stage 2: Geometric characterization of anti-alignment (Section B.5).**    We decompose the synthetic gradient into signal and bias components, introducing the angle $\phi$ between the model error $\varepsilon = \theta_r - \theta^*$ and the sampler-induced bias $b = \mathbb{E}_{q_{\theta_r, \kappa}}[\phi_{\theta^*}(X)]$.

**Key result:** Theorem B.4 provides a directional upper bound

$$s \leq M(1 + \eta_1)\|\varepsilon\|_{H_d}^2 - m\eta_0\|\varepsilon\|_{H_d}[-\cos\varphi]_+ + O(\|\varepsilon\|_{H_d}^3),$$

showing that when $\cos\varphi < 0$ (obtuse angle) and $\|\varepsilon\|_{H_d}$ is small, the linear term dominates and $s < 0$.

**Stage 3: Mode-seeking samplers induce anti-alignment (Sections B.6–B.7).**    We prove that all commonly-used inference procedures are mode-seeking and therefore guarantee $\cos\varphi < 0$ near good models.

**Key results:**

- **Autoregressive models** (Lemma B.6): Temperature $\tau < 1$, top-$k$, and top-$p$ sampling all induce $q(x) \propto f(\log p_{\theta_r}) p_{\theta_r}$ with $f$ nondecreasing, which implies $\cos \varphi < 0$ to first order in $\|\varepsilon\|_F$.
- **Diffusion/flow models** (Section B.7): Any finite-step ODE solver introduces discretization error that is mode-seeking (under mild curvature-density coupling assumptions), yielding $\cos \varphi < 0$ even at the neutral point (no explicit guidance).

Together, these results establish that $s < 0$ holds for all standard inference configurations, making Neon universally applicable.

**Stage 4: Complete characterization—diversity-seeking samplers (Section B.9).** We complete the theory by characterizing diversity-seeking samplers ($\tau > 1$ for AR, or inverse truncation), which reverse the gradient alignment to $\cos \varphi \geq 0$, yielding $s > 0$.

**Key result:** Theorem B.13 provides a lower bound showing that diversity-seeking configurations yield $s \geq (m - M\eta_1)\|\varepsilon\|_{H_d}^2 + O(\|\varepsilon\|_{H_d})$. In this regime, the Neon formula $\theta_{\text{Neon}} = (1+w)\theta_r - w\theta_s$ improves performance for $-1 < w < 0$ (interpolation toward $\theta_s$) rather than $w > 0$ (extrapolation away from $\theta_s$). Thus the sign of $s$ determines the sign of the optimal merge weight, and Neon's framework applies universally by tuning $w$ appropriately.

**Stage 5: Finite-sample effects and hyperparameter guidance (Sections B.10–B.11).** We analyze the impact of finite synthetic datasets $|\mathcal{S}|$ and provide theoretical guidance for choosing the training budget $\mathcal{B}$ and merge weight $w$.

**Key results:**

- Lemma B.17 shows that short fine-tuning (small $\alpha T$) with moderate $|\mathcal{S}|$ produces a low-variance, reliable estimate of the degradation direction $-Pr_s^{(\mathcal{S})}(\theta_r)$.
- Very small $|\mathcal{S}|$ suffers from high variance; very large $|\mathcal{S}|$ amplifies curvature effects via the quadratic term. This predicts the U-shaped performance observed empirically in Figure 3.
- Controlled toy experiments (Section B.11, Figures B.2 and B.3) directly validate the theory by measuring gradient alignment as a function of sampler type.

**Summary.** The complete proof proceeds as follows:

1. Show that Neon helps when $s < 0$ (Stage 1)
2. Express $s$ in terms of the geometric angle $\varphi$ between model error and sampler bias (Stage 2)
3. Prove that all standard samplers are mode-seeking, yielding $\cos \varphi < 0 \Rightarrow s < 0$ (Stage 3)
4. Characterize diversity-seeking samplers where $s > 0$ and optimal $w$ changes sign (Stage 4)
5. Provide finite-sample guidance and empirical validation (Stage 5)

This roadmap should guide the reader through the technical details that follow. Each subsection refines and formalizes one step in the logical chain, culminating in a complete characterization of when and why Neon works.

## B.3 ASSUMPTIONS, NOTATION, AND IDENTITIES

This subsection establishes the mathematical framework for our analysis. We introduce the basic objects (loss, risk, gradients), state our standing assumptions, define the geometric structures used throughout the proofs, and provide a comprehensive notation reference table.

**Basic setup.** Let $\ell_\theta(x)$ be a differentiable per-example loss. The *population risk* (or data risk) is

$$\mathcal{R}_{\text{data}}(\theta) := \mathbb{E}_{p_{\text{data}}}[\ell_\theta(X)].$$

The per-example gradient is denoted

$$\phi_\theta(x) := \nabla_\theta \ell_\theta(x) \in \mathbb{R}^p,$$

and its Jacobian (with respect to $\theta$) is

$$H_\theta(x) \;:=\; \partial_\theta \phi_\theta(x) \;=\; \nabla^2_\theta \ell_\theta(x) \in \mathbb{R}^{p \times p}.$$

The *data Hessian* (population Hessian at the optimum) is

$$H_d \;:=\; \nabla^2 \mathcal{R}_{\text{data}}(\theta^*) \;=\; \mathbb{E}_{p_{\text{data}}}[H_{\theta^*}(X)].$$

**Standing assumptions.** Throughout the appendix, we maintain the following assumptions:

(A1) **Optimality.** There exists a population risk minimizer $\theta^* \in \arg\min_\theta \mathcal{R}_{\text{data}}(\theta)$, which satisfies the first-order condition

$$\mathbb{E}_{p_{\text{data}}}[\phi_{\theta^*}(X)] \;=\; 0.$$

(A2) **Regularity.** The data distribution $p_{\text{data}}$ and model family satisfy:

- Common support between $p_{\text{data}}$ and the model distributions;
- Dominated convergence holds, allowing interchange of limits and expectations;
- The gradient map $\phi_\theta$ and Jacobian map $H_\theta(x)$ are locally Lipschitz continuous near $\theta^*$.

(A3) **Local analysis.** Our base model $\theta_r$ lies in a small neighborhood of the optimum:

$$\theta_r \;=\; \theta^* + \varepsilon \quad \text{with} \quad \|\varepsilon\|_{H_d} \text{ small,}$$

and all remainder terms are $O(\|\varepsilon\|^2_{H_d})$ or smaller.

(A4) **Rank condition.** If $H_d$ is not full rank, all statements are understood to hold on the image space $\text{Im}(H_d)$.

**Geometric structures.** For any positive-definite matrix $M \succ 0$, we define the $M$-induced geometry:

$$\text{Inner product:} \quad \langle x, y \rangle_M \;:=\; x^\top M y,$$

$$\text{Norm:} \quad \|x\|_M \;:=\; \sqrt{x^\top M x} \;=\; \|M^{1/2} x\|_2,$$

$$\text{Operator norm:} \quad \|A\|_{\text{op},M} \;:=\; \|M^{1/2} A M^{-1/2}\|_{\text{op}}.$$

When $M = H_d$, we write $\|\cdot\|_{H_d}$ and $\langle\cdot,\cdot\rangle_{H_d}$ for brevity.

For any preconditioner $P \succ 0$ (e.g., the inverse Hessian, an Adam-like diagonal, or identity), define the *whitened preconditioner*

$$K \;:=\; H_d^{1/2} P H_d^{1/2},$$

which satisfies spectral bounds

$$mI \;\preceq\; K \;\preceq\; MI \quad \text{for some} \quad 0 < m \leq M < \infty.$$

These constants $m$ and $M$ control the condition number of the preconditioned system and appear throughout our bounds.

**Notation reference.** Table B.1 provides a comprehensive reference for all notation used in this appendix. Symbols are grouped thematically for easier lookup during proofs.

**Conventions.**

- Scalars: lowercase $(s, w, \alpha)$; vectors: lowercase bold or Greek $(\theta, \varepsilon, r_d)$; matrices: uppercase $(H_d, P, K)$.
- Subscripts: $r$ (real/base), $s$ (synthetic), $*$ (optimal).
- All norms default to Euclidean unless a subscript specifies otherwise (e.g., $\|\cdot\|_{H_d}$).

### B.4 NEON IMPROVES UNDER ANTI-ALIGNMENT

Having established our framework and notation, we now prove the central result: Neon reduces population risk when the synthetic and real-data gradients are *anti-aligned*. We begin by defining the key quantity that governs Neon's behavior—the alignment scalar $s$—and then show how its sign determines whether Neon helps or harms.

Table B.1: Notation reference table

| Symbol | Description | Symbol | Description |
|--------|-------------|--------|-------------|
| *Data, models, and parameters* | | | |
| $p_{\text{data}}$ | Ground-truth data distribution | $\theta^*$ | Population risk minimizer |
| $G_\theta$ | Generative model | $\theta_r$ | Base model, $\theta_r = \theta^* + \varepsilon$ |
| $p_\theta$ | Model distribution | $\theta_s$ | Self-trained model |
| $q_{\theta,\kappa}$ | Inference distribution | $\theta_{\text{Neon}}$ | Neon model, $(1+w)\theta_r - w\theta_s$ |
| $\varepsilon$ | Model error, $\theta_r - \theta^*$ | $\kappa$ | Inference hyperparameters |
| *Losses, gradients, and Hessians* | | | |
| $\ell_\theta(x)$ | Per-example loss | $r_d$ | Real-data gradient |
| $\mathcal{R}_{\text{data}}(\theta)$ | Population risk | $r_s$ | Synthetic gradient |
| $\mathcal{R}_{\text{syn}}(\theta)$ | Synthetic risk | $r_s^{(\mathcal{S})}$ | Empirical synthetic gradient |
| $\phi_\theta(x)$ | Per-example gradient | $s$ | Alignment scalar, $\langle r_d, Pr_s \rangle$ |
| $H_\theta(x)$ | Per-example Hessian | $b$ | Sampler bias |
| $H_d$ | Data Hessian | $\Delta$ | Curvature tilt |
| *Geometric structures* | | | |
| $\langle x, y \rangle_M$ | $M$-inner product, $x^\top M y$ | $\eta_0$ | Linear bias, $\|b\|_{H_d^{-1}}$ |
| $\|x\|_M$ | $M$-norm, $(x^\top M x)^{1/2}$ | $\eta_1$ | Curvature tilt, $\|\Delta\|_{\text{op},H_d^{-1}}$ |
| $\|A\|_{\text{op},M}$ | $M$-operator norm | $\varphi$ | Angle between $\varepsilon$ and $H_d^{-1}b$ |
| *Optimization* | | | |
| $P$ | Preconditioner, $P \succ 0$ | $m, M$ | Spectral bounds, $mI \preceq K \preceq MI$ |
| $K$ | Whitened preconditioner, $H_d^{1/2}PH_d^{1/2}$ | $\alpha$ | Step size (learning rate) |
| $w$ | Merge weight | $T$ | Number of fine-tuning steps |
| $w^*$ | Optimal merge weight | $\mathcal{B}$ | Training budget (millions of images) |
| *Synthetic data and sampling* | | | |
| $\mathcal{S}$ | Synthetic dataset | $\tau$ | Temperature (AR sampling) |
| $|\mathcal{S}|, n$ | Synthetic dataset size | $\zeta$ | Score scale (toy experiments) |
| $\gamma$ | CFG scale | FID | Fréchet Inception Distance |
| *Model-specific notation* | | | |
| $u_\theta(x)$ | AR score, $\nabla_\theta \log p_\theta(x)$ | $s_\theta(x_t, t)$ | Diffusion score function |
| $F$ | Fisher matrix for AR | $v_\theta(x_t, t)$ | Flow matching velocity |
| *Operators and conventions* | | | |
| $\mathbb{E}[\cdot]$ | Expectation | $\|\cdot\|_{\text{Fr}}$ | Frobenius norm |
| $\text{tr}(\cdot)$ | Trace | $[\cdot]_+$ | Positive part, $\max(\cdot, 0)$ |
| $O(\cdot), o(\cdot)$ | Big-O, little-o | $A \succeq 0$ | PSD matrix |

**Alignment scalar and synthetic objective.** We work with two gradient directions evaluated at the base model $\theta_r$. The *real-data gradient* points toward improving performance on the true data distribution:

$$r_d := \nabla_\theta \mathcal{R}_{\text{data}}(\theta)\big|_{\theta_r}.$$

The *synthetic gradient* is computed with respect to the synthetic risk, which measures performance on samples drawn from the model's own inference procedure:

$$\mathcal{R}_{\text{syn}}(\theta) := \mathbb{E}_{q_{\theta_r,\kappa}}[\ell_\theta(X)], \qquad r_s := \nabla_\theta \mathcal{R}_{\text{syn}}(\theta)\big|_{\theta_r}.$$

The central quantity controlling Neon's effectiveness is their preconditioned inner product, which we call the *alignment scalar*:

$$s := \langle r_d, \ P r_s \rangle. \tag{B.2}$$

When $s < 0$, the gradients are *anti-aligned*: moving in the direction $-Pr_s$ (away from synthetic fine-tuning) actually *improves* the real-data risk. This anti-alignment is the key insight that makes Neon work.

**Main result: Neon reduces risk when $s < 0$.** We now formalize how Neon exploits anti-alignment. The following theorem provides a Taylor expansion of the risk after Neon's parameter merge, showing that the sign of $s$ determines whether performance improves.

**Theorem B.1** (One-step Neon improvement). *A short synthetic fine-tune produces $\theta_s = \theta_r - \alpha\,Pr_s + O(\alpha^2)$ for some $\alpha > 0$. For $w > 0$, the Neon merge is*

$$\theta_{Neon} \;=\; (1+w)\theta_r - w\theta_s \;=\; \theta_r + w\alpha\,Pr_s + O(w\alpha^2).$$

*Let $\widehat{H}_d := \nabla^2 \mathcal{R}_{\text{data}}(\theta_r)$. Then*

$$\mathcal{R}_{\text{data}}(\theta_{Neon}) \;=\; \mathcal{R}_{\text{data}}(\theta_r) \;+\; w\alpha\,s \;+\; \frac{(w\alpha)^2}{2}\, r_s^\top P^\top \widehat{H}_d\, P\, r_s \;+\; O\big((w\alpha)^3\big). \tag{B.3}$$

*In particular, if $s < 0$ then for all sufficiently small $w > 0$ we have $\mathcal{R}_{\text{data}}(\theta_{Neon}) < \mathcal{R}_{\text{data}}(\theta_r)$. If moreover $\widehat{H}_d \succeq 0$, writing $q := r_s^\top P^\top \widehat{H}_d P r_s \geq 0$, any*

$$0 \;<\; w \;<\; -\frac{2s}{\alpha q} \quad \text{guarantees} \quad \mathcal{R}_{\text{data}}(\theta_{Neon}) \leq \mathcal{R}_{\text{data}}(\theta_r) \;\; (\text{up to } O((w\alpha)^3)),$$

*and the quadratic proxy is minimized at $w^* = -s/(\alpha q) > 0$.*

*Proof.* The proof proceeds in three steps: (i) we express the Neon update as a step in the synthetic gradient direction, (ii) we perform a Taylor expansion of the risk along this direction, and (iii) we analyze when the expansion is negative.

*Step 1: Neon as a directional step.* From the short synthetic fine-tune we have

$$\theta_s = \theta_r - \alpha Pr_s + O(\alpha^2).$$

Therefore

$$\theta_{\text{Neon}} = (1+w)\theta_r - w\theta_s = \theta_r + w\alpha\,Pr_s \;+\; O(w\alpha^2).$$

Thus Neon moves in the direction $Pr_s$—the *opposite* of the synthetic gradient direction.

*Step 2: Taylor expansion along the Neon direction.* Define the univariate function

$$\psi(\tau) := \mathcal{R}_{\text{data}}\big(\theta_r + \tau\,Pr_s\big), \qquad \text{and set } \tau = w\alpha.$$

A Taylor expansion of $\psi$ at $\tau = 0$ gives

$$\psi(\tau) = \psi(0) \;+\; \tau\,\psi'(0) \;+\; \frac{\tau^2}{2}\,\psi''(0) \;+\; O(\tau^3).$$

By the chain rule,

$$\psi'(0) = \big\langle r_d,\; Pr_s \big\rangle \;=\; s, \qquad \psi''(0) = r_s^\top P^\top \widehat{H}_d\, P\, r_s.$$

Substituting $\tau = w\alpha$ yields

$$\mathcal{R}_{\text{data}}(\theta_{\text{Neon}}) = \mathcal{R}_{\text{data}}(\theta_r) \;+\; w\alpha\,s \;+\; \frac{(w\alpha)^2}{2}\, r_s^\top P^\top \widehat{H}_d P r_s \;+\; O\big((w\alpha)^3\big),$$

which is equation B.3.

*Step 3: When does the risk decrease?* If $s < 0$, the linear term $w\alpha s$ is negative and dominates for sufficiently small $w > 0$, giving $\mathcal{R}_{\text{data}}(\theta_{\text{Neon}}) < \mathcal{R}_{\text{data}}(\theta_r)$.

If, in addition, $\widehat{H}_d \succeq 0$, then $\psi''(0) \geq 0$ and the quadratic proxy $\tau \mapsto \psi(0) + \tau s + \frac{1}{2}\tau^2\,\psi''(0)$ is minimized at

$$\tau^* \;=\; -\frac{s}{\psi''(0)} \;>\; 0.$$

Since $\tau = w\alpha$, this gives the safe window $0 < w < -\frac{2s}{\alpha\,\psi''(0)}$ and the minimizer $w^* = -\frac{s}{\alpha\,\psi''(0)} = -\frac{s}{\alpha\, r_s^\top P^\top \widehat{H}_d P r_s}$. $\qquad\qquad\square$

**Interpretation.** Theorem B.1 reveals the fundamental trade-off in Neon. The expansion equation B.3 is a balance between:

- **Linear term** ($w\alpha s$): Negative when $s < 0$, this drives improvement.

- **Quadratic term** ($\frac{(w\alpha)^2}{2}q$): Always non-negative (when $\widehat{H}_d \succeq 0$), this penalizes moving too far.

For small $w$, the linear term dominates, so Neon helps. For large $w$, the quadratic term dominates, so performance degrades. The optimal $w^*$ balances these forces, and depends on both the alignment strength $|s|$ and the curvature $q$.

*Remark* B.2 (No convexity needed: directional smoothness). The PSD requirement on $\widehat{H}_d$ can be replaced by an upper curvature bound *along the step direction* $d := Pr_s$. If there is $L_{\mathrm{dir}} \geq 0$ with $d^\top \nabla^2 \mathcal{R}_{\mathrm{data}}(\theta_r + \tau d)d \leq L_{\mathrm{dir}}\|d\|_2^2$ for $\tau$ near $0$, then the same conclusion holds whenever $0 < w < -\frac{2s}{\alpha L_{\mathrm{dir}}\|d\|_2^2}$. This weaker condition is sufficient because Neon only moves along a single direction, so we only need smoothness along that ray rather than global convexity.

### B.5 AN UPPER BOUND ON $s$ AND SUFFICIENT CONDITIONS FOR ANTI-ALIGNMENT

We now pursue the question posed at the end of the previous subsection: *when does $s < 0$ hold?* The key is to decompose the synthetic gradient into two parts—a "signal" component that matches the real-data gradient, and a "bias" component induced by the sampler. We will show that the sign of $s$ depends on a geometric angle between the model error and this sampler-induced bias. This geometric characterization then allows us to identify simple, verifiable conditions that guarantee anti-alignment.

**Local expansion at $\theta_r$.** We begin by establishing first-order expansions of both gradients. The following lemma isolates the sampler-induced bias terms that drive anti-alignment.

**Lemma B.3** (First-order expansions of real and synthetic gradients). *Let $\theta_r = \theta^* + \varepsilon$ with $\|\varepsilon\|_{H_d}$ small and assume* (A1)–(A4). *Then*

$$r_d := \nabla_\theta \mathcal{R}_{\mathrm{data}}(\theta)\big|_{\theta_r} = H_d\,\varepsilon + O(\|\varepsilon\|_{H_d}^2), \tag{B.4}$$

*and, with*

$$b := \mathbb{E}_{q_{\theta_r,\kappa}}\big[\phi_{\theta^*}(X)\big], \qquad \Delta := \mathbb{E}_{q_{\theta_r,\kappa}}\big[H_{\theta^*}(X)\big] - \mathbb{E}_{p_{data}}\big[H_{\theta^*}(X)\big],$$

$$r_s := \nabla_\theta \mathcal{R}_{\mathrm{syn}}(\theta)\big|_{\theta_r} = H_d\,\varepsilon + \underbrace{(b + \Delta\,\varepsilon)}_{=:R_\kappa} + O(\|\varepsilon\|_{H_d}^2), \tag{B.5}$$

*Proof.* The proof proceeds by Taylor-expanding the per-example gradient $\phi_\theta$ around the optimum $\theta^*$, then taking expectations under the real and synthetic distributions.

*First-order expansion of the per-example gradient.* By (A2) (regularity) and a first-order Taylor expansion at $\theta^*$,

$$\phi_{\theta_r}(x) = \phi_{\theta^*}(x) + H_{\theta^*}(x)\,\varepsilon + \rho(x),$$

where the remainder satisfies $\mathbb{E}_{p_{\mathrm{data}}}\big[\|\rho(X)\|\big] = O(\|\varepsilon\|_{H_d}^2)$ and similarly $\mathbb{E}_{q_{\theta_r,\kappa}}\big[\|\rho(X)\|\big] = O(\|\varepsilon\|_{H_d}^2)$.

*Real-risk gradient.* Taking expectation under $p_{\mathrm{data}}$ and using (A1)–(A3),

$$r_d = \mathbb{E}_{p_{\mathrm{data}}}\big[\phi_{\theta_r}(X)\big] = \underbrace{\mathbb{E}_{p_{\mathrm{data}}}\big[\phi_{\theta^*}(X)\big]}_{=0} + \mathbb{E}_{p_{\mathrm{data}}}\big[H_{\theta^*}(X)\big]\varepsilon + \mathbb{E}_{p_{\mathrm{data}}}\big[\rho(X)\big] = H_d\,\varepsilon + O(\|\varepsilon\|_{H_d}^2).$$

The first term vanishes by optimality (A1), leaving only the Hessian-weighted error.

*Synthetic-risk gradient.* Taking expectation under $q_{\theta_r,\kappa}$,

$$r_s = \mathbb{E}_{q_{\theta_r,\kappa}}\big[\phi_{\theta_r}(X)\big] = \underbrace{\mathbb{E}_{q_{\theta_r,\kappa}}\big[\phi_{\theta^*}(X)\big]}_{=:b} + \underbrace{\mathbb{E}_{q_{\theta_r,\kappa}}\big[H_{\theta^*}(X)\big]}_{=H_d+\Delta}\varepsilon + \mathbb{E}_{q_{\theta_r,\kappa}}\big[\rho(X)\big].$$

Hence
$$r_s = b + (H_d + \Delta)\,\varepsilon \; + \; O(\|\varepsilon\|_{H_d}^2).$$
Crucially, the synthetic gradient has two extra terms compared to $r_d$: the *linear bias* $b$ (because the sampler does not satisfy the optimality condition), and the *curvature tilt* $\Delta\varepsilon$ (because the sampler changes the Hessian).

*Equivalent residual form used later.* It is convenient (and used in subsequent bounds) to rewrite this as
$$r_s = H_d\,\varepsilon \; - \; R_\kappa \; + \; O(\|\varepsilon\|_{H_d}^2), \qquad \text{where} \quad R_\kappa \; := \; -\big(b + \Delta\,\varepsilon\big).$$
Both expressions are identical up to the first-order terms, and the latter isolates the "useful" $H_d\varepsilon$ part from the sampler-induced mismatch $R_\kappa$. $\qquad\square$

**Geometric interpretation via angle and magnitudes.** The lemma shows that $r_s$ and $r_d$ share a common "signal" $H_d\varepsilon$, but differ by the sampler-induced residual $R_\kappa = -(b + \Delta\varepsilon)$. To understand when $s < 0$, we characterize the geometry of this residual. Define the $H_d$-whitened magnitudes
$$\eta_0 := \|b\|_{H_d^{-1}}, \qquad \eta_1 := \|\Delta\|_{\mathrm{op},\,H_d^{-1}},$$
which measure the strength of the linear bias and curvature tilt, respectively. The key geometric quantity is the angle between the model error $\varepsilon$ and the bias direction $H_d^{-1}b$:
$$\cos\varphi \; := \; \frac{\langle \varepsilon,\, H_d^{-1}b\rangle_{H_d}}{\|\varepsilon\|_{H_d}\,\big\|H_d^{-1}b\big\|_{H_d}} \; \in [-1, 1]. \tag{B.6}$$

Equivalently, $\varphi$ is the Euclidean angle between $H_d^{1/2}\varepsilon$ and $H_d^{-1/2}b$. Recall that the whitened preconditioner is $K := H_d^{1/2}PH_d^{1/2}$ with spectral bounds $mI \preceq K \preceq MI$.

**Main geometric result: when does $s < 0$?** We now express $s$ in terms of the angle $\varphi$ and the bias magnitudes. The following theorem provides an upper bound showing that $s < 0$ whenever $\varphi$ is obtuse and the model is sufficiently good.

**Theorem B.4** (Directional upper bound for $s$). *With $\theta_r = \theta^* + \varepsilon$ and $\|\varepsilon\|_{H_d}$ small,*
$$s \; \le \; M(1 + \eta_1)\,\|\varepsilon\|_{H_d}^2 \; - \; m\,\eta_0\,\|\varepsilon\|_{H_d}\big[-\cos\varphi\big]_+ \; + \; O(\|\varepsilon\|_{H_d}^3).$$
*Consequently, a sufficient condition for $s < 0$ is*

$$\boxed{\; \|\varepsilon\|_{H_d} \; < \; \frac{m\,\eta_0}{M(1+\eta_1)}\big(-\cos\varphi\big) \quad \text{with } \cos\varphi < 0. \;}$$

*Proof.* We compute $s = \langle r_d, Pr_s\rangle$ using the local expansions from Lemma B.3, then bound each term by whitening to Euclidean geometry.

Using Lemma B.3, write
$$s = \varepsilon^\top H_d P H_d \varepsilon \; - \; \varepsilon^\top H_d P b \; - \; \varepsilon^\top H_d P \Delta\varepsilon \; + \; O(\|\varepsilon\|_{H_d}^3).$$
Whiten with $a := H_d^{1/2}\varepsilon$, $\tilde{b} := H_d^{-1/2}b$, $\tilde{\Delta} := H_d^{-1/2}\Delta H_d^{-1/2}$, and $K := H_d^{1/2}PH_d^{1/2}$ to get
$$s \; = \; a^\top K a \; - \; a^\top K\tilde{b} \; - \; a^\top K\tilde{\Delta}\,a \; + \; O(\|a\|_2^3).$$
Now bound the three pieces using the spectral bounds on $K$:
$$a^\top K a \le M\|a\|_2^2 = M\|\varepsilon\|_{H_d}^2, \qquad -a^\top K\tilde{\Delta}a \le M\,\eta_1\,\|\varepsilon\|_{H_d}^2.$$
For the linear term, write $a^\top K\tilde{b} = \|K^{1/2}a\|_2\,\|K^{1/2}\tilde{b}\|_2\cos\theta$, with $\theta$ the angle between $K^{1/2}a$ and $K^{1/2}\tilde{b}$. Since $\|K^{1/2}x\|_2 \ge \sqrt{m}\|x\|_2$,
$$a^\top K\tilde{b} \; \ge \; m\,\|a\|_2\,\|\tilde{b}\|_2\,[\cos\theta]_+ \; = \; m\,\|\varepsilon\|_{H_d}\,\eta_0\,[\cos\varphi]_+.$$
Thus $-a^\top K\tilde{b} \le -m\,\eta_0\,\|\varepsilon\|_{H_d}[\cos\varphi]_+$. Since $[\cos\varphi]_+ \ge 0$ and $[-\cos\varphi]_+ \ge [\cos\varphi]_-$, we can replace $-[\cos\varphi]_+$ by the slightly looser but sign-robust term $-[-\cos\varphi]_+$, yielding the stated bound after collecting terms and absorbing $O(\|a\|_2^3)$. $\qquad\square$

**Corollary B.5** (Natural-gradient geometry). *If $P = H_d^{-1}$, then $K = I$ (so $m = M = 1$) and*
$$s \; \le \; (1 + \eta_1)\,\|\varepsilon\|_{H_d}^2 \; - \; \eta_0\,\|\varepsilon\|_{H_d}\big[-\cos\varphi\big]_+ \; + \; O(\|\varepsilon\|_{H_d}^3).$$
*Thus it suffices that $\|\varepsilon\|_{H_d} < \dfrac{\eta_0}{1 + \eta_1}\big(-\cos\varphi\big)$ with $\cos\varphi < 0$ to guarantee $s < 0$.*

**Interpretation: why obtuse angles guarantee anti-alignment.** Theorem B.4 reveals the structure of $s$ as a balance between two competing forces:

- **Quadratic terms** ($M(1 + \eta_1)\|\varepsilon\|_{H_d}^2$): Always positive, these tend to make $s > 0$.

- **Linear term** ($-m\eta_0\|\varepsilon\|_{H_d}[-\cos\varphi]_+$): Negative when $\cos\varphi < 0$, this drives $s$ negative.

The key insight is that $\eta_0$ captures the sampler's *linear bias* (whitened by $H_d$), while $\eta_1$ measures its *curvature tilt*. From Theorem B.4, the leading terms obey

$$s \;\lesssim\; M(1 + \eta_1)\,\|\varepsilon\|_{H_d}^2 \;-\; m\,\eta_0\,\|\varepsilon\|_{H_d}\,(-\cos\varphi),$$

so whenever the angle is *obtuse* ($\cos\varphi < 0$, i.e., $H_d^{-1}b$ points mostly *against* $\varepsilon$), the subtractive linear term eventually dominates as $\|\varepsilon\|_{H_d} \to 0$. Equivalently: there exists a threshold $\varepsilon_0 > 0$ (depending on $m, M, \eta_0, \eta_1$ and $-\cos\varphi$) such that if the model is sufficiently close to optimal, $\|\varepsilon\|_{H_d} < \varepsilon_0$, then $s < 0$. In this small-error regime, Neon reduces the real-data risk by Theorem B.1.

## B.6 Acute-angle conditions that imply $s < 0$ (AR models)

We now verify that the geometric condition $\cos\varphi < 0$ holds for autoregressive models under standard sampling procedures. The key insight is that common inference methods—temperature scaling, top-$k$, and top-$p$ sampling—are all *mode-seeking*: they systematically reweight the model distribution to favor high-probability regions. We prove that this mode-seeking behavior guarantees the obtuse angle condition, and hence anti-alignment.

**Loss and geometry (AR).** For autoregressive (AR) models we use negative log-likelihood:

$$\ell_\theta(x) = -\log p_\theta(x), \qquad \phi_\theta(x) = \nabla_\theta \ell_\theta(x) = -u_\theta(x),$$

so the data Hessian is the Fisher, $H_d = F = \mathbb{E}_{p_{\text{data}}}[u_{\theta^*} u_{\theta^*}^\top]$. For a sampler $q$ let

$$b \;:=\; \mathbb{E}_q\big[\phi_{\theta^*}(X)\big] \;=\; -\mathbb{E}_q\big[u_{\theta^*}(X)\big].$$

Our global angle is

$$\cos\varphi \;:=\; \frac{\langle \varepsilon,\, F^{-1}b\rangle_F}{\|\varepsilon\|_F\,\|F^{-1}b\|_F} \in [-1, 1],$$

so *anti-alignment* corresponds to $\cos\varphi < 0$.

**Definition (mode-seeking samplers).** Fix $\theta_r = \theta^* + \varepsilon$. We call $q$ *mode-seeking* if it is a monotone reweighting of the reference model:

$$q(x) \;\propto\; w(x)\, p_{\theta_r}(x), \qquad w(x) = f\big(\log p_{\theta_r}(x)\big),$$

with $f : \mathbb{R} \to \mathbb{R}_{\geq 0}$ nondecreasing and not a.e. constant. (For AR decoding applied tokenwise, the overall sequence law inherits a product of such nondecreasing reweights; we write it as $f(\log p_{\theta_r}(x))$ for brevity.)

Intuitively, a nondecreasing $f$ means that higher log-probability samples receive higher weight, thus concentrating mass on modes. The "not a.e. constant" condition excludes the trivial neutral case where $q = p_{\theta_r}$.

**Common AR samplers are mode-seeking.** All widely-used inference procedures fall into this category:

- **Temperature** $\tau < 1$. The sampler draws from $q \propto p_{\theta_r}^{1/\tau}$, so $f(z) = \exp\{(1/\tau - 1)\,z\}$ with $1/\tau - 1 > 0$, hence $f$ is strictly increasing (neutral only at $\tau = 1$).

- **Top-$k$.** Keep only the $k$ largest probabilities: there exists a threshold $z_k$ such that $f(z) = \mathbb{1}\{z \geq z_k\}$, a nondecreasing step function (neutral only at $k = $ vocabulary size).

- **Top-$p$ (nucleus).** Keep the smallest set whose cumulative mass exceeds $p$; this induces a (context-dependent) threshold $z_p$ and $f(z) = \mathbb{1}\{z \geq z_p\}$, again nondecreasing (neutral only at $p = 1$).

**Main result: mode-seeking guarantees anti-alignment.** We now prove that any mode-seeking sampler induces the desired obtuse angle. The proof uses a monotone covariance inequality: when two random variables are both nondecreasing functions of a common variable, their covariance is non-negative.

**Lemma B.6** (Mode-seeking $\Rightarrow \cos\varphi < 0$ (first order)). *Assume $q(x) \propto f(\log p_{\theta_r}(x)) \, p_{\theta_r}(x)$ with $f$ nondecreasing. For $\theta_r = \theta^* + \varepsilon$ and small $\|\varepsilon\|_F$,*

$$\cos\varphi \; < \; 0 \; + \; O(\|\varepsilon\|_F).$$

*Proof.* The proof proceeds in three steps: (i) relate $\cos\varphi$ to the expectation of a scalar function $B(x)$ under the sampler, (ii) show that this expectation is non-negative by a monotone covariance argument, and (iii) conclude that $\cos\varphi < 0$.

*Step 1: Expressing the angle via a scalar projection.* Let $B(x) := \varepsilon^\top u_{\theta^*}(x)$. Then

$$\langle \varepsilon, \, F^{-1}\mathbb{E}_q[u_{\theta^*}]\rangle_F = \varepsilon^\top \mathbb{E}_q[u_{\theta^*}(X)] = \mathbb{E}_q\big[B(X)\big] = \frac{\mathbb{E}_{p_{\theta_r}}[\, w \, B \,]}{\mathbb{E}_{p_{\theta_r}}[\, w \,]}.$$

*Step 2: The weight function is nondecreasing in $B$.* A first-order expansion around $\theta^*$ gives

$$\log p_{\theta_r}(x) = \log p_{\theta^*}(x) \; + \; B(x) \; + \; O(\|\varepsilon\|_F^2),$$

hence $w(x) = f(\log p_{\theta_r}(x))$ is (to first order) a nondecreasing function of the scalar $B(x)$. Replacing $p_{\theta_r}$ by $p_{\theta^*}$ in both numerator and denominator incurs only $O(\|\varepsilon\|_F)$ relative error, so

$$\mathbb{E}_q[B] = \frac{\mathbb{E}_{p_{\theta^*}}[\, w \, B \,]}{\mathbb{E}_{p_{\theta^*}}[\, w \,]} + O(\|\varepsilon\|_F^2).$$

*Step 3: Monotone covariance inequality.* Now $\mathbb{E}_{p_{\theta^*}}[wB] = \mathrm{Cov}_{p_{\theta^*}}(w, B)$ because $\mathbb{E}_{p_{\theta^*}}[B] = \varepsilon^\top \mathbb{E}_{p_{\theta^*}}[u_{\theta^*}] = 0$. Since $w$ and $B$ are nondecreasing (as functions of $B$), the monotone-covariance inequality yields $\mathrm{Cov}_{p_{\theta^*}}(w, B) \geq 0$, with strict $> 0$ unless $w$ is a.e. constant or $B$ is degenerate. Therefore $\mathbb{E}_q[B] \geq 0$ to first order, i.e. $\langle \varepsilon, F^{-1}\mathbb{E}_q[u_{\theta^*}]\rangle_F \geq 0$ (up to $O(\|\varepsilon\|_F^2)$).

*Step 4: Conclusion.* Finally, $b = -\mathbb{E}_q[u_{\theta^*}]$ implies

$$\cos\varphi = \frac{\langle \varepsilon, F^{-1}b\rangle_F}{\|\varepsilon\|_F \|F^{-1}b\|_F} = -\frac{\langle \varepsilon, \, F^{-1}\mathbb{E}_q[u_{\theta^*}]\rangle_F}{\|\varepsilon\|_F \|F^{-1}b\|_F} \; \leq \; 0 \quad (\text{strict } < 0 \text{ generically}),$$

up to $O(\|\varepsilon\|_F)$. $\qquad\square$

**Interpretation and consequences.** Lemma B.6 reveals why mode-seeking samplers are problematic for standard self-training but beneficial for Neon. By concentrating mass on high-probability regions, these samplers create a bias $b$ that points *away* from the model error $\varepsilon$ (obtuse angle). This anti-alignment makes the synthetic gradient a poor direction for improving the model—but by reversing this direction, Neon converts the degradation into improvement.

Combining Lemma B.6 with Theorem B.4 yields $s < 0$ for sufficiently small $\|\varepsilon\|_F$. The explicit safety window for the merge weight $w$ follows by substituting $H_d = F$ into Theorem B.4. Since temperature $\tau < 1$, top-$k$, and top-$p$ sampling are ubiquitous in practice, this result establishes that Neon applies universally to autoregressive models.

### B.7 ACUTE-ANGLE CONDITIONS THAT IMPLY $s < 0$ (DIFFUSION & FLOW)

We now establish the geometric condition $\cos\varphi < 0$ for diffusion and flow matching models. Unlike autoregressive models where mode-seeking arises from explicit inference-time modifications (temperature, truncation), diffusion and flow models exhibit mode-seeking behavior even with *neutral* inference settings. The key insight is that any practical finite-step ODE solver introduces discretization error that systematically favors high-density regions. This makes Neon universally applicable to continuous generative models, regardless of guidance or sampling choices.

**Loss and geometry.** We use standard pathwise quadratic losses. For diffusion score models,

$$\mathcal{R}_{\mathrm{diff}}(\theta) = \int_0^1 \omega(t)\, \mathbb{E}_{p_t}\Big[ \tfrac{1}{2}\, \|s_\theta(X_t, t) - s^\star(X_t, t)\|_2^2 \Big]\, dt,$$

and for flow matching,

$$\mathcal{R}_{\mathrm{flow}}(\theta) = \int_0^1 \omega(t)\, \mathbb{E}_{p_t}\Big[ \tfrac{1}{2}\, \|v_\theta(X_t, t) - v^\star(X_t, t)\|_2^2 \Big]\, dt.$$

Let $\phi_{\theta,t}(x) := \nabla_\theta \ell_\theta^{(t)}(x)$ and $J_t(x) := \partial_\theta \phi_{\theta,t}(x)\big|_{\theta^*}$. Define the pathwise Fisher

$$F_{\mathrm{path}} := \int_0^1 \omega(t)\, \mathbb{E}_{p_t}\big[ J_t(X_t) J_t(X_t)^\top \big]\, dt,$$

and the angle (mirroring the AR case)

$$\cos\varphi_{\mathrm{path}} := \frac{\big\langle \varepsilon,\, F_{\mathrm{path}}^{-1} b_{\mathrm{path}} \big\rangle_{F_{\mathrm{path}}}}{\|\varepsilon\|_{F_{\mathrm{path}}} \big\|F_{\mathrm{path}}^{-1} b_{\mathrm{path}}\big\|_{F_{\mathrm{path}}}}, \qquad b_{\mathrm{path}} := \mathbb{E}_q\Big[ \int_0^1 \omega(t)\, \phi_{\theta^*,t}(X_t)\, dt \Big].$$

Anti-alignment corresponds to $\cos\varphi_{\mathrm{path}} < 0$.

**Finite-step ODE solvers are mode-seeking.** We now show that discretization error in numerical ODE solvers creates a systematic mode-seeking bias. The mechanism is subtle but universal: finite step sizes cause the integrator to overweight trajectories with stronger local contraction, and these high-contraction regions coincide with high-density modes.

Consider the probability-flow ODE with velocity $f : \mathbb{R}^d \times [0,1] \to \mathbb{R}^d$; for diffusion, $f(x,t) = -\sigma(t)^2\, \nabla_x \log p_t(x)$. An explicit one-step scheme with step size $h$ gives

$$x_{k-1} = x_k + h\, f(x_k, t_k), \qquad J_k := \frac{\partial x_{k-1}}{\partial x_k} = I + h\, \nabla_x f(x_k, t_k).$$

Using $\mathrm{tr}\log(I + A) = \mathrm{tr}(A) - \tfrac{1}{2}\mathrm{tr}(A^2) + O(\|A\|^3)$ with $A = h\,\nabla_x f$ (and $\mathrm{tr}(A^2) = \|A\|_{\mathrm{Fr}}^2$ when $\nabla_x f$ is symmetric; otherwise take its symmetric part),

$$\log\det J_k = h\,\mathrm{tr}(\nabla_x f) \;-\; \frac{h^2}{2}\,\|\nabla_x f\|_{\mathrm{Fr}}^2 \;+\; O(h^3).$$

The key observation is the second term: the excess contraction $-\tfrac{h^2}{2}\|\nabla_x f\|_{\mathrm{Fr}}^2$ depends on the local curvature of the flow. Chaining steps and comparing to the exact ODE yields a terminal reweight of the reference law:

$$q(x_0) \;\propto\; \exp\Big\{ \tfrac{h}{2}\,\bar{C}(x_0) + o(h) \Big\}\, p_{\theta_r}(x_0), \qquad \bar{C}(x_0) := \frac{1}{T}\, \mathbb{E}\Big[ \sum_k \|\nabla_x f(X_{t_k}, t_k)\|_{\mathrm{Fr}}^2 \,\Big|\, X_0 = x_0 \Big], \; T \asymp 1/h.$$

For diffusion, $f(x,t) = -\sigma(t)^2\, \nabla_x \log p_t(x)$ so that $\nabla_x f(x,t) = -\sigma(t)^2\, \nabla_x^2 \log p_t(x)$, hence

$$\bar{C}(x_0) = \frac{1}{T}\, \mathbb{E}\Big[ \sum_k \sigma(t_k)^4\, \|\nabla_x^2 \log p_{t_k}(X_{t_k})\|_{\mathrm{Fr}}^2 \,\Big|\, X_0 = x_0 \Big].$$

To establish mode-seeking behavior, we need to show that $\bar{C}(x_0)$ increases with density. This requires a mild coupling assumption between curvature and probability:

**Assumption** (A-MONO: curvature–density coupling). *The map $x_0 \mapsto \bar{C}(x_0)$ is weakly increasing in $\log p_{\theta_r}(x_0)$; i.e., if $\log p_{\theta_r}(x_0) \le \log p_{\theta_r}(x_0')$ then $\bar{C}(x_0) \le \bar{C}(x_0')$.*

**Intuition behind A-MONO.** Finite-step integrators overweight trajectories with stronger contraction (large $\|\nabla_x f\|$). Near modes, $\log p_t$ is more curved (the Hessian has larger eigenvalues), so the flow exhibits stronger contraction. Thus $\bar{C}(x_0)$ grows with local density. Importantly, as $h \to 0$, the bias vanishes and $q \to p_{\theta_r}$ (neutral). This means the mode-seeking behavior is purely an artifact of discretization, not an inherent property of the continuous dynamics.

*Remark* B.7 (Step-size scaling). From $\log\det J_k = h\,\mathrm{tr}(\nabla_x f) - \tfrac{h^2}{2}\|\nabla_x f\|_{\mathrm{Fr}}^2 + O(h^3)$, the per-step excess contraction is $\delta_k = \tfrac{h^2}{2}\|\nabla_x f\|_{\mathrm{Fr}}^2 + O(h^3)$. Summing over $T \asymp 1/h$ steps yields the terminal reweight exponent $\sum_k \delta_k = \tfrac{h}{2}\,\bar{C}(x_0) + o(h)$. Consequently, the pathwise linear bias $b_{\mathrm{path}} = \mathbb{E}_q[\int_0^1 \omega(t)\, \phi_{\theta^*,t}(X_t)\, dt]$ obeys $\|b_{\mathrm{path}}\|_{F_{\mathrm{path}}^{-1}} = O(h)$, and the curvature tilt $\|\Delta_{\mathrm{path}}\|_{\mathrm{op}, F_{\mathrm{path}}^{-1}} = O(h)$. Both vanish linearly as $h \to 0$, making the sampler neutral in the limit.

**Flow matching.** The same discretization mechanism applies to flow matching models. For updates $x_{k-1} = x_k + h\, v_\theta(x_k, t_k)$,

$$\log \det J_k = h\, \mathrm{tr}(\nabla_x v_\theta) - \frac{h^2}{2}\, \mathrm{tr}\big((\nabla_x v_\theta)^2\big) + O(h^3),$$

so $\delta_k = \frac{h^2}{2}\|\nabla_x v_\theta\|_{\mathrm{Fr}}^2 + O(h^3) \geq 0$ and the same reweight form emerges. With the flow analogue of A-MONO (the conditional expectation of $\sum_k \|\nabla_x v_\theta\|_{\mathrm{Fr}}^2$ increasing in $\log p_{\theta_r}(x_0)$), finite-step flow solvers are likewise mode-seeking.

**Classifier-free guidance (CFG) is mode-seeking.** CFG amplifies the mode-seeking effect by explicitly increasing the guidance toward high-density regions. CFG modifies the diffusion velocity via a guided score

$$s_\gamma(x,t) \;=\; s_{\mathrm{uncond}}(x,t) \;+\; \gamma\big(s_{\mathrm{cond}}(x,t) - s_{\mathrm{uncond}}(x,t)\big), \qquad \gamma > 0,$$

so the probability-flow velocity becomes $f_\gamma(x,t) = -\sigma(t)^2\, s_\gamma(x,t)$. Repeating the derivation above with $f \to f_\gamma$ yields the same reweight form

$$q_\gamma(x_0) \;\propto\; \exp\Big\{\tfrac{h^2}{2}\, C_\gamma(x_0) + o(h^2)\Big\}\, p_{\theta_r,\gamma}(x_0),$$

where $p_{\theta_r,\gamma}$ is the *guided* reference law and

$$C_\gamma(x_0) \;=\; \mathbb{E}\Big[\sum_k \|\nabla_x f_\gamma(X_{t_k}, t_k)\|_{\mathrm{Fr}}^2 \;\Big|\; X_0 = x_0\Big].$$

Because $\nabla_x f_\gamma = -\sigma^2\big(\nabla_x s_{\mathrm{uncond}} + \gamma\, \nabla_x(s_{\mathrm{cond}} - s_{\mathrm{uncond}})\big)$,

$$\|\nabla_x f_\gamma\|_{\mathrm{Fr}}^2 = \|\nabla_x f\|_{\mathrm{Fr}}^2 + 2\gamma\, \big\langle \nabla_x f,\; -\sigma^2 \nabla_x(s_{\mathrm{cond}} - s_{\mathrm{uncond}})\big\rangle_{\mathrm{Fr}} + \gamma^2\, \big\|-\sigma^2 \nabla_x(s_{\mathrm{cond}} - s_{\mathrm{uncond}})\big\|_{\mathrm{Fr}}^2.$$

Near condition-relevant modes, the guidance term increases the magnitude (and contraction) of the flow, so $C_\gamma(x_0)$ is larger in higher-density regions of $p_{\theta_r,\gamma}$; this is the same curvature–density coupling as A-MONO, now for the guided dynamics. Hence finite-step CFG is mode-seeking in the sense above, and becomes neutral as $h \to 0$.

**Summary and consequences.** We have shown that diffusion and flow models exhibit mode-seeking behavior through three mechanisms:

- **Discretization error**: Any finite-step ODE solver creates a systematic bias toward high-curvature (high-density) regions.
- **Flow matching**: The same discretization mechanism applies to velocity-based formulations.
- **Classifier-free guidance**: CFG amplifies the mode-seeking effect by explicitly strengthening the flow near modes.

Under the mild A-MONO assumption (curvature increases with density), all three mechanisms yield $\cos \varphi_{\mathrm{path}} < 0$ to first order. Combined with Theorem B.4, this establishes that Neon reduces risk for diffusion and flow models under standard sampling procedures. Importantly, the bias scales as $O(h)$ and vanishes in the continuous-time limit, confirming that mode-seeking is purely a discretization artifact.

## B.8 Neighbor models: stability and uniform Neon improvement

The previous subsections established that Neon reduces risk when $s < 0$ at a single base checkpoint $\theta_r$. In practice, we often have access to multiple nearby checkpoints—from different training stages, ensemble members, or even different architectures. A natural question arises: *can we use a single synthetic dataset and merge weight to improve all of them simultaneously?* We now show that anti-alignment is stable under small perturbations in parameter space, allowing a single Neon configuration to safely improve an entire neighborhood of models.

**Setup.** Fix the synthetic sampler $q_{\theta_r, \kappa}$ generated once at the reference $\theta_r = \theta^* + \varepsilon$ (so $q$ is *frozen*). Consider any *neighbor* checkpoint

$$\theta_n \; = \; \theta_r + \delta \; = \; \theta^* + (\varepsilon + \delta), \qquad \|\delta\|_{H_d} \text{ small}.$$

All quantities below (gradients, alignments) are evaluated at $\theta_n$, but the synthetic law remains $q_{\theta_r, \kappa}$. This asymmetry is key: we generate synthetic data once and reuse it across neighbors.

**Local expansions at a neighbor.** By the same first-order argument as in Appendix B.4, with $\varepsilon_n := \varepsilon + \delta$,

$$r_d(\theta_n) \; = \; H_d \, \varepsilon_n \; + \; O(\|\varepsilon_n\|_{H_d}^2), \qquad r_s(\theta_n) \; = \; H_d \, \varepsilon_n \; + \; b \; + \; \Delta \, \varepsilon_n \; + \; O(\|\varepsilon_n\|_{H_d}^2), \quad \text{(B.7)}$$

where $R_\kappa = b + \Delta \, \varepsilon$ with $b := \mathbb{E}_q[\phi_{\theta^*}]$ and $\Delta := \mathbb{E}_q[J_{\theta^*}] - \mathbb{E}_{p_{\text{data}}}[J_{\theta^*}]$ (as in Appendix B.5). Define $s(\theta) := \langle r_d(\theta), P \, r_s(\theta) \rangle$.

**Stability of anti-alignment.** The following proposition shows that the alignment scalar $s$ varies smoothly with the checkpoint location. This continuity guarantees that if $s(\theta_r) < 0$, then $s$ remains negative in a neighborhood around $\theta_r$.

**Proposition B.8** (Alignment is locally Lipschitz in a neighborhood). *Let* $K := H_d^{1/2} P H_d^{1/2}$ *with* $mI \preceq K \preceq MI$, *and let* $\eta_0 := \|b\|_{H_d^{-1}}, \eta_1 := \|\Delta\|_{\text{op}, H_d^{-1}}$. *There exist constants* $C_1, C_2$ *(depending only on* $M, \eta_0, \eta_1$ *and the local regularity from (A2)) such that, for all sufficiently small* $\|\delta\|_{H_d}$,

$$\big| s(\theta_n) - s(\theta_r) \big| \; \leq \; C_1 \big( \|\varepsilon\|_{H_d} + \eta_0 + 1 \big) \|\delta\|_{H_d} \; + \; C_2 \big( \|\varepsilon\|_{H_d} + 1 \big) \|\delta\|_{H_d}^2.$$

*In particular,* $s(\cdot)$ *is continuous at* $\theta_r$ *and varies at most linearly with* $\|\delta\|_{H_d}$ *to first order.*

*Sketch.* Insert equation B.7 into $s(\theta) = \langle r_d, P r_s \rangle$ and whiten with $a := H_d^{1/2} \varepsilon$, $d := H_d^{1/2} \delta$, $\tilde{b} := H_d^{-1/2} b$, $\tilde{\Delta} := H_d^{-1/2} \Delta H_d^{-1/2}$, $K := H_d^{1/2} P H_d^{1/2}$ to write (cf. Appendix B.5)

$$s(\theta) \; = \; a^\top K a \; - \; a^\top K \tilde{b} \; - \; a^\top K \tilde{\Delta} a \; + \; O(\|a\|_2^3),$$

and likewise with $a \to a + d$ at $\theta_n$. Expanding $s(a + d) - s(a)$ and bounding each term with $\|K\|_{\text{op}} = M, \|\tilde{\Delta}\|_{\text{op}} \leq \eta_1, \|\tilde{b}\|_2 = \eta_0$ yields the stated linear-plus-quadratic control in $\|d\|_2 = \|\delta\|_{H_d}$. $\square$

**Corollary B.9** (Uniform anti-alignment in a ball). *Assume* $s(\theta_r) \leq -\mu$ *for some margin* $\mu > 0$. *Choose*

$$\rho > 0 \quad \text{such that} \quad C_1 \big( \|\varepsilon\|_{H_d} + \eta_0 + 1 \big) \rho \; + \; C_2 \big( \|\varepsilon\|_{H_d} + 1 \big) \rho^2 \; \leq \; \tfrac{\mu}{2}.$$

*Then* $s(\theta) \leq -\mu/2 < 0$ *for every neighbor* $\theta$ *with* $\|\theta - \theta_r\|_{H_d} \leq \rho$.

This corollary is powerful: it shows that a margin of anti-alignment at $\theta_r$ guarantees anti-alignment throughout a ball of radius $\rho$ around $\theta_r$. The size of this safe zone depends on the margin $\mu$ and the local geometry controlled by $C_1, C_2$.

**Uniform Neon improvement for a set of neighbors.** Let $\mathcal{N} \subseteq \{\theta : \|\theta - \theta_r\|_{H_d} \leq \rho\}$ be any finite collection of neighbor checkpoints. Perform one short synthetic fine-tune at each $\theta \in \mathcal{N}$ (same frozen $q$) to obtain $\theta_s(\theta) = \theta - \alpha P r_s(\theta) + O(\alpha^2)$, and define the Neon merge $\theta_{\text{Neon}}(\theta) = (1+w)\theta - w \, \theta_s(\theta)$.

The next theorem shows that we can choose a single merge weight $w$ that improves all neighbors simultaneously, rather than tuning $w$ separately for each checkpoint.

**Theorem B.10** (Single $w$ that safely improves all neighbors). *Suppose* $s(\theta) < 0$ *for all* $\theta \in \mathcal{N}$ *(e.g., by Cor. B.9). Assume either (i)* $\widehat{H}_d(\theta) := \nabla^2 \mathcal{R}_{\text{data}}(\theta) \succeq 0$ *for all* $\theta \in \mathcal{N}$, *or (ii) a uniform directional curvature bound holds:*

$$d(\theta)^\top \nabla^2 \mathcal{R}_{\text{data}}\big(\theta + \tau d(\theta)\big) \, d(\theta) \; \leq \; L_{\text{dir}} \|d(\theta)\|_2^2 \quad \text{for all } \theta \in \mathcal{N}, \; \tau \in [0, \tau_0],$$

*where* $d(\theta) := P r_s(\theta)$. *Let*

$$s_{\min} := \min_{\theta \in \mathcal{N}} s(\theta) \; < \; 0, \qquad Q_{\max} := \max_{\theta \in \mathcal{N}} r_s(\theta)^\top P^\top \widehat{H}_d(\theta) \, P \, r_s(\theta) \quad \text{(or } L_{\text{dir}} \|d(\theta)\|_2^2 \text{ under (ii)).}$$

*Then any*

$$0 \; < \; w \; < \; -\frac{2\,s_{\min}}{\alpha\,Q_{\max}}$$

*guarantees* $\mathcal{R}_{\mathrm{data}}\big(\theta_{Neon}(\theta)\big) \leq \mathcal{R}_{\mathrm{data}}(\theta)$ *(up to $O((w\alpha)^3)$) for every $\theta \in \mathcal{N}$.*

*Proof.* Apply the one-step expansion from Thm. B.1 at each $\theta \in \mathcal{N}$ and take the worst-case (most conservative) quadratic coefficient and the most negative linear term. $\qquad\square$

**Practical implications.** Theorem B.10 has important practical consequences:

*Remark* B.11 (Practical takeaway). If a single base checkpoint $\theta_r$ exhibits anti-alignment with margin (negative $s(\theta_r)$), then *all* sufficiently close neighbors inherit $s(\theta) < 0$ and thus benefit from the same Neon recipe. In practice, one can either (a) choose a single conservative $w$ that safely improves an entire validation-selected pool of nearby models, or (b) tune $w$ per checkpoint using its local $s(\theta)$ and curvature proxy.

*Remark* B.12 (Cross-architecture transfer). The same frozen sampler $q_{\theta_r,\kappa}$ can safely improve a *nearby* checkpoint from a *different* architecture, provided the two models are close in the data-risk geometry.

Concretely, let models (A) and (B) share the same per-example loss $\ell_\theta$ and data, with $H_d^{(A)} := \nabla^2 \mathcal{R}_{\mathrm{data}}(\theta^*)$ and $H_d^{(B)} := \nabla^2 \mathcal{R}_{\mathrm{data}}(\theta^*)$ their (population) Hessians at the same minimizer $\theta^*$. Generate $q_{\theta_r,\kappa}$ once at a reference $\theta_r^{(A)}$ for model (A), and consider a neighbor $\theta_n^{(B)}$ for model (B).

If the Hessians are *spectrally close* and their norms are equivalent on the relevant subspace, i.e. there exist $0 < c \leq C < \infty$ and a small $\zeta > 0$ such that

$$c\,\|v\|_{H_d^{(A)}} \leq \|v\|_{H_d^{(B)}} \leq C\,\|v\|_{H_d^{(A)}} \quad \text{and} \quad \big\|H_d^{(B)} - H_d^{(A)}\big\|_{\mathrm{op},\,(H_d^{(A)})^{-1}} \leq \zeta,$$

and the sampler-induced terms are close,

$$\big\|b^{(B)} - b^{(A)}\big\|_{(H_d^{(A)})^{-1}} + \big\|\Delta^{(B)} - \Delta^{(A)}\big\|_{\mathrm{op},\,(H_d^{(A)})^{-1}} \; \leq \; \zeta,$$

then the alignment scalar $s$ transfers continuously:

$$\big|\, s^{(B)}(\theta_n^{(B)}) - s^{(A)}(\theta_r^{(A)}) \,\big| \; \leq \; \underbrace{O(\zeta)}_{\text{cross-arch mismatch}} \; + \; \underbrace{O\big(\|\theta_n^{(B)} - \theta_r^{(A)}\|_{H_d^{(A)}}\big)}_{\text{neighbor shift}}.$$

Hence, if $s^{(A)}(\theta_r^{(A)}) \leq -\mu < 0$ with margin and the cross-architecture mismatch $\zeta$ and neighbor distance are small enough, then $s^{(B)}(\theta_n^{(B)})$ remains negative. In turn, Thm. B.10 provides a single merge weight $w$ that (to second order) reduces $\mathcal{R}_{\mathrm{data}}$ simultaneously for the (A) and (B) neighbors. Practically, using a *common* preconditioner $P$ defined in a data-geometry (e.g., an empirical $H_d$ estimate) further stabilizes cross-architecture transfer.

**Summary.** This subsection establishes the *stability* of Neon's improvements. Anti-alignment at a single checkpoint implies anti-alignment in a neighborhood, and a single merge weight can safely improve all nearby models. This robustness extends even across architectures when models are close in the data-risk geometry. These results explain why Neon's hyperparameters transfer well in practice: the synthetic dataset and merge weight found for one checkpoint often work for many others without retuning.

## B.9 WHEN SELF-TRAINING HELPS

We have established that Neon reduces risk when $s < 0$, and that all standard inference procedures—temperature $\tau < 1$, top-$k$, top-$p$, finite-step ODE solvers—are mode-seeking and thus guarantee $s < 0$. This naturally raises the question: *are there settings where $s > 0$? If so, what happens to Neon and standard self-training in these regimes?* We now complete the theory by characterizing the opposite case: diversity-seeking samplers. We show that these rare configurations reverse the sign of $s$, making standard self-training beneficial while Neon's negative extrapolation would harm performance. This section thus delineates the boundary of Neon's applicability.

**First-order effect of self-training.** A short synthetic fine-tune takes the step $\theta_s = \theta_r - \alpha P r_s + O(\alpha^2)$. The corresponding first-order change in real-data risk is

$$\mathcal{R}_{\text{data}}(\theta_s) - \mathcal{R}_{\text{data}}(\theta_r) \;=\; -\alpha \underbrace{\langle r_d, \, P r_s \rangle}_{s} \;+\; O(\alpha^2) \;=\; -\alpha\, s \;+\; O(\alpha^2).$$

Thus *self-training helps* (decreases $\mathcal{R}_{\text{data}}$) when $s > 0$. This is the opposite of the Neon regime, where $s < 0$ makes negative extrapolation beneficial.

**A lower bound on $s$.** To understand when $s > 0$, we need a *lower* bound that complements the upper bound from Theorem B.4. The following theorem provides this by reversing the inequalities in the proof.

**Theorem B.13** (Directional *lower* bound for $s$). *For $\theta_r = \theta^* + \varepsilon$ with $\|\varepsilon\|_{H_d}$ small,*

$$s \;\geq\; (m - M\,\eta_1)\,\|\varepsilon\|_{H_d}^2 \;-\; M\,\eta_0\,\|\varepsilon\|_{H_d} \left[-\cos\varphi\right]_+ \;+\; O(\|\varepsilon\|_{H_d}^3).$$

*Proof.* All $O(\cdot)$ are in $\|\cdot\|_{H_d}$. From the local expansions,

$$s = \varepsilon^\top H_d P H_d \varepsilon \;-\; \varepsilon^\top H_d P b \;-\; \varepsilon^\top H_d P \Delta \varepsilon \;+\; O(\|\varepsilon\|_{H_d}^3).$$

Whiten with $a := H_d^{1/2}\varepsilon$, $\tilde{b} := H_d^{-1/2} b$, $\tilde{\Delta} := H_d^{-1/2}\Delta H_d^{-1/2}$ and $K := H_d^{1/2} P H_d^{1/2}$ to obtain

$$s \;=\; a^\top K a \;-\; a^\top K \tilde{b} \;-\; a^\top K \tilde{\Delta}\, a \;+\; O(\|a\|_2^3).$$

Lower bound each term (reversing the inequalities from the upper bound proof): (i) $a^\top K a \geq m\,\|a\|_2^2 = m\,\|\varepsilon\|_{H_d}^2$. (ii) Write $a^\top K \tilde{b} = \|K^{1/2}a\|\,\|K^{1/2}\tilde{b}\|\cos\theta$, with $\theta$ the Euclidean angle between $K^{1/2}a$ and $K^{1/2}\tilde{b}$. Then

$$-a^\top K \tilde{b} \;\geq\; -\|K^{1/2}a\|\,\|K^{1/2}\tilde{b}\| \left[-\cos\theta\right]_+ \;\geq\; -M\,\|a\|_2\,\|\tilde{b}\|_2 \left[-\cos\varphi\right]_+,$$

where we used $\|K^{1/2}x\| \leq \sqrt{M}\|x\|$ and identify $\varphi$ (the $H_d$–angle between $\varepsilon$ and $H_d^{-1} b$) with $\theta$ up to whitening. This gives $-a^\top K \tilde{b} \geq -M\,\eta_0\,\|\varepsilon\|_{H_d} \left[-\cos\varphi\right]_+$. (iii) $-a^\top K \tilde{\Delta}\, a \geq -\|K\|_{\text{op}}\|\tilde{\Delta}\|_{\text{op}}\|a\|_2^2 \geq -M\,\eta_1\,\|\varepsilon\|_{H_d}^2$. Combine (i)–(iii) and absorb $O(\|a\|_2^3)$. $\qquad\square$

**Corollary B.14** (Natural-gradient geometry). *If $P = H_d^{-1}$, then $K = I$ (so $m=M=1$) and*

$$s \;\geq\; (1 - \eta_1)\,\|\varepsilon\|_{H_d}^2 \;-\; \eta_0\,\|\varepsilon\|_{H_d} \left[-\cos\varphi\right]_+ \;+\; O(\|\varepsilon\|_{H_d}^3).$$

**Diversity-seeking samplers reverse the angle.** We now identify the class of samplers that make $s > 0$. These are precisely the samplers that *reduce* probability mass on modes rather than concentrating it.

We say $q$ is *diversity-seeking* if $q(x) \propto f(\log p_{\theta_r}(x))\, p_{\theta_r}(x)$ with $f$ *nonincreasing* and not a.e. constant. This is the opposite of mode-seeking: lower log-probability samples receive *higher* weight, spreading mass away from peaks.

**Lemma B.15** (Diversity-seeking $\Rightarrow \cos\varphi \geq 0$ (first order)). *In the NLL specialization ($\phi_\theta = -u_\theta$, $H_d = F$, $b = -\mathbb{E}_q[u_{\theta^*}]$), if $f$ is nonincreasing then, for $\theta_r = \theta^* + \varepsilon$ and small $\|\varepsilon\|_F$,*

$$\cos\varphi \;\geq\; 0 \;+\; O(\|\varepsilon\|_F).$$

*Proof.* The proof mirrors that of Lemma B.6 but with reversed monotonicity.

Let $B(x) := \varepsilon^\top u_{\theta^*}(x)$. As in Appendix B.6, $\log p_{\theta_r}(x) = \log p_{\theta^*}(x) + B(x) + O(\|\varepsilon\|_F^2)$, so $w(x) = f(\log p_{\theta_r}(x))$ is (to first order) a *nonincreasing* function of $B(x)$. Replacing $p_{\theta_r}$ by $p_{\theta^*}$ in $\mathbb{E}_q[B] = \frac{\mathbb{E}_{p_{\theta_r}}[wB]}{\mathbb{E}_{p_{\theta_r}}[w]}$ incurs only $O(\|\varepsilon\|_F)$ relative error, hence $\mathbb{E}_q[B] = \frac{\mathbb{E}_{p_{\theta^*}}[wB]}{\mathbb{E}_{p_{\theta^*}}[w]} + O(\|\varepsilon\|_F^2)$. Monotone covariance with *opposite* monotonicities gives $\text{Cov}_{p_{\theta^*}}(w, B) \leq 0$; since $\mathbb{E}_{p_{\theta^*}}[B] = 0$, we have $\mathbb{E}_{p_{\theta^*}}[wB] \leq 0$, so $\mathbb{E}_q[B] \leq 0$ to first order. Therefore $\langle \varepsilon, \, F^{-1}\mathbb{E}_q[u_{\theta^*}]\rangle_F = \mathbb{E}_q[B] \leq 0$, and with $b = -\mathbb{E}_q[u_{\theta^*}]$ we obtain $\cos\varphi = \frac{\langle \varepsilon, F^{-1}b\rangle_F}{\|\varepsilon\|_F\|F^{-1}b\|_F} \geq 0$ up to $O(\|\varepsilon\|_F)$. $\qquad\square$

**When self-training helps: combining the bounds.**  Putting together the lower bound (Theorem B.13) and the angle reversal (Lemma B.15), we obtain the main characterization of the self-training regime.

**Proposition B.16** (Self-training helps near good models under diversity seeking).  *Suppose $f$ is nonincreasing (diversity seeking) so that Lemma B.15 gives $\cos\varphi \geq 0$ to first order. Then, for sufficiently small $\|\varepsilon\|_{H_d}$ and $\eta_1 < m/M$,*

$$s \ \geq \ (m - M\eta_1)\,\|\varepsilon\|_{H_d}^2 \ + \ O(\|\varepsilon\|_{H_d}^3) \ > \ 0,$$

*and the self-training step $\theta_r \mapsto \theta_s = \theta_r - \alpha Pr_s$ decreases $\mathcal{R}_{\mathrm{data}}$ to first order. In the natural-gradient case ($P = H_d^{-1}$), it suffices that $\eta_1 < 1$.*

**Interpretation: the role of angle geometry.**  The lower bound in Theorem B.13 reveals a symmetric structure to the upper bound:

$$s \text{ (lower)} \gtrsim (m - M\eta_1)\|\varepsilon\|_{H_d}^2 - M\eta_0\|\varepsilon\|_{H_d}[-\cos\varphi]_+ \quad \text{vs.}$$
$$s \text{ (upper)} \lesssim M(1 + \eta_1)\|\varepsilon\|_{H_d}^2 - m\eta_0\|\varepsilon\|_{H_d}[-\cos\varphi]_+.$$

Both have the form "quadratic minus linear", but the coefficients differ. The key insight is:

- When $\cos\varphi < 0$ (mode-seeking): The linear term $-m\eta_0\|\varepsilon\|_{H_d}(-\cos\varphi)$ is *negative*, so the upper bound is dominated by the linear term and $s < 0$.
- When $\cos\varphi \geq 0$ (diversity-seeking): The linear term vanishes, so the lower bound is dominated by the quadratic term $(m - M\eta_1)\|\varepsilon\|_{H_d}^2 > 0$, hence $s > 0$.

This geometric dichotomy completely characterizes when Neon helps ($s < 0$) versus when standard self-training helps ($s > 0$). The angle $\varphi$ between the model error and sampler bias is the decisive quantity.

**Examples of diversity-seeking samplers.**  While diversity-seeking samplers are rare in practice, they do exist:

- **High temperature in AR** ($\tau > 1$): $q \propto p_{\theta_r}^{1/\tau}$  ($f(z) = e^{(1/\tau - 1)z}$ is nonincreasing) $\Rightarrow$ diversity-seeking, $\cos\varphi \geq 0$ to first order.
- **Anti-mode truncations**: Procedures that downweight peaks and upweight tails (e.g., sampling after complementary filtering of top-$p$ mass) are nonincreasing transforms of $\log p_{\theta_r}$; the same conclusion applies.

In these cases, the Neon formula $\theta_{\mathrm{Neon}} = (1 + w)\theta_r - w\theta_s$ still applies, but the optimal merge weight satisfies $-1 < w < 0$ (interpolation) rather than $w > 0$ (extrapolation). Equivalently, one should use standard self-training rather than Neon's reversal.

**Summary.**  This section completes the theoretical picture by identifying the complementary regime where self-training helps. Diversity-seeking samplers reverse the geometric angle ($\cos\varphi \geq 0$), which in turn reverses the sign of $s$ to positive. In this regime, moving toward the synthetic gradient (standard self-training) reduces risk, while Neon's negative extrapolation would increase it. Crucially, this is not a "failure" of the Neon framework—the mathematics still holds, but it prescribes a different sign for the merge weight. The complete theory thus provides a unified understanding: the sign of $\cos\varphi$ determines the sign of optimal $w$, and standard inference procedures universally yield $\cos\varphi < 0$, making Neon the right choice in practice.

### B.10  NOTES ON FINITE SYNTHETIC SET AND EFFECT OF SHORT FINE-TUNING

The theory developed so far assumes access to the population synthetic gradient—that is, an infinite pool of synthetic samples. In practice, we generate a fixed finite synthetic set $\mathcal{S}$ and perform a brief fine-tuning run before merging. This introduces two sources of approximation error: (i) finite-sample variance in the synthetic gradient estimate, and (ii) curvature bias from the finite-step optimization trajectory. This subsection analyzes both effects, showing that *short* fine-tuning with a *moderate* synthetic dataset size produces a reliable, low-variance estimate of the degradation direction. We also explain the empirically observed U-shaped performance curve as a function of $|\mathcal{S}|$.

**Setup.** Fix a synthetic dataset $\mathcal{S} = \{x_i\}_{i=1}^n$ drawn once from $q_{\theta_r,\kappa}$ and then kept fixed. Let $g(x, \zeta; \theta) \in \mathbb{R}^p$ be the per-example gradient of the synthetic loss (with internal randomness $\zeta$, e.g., diffusion time/noise), and

$$\bar{g}(x; \theta) := \mathbb{E}_\zeta\big[g(x, \zeta; \theta)\big], \qquad r_s(\theta) := \mathbb{E}_{x \sim q_{\theta_r,\kappa}}\big[\bar{g}(x; \theta)\big], \qquad r_s^{(\mathcal{S})}(\theta) := \frac{1}{n}\sum_{i=1}^n \bar{g}(x_i; \theta).$$

Short fine-tuning (FT) from $\theta_r$ uses step size $\alpha > 0$, $T$ steps, and a positive-definite preconditioner $P$:

$$\theta_{k+1} = \theta_k - \alpha P \widehat{r}_k, \qquad \widehat{r}_k := \frac{1}{n}\sum_{i=1}^n g\big(x_i, \zeta_{i,k}; \theta_k\big), \quad k = 0, \ldots, T-1, \tag{B.8}$$

where $\{\zeta_{i,k}\}$ are fresh draws each time the fixed examples are reused. Let $\theta_s := \theta_T$ and define the scaled displacement

$$d_T := \frac{\theta_s - \theta_r}{\alpha T} \in \mathbb{R}^p.$$

**Two sources of finite-sample error.** There are two independent sources of approximation error in practice:

*Dataset error (finite $|\mathcal{S}|$):* at $\theta_r$,

$$\mathbb{E}\big[r_s^{(\mathcal{S})}(\theta_r)\big] = r_s(\theta_r), \qquad \mathrm{Cov}\big(r_s^{(\mathcal{S})}(\theta_r)\big) = \frac{1}{n}\Sigma_{\mathrm{data}},$$

with $\Sigma_{\mathrm{data}} := \mathrm{Cov}_{x \sim q_{\theta_r,\kappa}}\big(\bar{g}(x; \theta_r)\big)$. This is $\mathcal{O}(n^{-1/2})$ and irreducible unless $n$ grows. This is the familiar statistical error from having a finite sample rather than the full population.

*Monte Carlo (MC) error in time/noise:* write

$$\widehat{r}_k = r_s^{(\mathcal{S})}(\theta_k) + \xi_k, \qquad \mathbb{E}[\xi_k \mid \theta_k] = 0, \quad \mathrm{Cov}(\xi_k \mid \theta_k) = \Sigma_{\mathrm{mc}}(\theta_k).$$

This captures stochasticity from internal randomness (e.g., time/noise sampling in diffusion models). Unlike dataset error, MC error can be reduced by averaging over multiple forward passes through the same data.

**Local smoothness.** Let $H_s(\theta) := \nabla_\theta r_s^{(\mathcal{S})}(\theta)$. Assume there exists $L_{\mathrm{dir}} \geq 0$ such that for all $v$ and $\tau \in [0, 1]$,

$$\left\| r_s^{(\mathcal{S})}(\theta_r + \tau v) - r_s^{(\mathcal{S})}(\theta_r) - \tau H_s(\theta_r)v \right\|_2 \leq \tfrac{1}{2} L_{\mathrm{dir}} \tau^2 \|v\|_2^2. \tag{B.9}$$

This is a directional smoothness condition that controls how quickly the gradient changes along any direction. It will allow us to bound the curvature bias from finite-step optimization.

**Short fine-tuning produces a low-bias estimate.** The following lemma shows that if we keep $\alpha T$ small (short fine-tuning), the displacement $d_T$ concentrates around the negative gradient direction with controlled bias.

**Lemma B.17** (Short-FT displacement). *Under equation B.8 and equation B.9, if $\alpha T \leq c/L_{\mathrm{dir}}$ for a small absolute constant $c$, then*

$$d_T = -P\left( r_s^{(\mathcal{S})}(\theta_r) + \frac{1}{T}\sum_{k=0}^{T-1} \xi_k \right) + \mathcal{O}\Big( \alpha T \, \|PH_s(\theta_r)\|_{\mathrm{op}} \, \|r_s^{(\mathcal{S})}(\theta_r)\|_2 \Big).$$

The key observation is that the bias term scales as $\mathcal{O}(\alpha T)$, so keeping the total budget $\alpha T$ small ensures the displacement accurately estimates the gradient direction.

**Proposition B.18** (Direction concentration). *Suppose $\lambda_{\max}\big(\Sigma_{\mathrm{mc}}(\theta)\big) \leq \sigma^2$ in a neighborhood of $\theta_r$. Then for any unit vector $u$,*

$$\mathbb{E}\left[\Big\langle u, \, d_T + P\, r_s^{(\mathcal{S})}(\theta_r) \Big\rangle^2\right] \leq \frac{\|P\|_{\mathrm{op}}^2 \, \sigma^2}{T} + C^2 \, (\alpha T)^2,$$

*where $C$ depends only on $L_{\mathrm{dir}}$, $\|PH_s(\theta_r)\|_{\mathrm{op}}$ and $\|r_s^{(\mathcal{S})}(\theta_r)\|_2$. Hence, if $T \to \infty$ and $\alpha T \to 0$,*

$$d_T \xrightarrow{\mathbb{P}} -P\, r_s^{(\mathcal{S})}(\theta_r).$$

This proposition reveals the fundamental trade-off: the MC variance decreases as $1/T$ (so larger $T$ is better), but the curvature bias increases as $(\alpha T)^2$ (so larger $\alpha T$ is worse). The sweet spot is to increase $T$ while keeping $\alpha$ small enough that $\alpha T$ stays small.

**Practical guidance for learning rate and budget.** Lemma B.17 and Proposition B.18 provide concrete guidance for choosing the fine-tuning hyperparameters.

The curvature bias of $d_T$ scales like $\mathcal{O}(\alpha T)$, while the MC variance shrinks like $1/T$. Thus:

- **Decreasing** $\alpha$ reduces bias (keeps the trajectory in the local linear region) but does not change the $1/T$ variance term.

- **Increasing** $T$ averages MC noise but increases bias unless $\alpha$ is reduced so that $\alpha T$ stays small.

A practical regime is

$$\alpha T \;\leq\; \frac{c}{L_{\mathrm{dir}}} \quad \text{and} \quad T \text{ large enough that} \quad \frac{\|P\|_{\mathrm{op}}\sigma}{\sqrt{T}} \;\ll\; \|P\,r_s^{(\mathcal{S})}(\theta_r)\|_2.$$

This balances bias and variance optimally: the budget $\alpha T$ is small enough to stay in the linear regime, while $T$ is large enough to average out MC noise.

**Quadratic proxy for Neon and finite $|\mathcal{S}|$.** Let $r_d(\theta_r) \;:=\; \nabla_\theta \mathcal{R}_{\mathrm{data}}(\theta)\big|_{\theta_r}$ and $H_d \;:=\; \nabla_\theta^2 \mathcal{R}_{\mathrm{data}}(\theta)\big|_{\theta_r}$. Define

$$s_{\mathcal{S}} := \big\langle r_d(\theta_r),\, P\,r_s^{(\mathcal{S})}(\theta_r)\big\rangle, \qquad z_{\mathcal{S}} := \big(r_s^{(\mathcal{S})}(\theta_r)\big)^{\top} P^{\top} H_d\, P\, r_s^{(\mathcal{S})}(\theta_r).$$

For the Neon merge $\theta_{\mathrm{Neon}} = (1+w)\theta_r - w\theta_s$ and short FT, the real-risk change admits the local expansion

$$\Delta\mathcal{R}(w) \;\approx\; w\alpha\,s_{\mathcal{S}} \;+\; \tfrac{1}{2}\,(w\alpha)^2\,z_{\mathcal{S}},$$

with minimizer and minimum

$$w_{\mathcal{S}} = -\frac{s_{\mathcal{S}}}{\alpha z_{\mathcal{S}}}, \qquad \Delta\mathcal{R}_{\mathcal{S}} \;\approx\; -\frac{s_{\mathcal{S}}^2}{2\,z_{\mathcal{S}}}.$$

Using $d_T$ as a plug-in estimate for $-P\,r_s^{(\mathcal{S})}(\theta_r)$, set $\widehat{s}_T := \langle r_d(\theta_r),\, -d_T\rangle$ and $\widehat{z}_T := \langle d_T, H_d d_T\rangle$. Then

$$\widehat{s}_T = s_{\mathcal{S}} + \mathcal{O}_{\mathbb{P}}\big(T^{-1/2} + \alpha T\big), \qquad \widehat{z}_T = z_{\mathcal{S}} + \mathcal{O}_{\mathbb{P}}\big(T^{-1/2} + \alpha T\big),$$

so $\widehat{w} \approx -\widehat{s}_T/(\alpha\widehat{z}_T)$ concentrates on $w_{\mathcal{S}}$ as $T \to \infty$ and $\alpha T \to 0$.

**Explaining the U-shaped performance curve.** One of the striking empirical observations in the main paper (Figure 3) is that Neon's performance as a function of synthetic dataset size $|\mathcal{S}|$ exhibits a U-shape: too few or too many samples both hurt, with an optimal intermediate size. The following remark explains this phenomenon.

*Remark* B.19 (Why performance vs. $|\mathcal{S}|$ is U-shaped). Write $r_s^{(\mathcal{S})} = r_s + \varepsilon_{\mathcal{S}}$ with $\varepsilon_{\mathcal{S}} = \mathcal{O}_{\mathbb{P}}(n^{-1/2})$. Then

$$s_{\mathcal{S}} = \langle r_d, Pr_s\rangle + \langle r_d, P\varepsilon_{\mathcal{S}}\rangle, \qquad z_{\mathcal{S}} = r_s^{\top} P^{\top} H_d Pr_s \;+\; (\text{cross}/\varepsilon_{\mathcal{S}} \text{ terms}).$$

For very small $|\mathcal{S}|$, variance dominates: $s_{\mathcal{S}}$ and $z_{\mathcal{S}}$ are noisy and the attainable improvement $\Delta\mathcal{R}_{\mathcal{S}} \approx -s_{\mathcal{S}}^2/(2z_{\mathcal{S}})$ is weak. For very large $|\mathcal{S}|$, variance vanishes ($\varepsilon_{\mathcal{S}} \to 0$) but the synthetic direction $Pr_s$ tends to align with high-curvature eigenvectors of $H_d$ induced by mode-seeking samplers, increasing $z_{\mathcal{S}}$ faster than $|s_{\mathcal{S}}|$ grows; consequently $|\Delta\mathcal{R}_{\mathcal{S}}|$ shrinks slightly. A moderate $|\mathcal{S}|$ balances these effects: variance is small enough to stabilize $s_{\mathcal{S}}$ while the direction has not collapsed onto the sharpest curvature, keeping $z_{\mathcal{S}}$ moderate. This yields the empirically observed U-shaped curve in Neon performance as a function of $|\mathcal{S}|$.

**Summary.** This subsection connects the idealized population theory to practical implementation. With a fixed, finite synthetic set generated once, *short* fine-tuning (small $\alpha$, modest $T$ so that $\alpha T$ is small) produces a variance-reduced and reliable estimate of the synthetic gradient direction $P\,r_s^{(\mathcal{S})}(\theta_r)$, stabilizing the empirical coefficients $(s_{\mathcal{S}}, z_{\mathcal{S}})$ and the merge weight $w$. Very small $|\mathcal{S}|$ is variance-limited; very large $|\mathcal{S}|$ inflates $z_{\mathcal{S}}$ via curvature. Therefore, a broad, *moderate* $|\mathcal{S}|$ is typically best, as confirmed by the U-shaped performance curve observed empirically. These results provide theoretical justification for the hyperparameter choices used in practice and explain why Neon is robust to reasonable variations in synthetic dataset size and fine-tuning budget.

### B.11 TOY EXPERIMENT

Now we present a toy experiment to empirically validate and provide deeper intuition for the theoretical results presented in the paper. The goal is to create a controlled environment where we can directly observe the effects of sampler behavior on self-training and measure the key theoretical quantity: the directional alignment between gradients.

**Setup.** The task is to learn a 2D Gaussian distribution, $\mathcal{N}(\mu_{\text{ref}}, \Sigma_{\text{ref}})$, where the mean is $\mu_{\text{ref}} = [0, 0]^\top$ and the covariance is $\Sigma_{\text{ref}} = [2, 1; 1, 2]^\top$. We use a small Denoising Diffusion Probabilistic Model (DDPM) with an MLP backbone, trained over a short diffusion process of $T = 20$ steps with a cosine noise schedule. A base model, $\theta_r$, is trained for a long duration ($10,000$ epochs) on a small dataset of $N_{\text{base}} = 10^3$ real samples with a learning rate of $10^{-4}$ to ensure it has converged.

To control the sampler's behavior during synthetic data generation, we introduce a scalar hyperparameter, $\zeta$, which directly scales the model's score. The standard score is defined as $s_\theta(x_t, t) = -\epsilon_\theta(x_t, t)/\sqrt{1 - \bar{\alpha}_t}$, where $\epsilon_\theta$ is the model's noise prediction. During sampling, we use a modified score, $\tilde{s}_\theta(x_t, t) = \zeta \cdot s_\theta(x_t, t)$, to generate samples. This allows us to precisely control the sampler's characteristics:

- $\zeta > 1$: The sampler becomes *mode-seeking*.
- $\zeta < 1$: The sampler becomes *diversity-seeking*.
- $\zeta = 1$: The sampler is neutral.

**Experiment 1: FID vs. Merge Weight.** In our first experiment, we validate the main prediction of our paper. We generate synthetic datasets using a mode-seeking sampler ($\zeta = 1.1$) and a diversity-seeking sampler ($\zeta = 0.9$). We then fine-tune $\theta_r$ on each of these datasets to obtain a self-trained model $\theta_s$. We form a merged model via the one-parameter extrapolation formula:

$$\theta_w = (1 + w)\theta_r - w\theta_s = \theta_r - w(\theta_s - \theta_r)$$

A positive weight ($w > 0$) corresponds to Neon's *negative extrapolation*, moving away from the self-trained model. A negative weight ($w < 0$) corresponds to *positive extrapolation* (interpolation). Letting $w = -\alpha$ for $\alpha > 0$, the formula becomes $\theta_w = (1 - \alpha)\theta_r + \alpha\theta_s$, which is standard interpolation and equivalent to a step of self-training.

The results, shown in Figure B.2, perfectly match our theory. For the mode-seeking sampler, the optimal FID is achieved at $w^* > 0$, demonstrating that negative extrapolation (Neon) helps. Conversely, for the diversity-seeking sampler, the optimal FID is achieved at $w^* < 0$, showing that positive extrapolation (self-training) is beneficial.

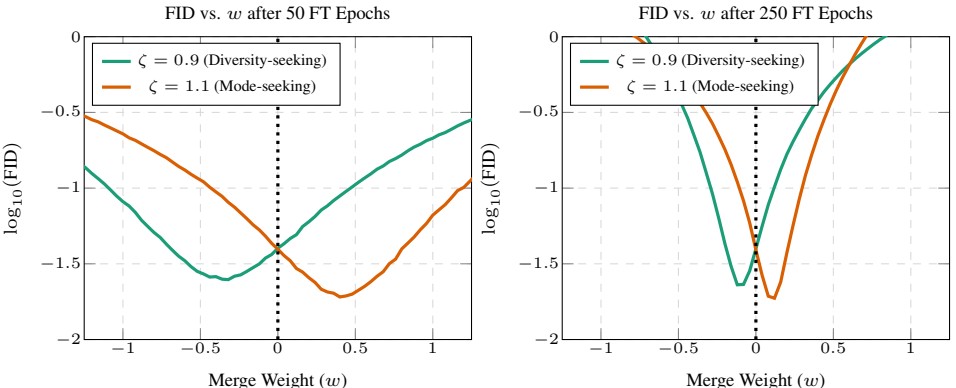

Figure B.2: **FID vs. Merge Weight ($w$) validation.** For the mode-seeking sampler ($\zeta = 1.1$), the optimal FID is at $w > 0$ (Neon helps). For the diversity-seeking sampler ($\zeta = 0.9$), the optimum is at $w < 0$ (self-training helps).

**Experiment 2: Gradient Alignment vs. Sampler Type.** In our second experiment, we directly measure the directional alignment between the real and synthetic gradients by computing their cosine

similarity, $\cos(\vartheta) = \frac{\langle r_d, P_{\text{Adam}} r_s \rangle}{\|r_d\|_{P_{\text{Adam}}} \|r_s\|_{P_{\text{Adam}}}}$. We estimate the population real-data gradient $r_d$ and the Adam preconditioner $P_{\text{Adam}}$ from a large set of $N_{\text{pop}} = 10^5$ real samples. We then sweep the score scale $\zeta$ across the range $[0.8, 1.25]$ and compute the cosine similarity for each value.

The results in Figure B.3 provide a clear visualization of the alignment direction. The cosine similarity is positive for diversity-seeking samplers ($\zeta < 1$), corresponding to an acute angle between the gradients. This confirms they are aligned, and self-training should help. The similarity becomes negative for mode-seeking samplers ($\zeta > 1$), corresponding to an obtuse angle. This confirms they are anti-aligned, and negative extrapolation (Neon) is the correct approach. Furthermore, we note that **at the neutral point $\zeta = 1$, the cosine similarity is still negative.** This provides a powerful validation of our theoretical finding (Appendix B.7) that any practical, finite-step ODE solver—which our DDPM sampler is an instance of—introduces a small discretization error that is inherently mode-seeking, thus producing a negative alignment even without explicit score scaling.

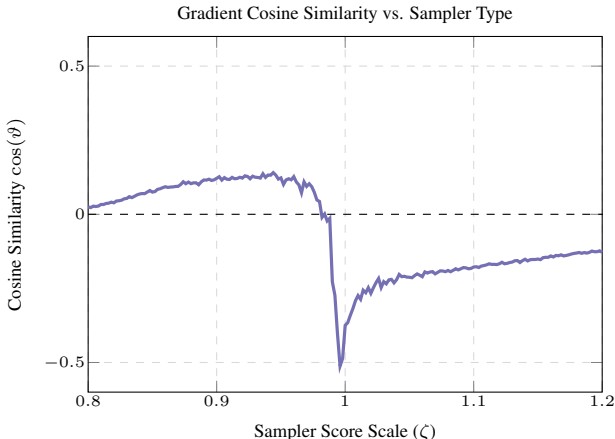

Figure B.3: **Direct measurement of the gradient alignment direction.** The cosine similarity $\cos(\vartheta)$ is positive for diversity-seeking samplers ($\zeta < 1$) and negative for mode-seeking samplers ($\zeta > 1$), crossing zero at the neutral point $\zeta = 1$.

## C  Experiments Details

A key advantage of Neon is its implementation simplicity. Given an existing training and generation script for a base model, Neon requires only a minimal add on script that takes two model checkpoints and a weight $w$ to construct the final model parameters. To ensure reproducibility and build directly on prior work, all our experiments start from official public codebases and use publicly available pre trained checkpoints as our base models. The repositories we used for each model family are listed below:

- **Diffusion Models (EDM):** NVlabs/edm
- **Flow Matching:** atong01/conditional-flow-matching
- **Autoregressive Models (VAR, xAR):** FoundationVision/VAR and OliverRensu/xAR
- **Few Step Models (IMM):** lumalabs/imm

For the fine tuning stage, we adhere closely to the default training configurations proposed by the original authors for each model. Our primary modification involves adapting the learning rate policy for the fine tuning context. This typically means using a small target learning rate, which in some cases is reached via a linear warmup schedule. All other settings, such as the optimizer and batch size, remain unchanged. During this process, we save model checkpoints periodically (typically every 250k or 500k images seen) to evaluate performance over the course of training.

Our evaluation procedure is as follows. For each saved checkpoint, we perform a hyperparameter search to find the optimal merge weight $w$ (and CFG scale $\gamma$, where applicable). This search is conducted by generating 10k samples per setting to calculate a preliminary FID score. Once the optimal hyperparameters are identified, we generate a final set of 50k samples to compute the final FID score reported in this paper.

Below, we detail the specific configurations for each experiment.

**EDM-VP on CIFAR-10.**

- $\mathcal{S}$ **Generation:** Generated with `-steps=18 -rho=7 -S_churn=0`.
- **Fine tuning:** Default script of `-cond=1 -arch=ddpmpp` with a modified `-lr=1e-4`. For the unconditional experiment, the script used `-cond=0`.
- **Neon Evaluation:** Grid search over merge weight $w \in [0, 3.0]$.

**EDM-VP on FFHQ-64.**

- $\mathcal{S}$ **Generation:** Generated with `-steps=40 -rho=7 -S_churn=0`.
- **Fine tuning:** Default script of `-cond=0 -arch=ddpmpp -batch=256 -cres=1,2,2,2 -dropout=0.05 -augment=0.15` with a modified `-lr=4e-6`.
- **Neon Evaluation:** Grid search over merge weight $w \in [0, 3.0]$.

**Flow Matching on CIFAR-10.**

- $\mathcal{S}$ **Generation:** Generated using the `dopri5` ODE solver with `-integration-steps=100`.
- **Fine tuning:** Default script of `-ema_decay=0.9999` with a modified learning rate of `-lr=2e-4`.
- **Neon Evaluation:** Grid search over merge weight $w \in [0, 3.0]$.

**xAR-B on ImageNet-256.**

- $\mathcal{S}$ **Generation:** Generated with `-cfg=2.7 -flow_steps=40 -num_iter=256`.
- **Fine tuning:** Default script of `-model=xar_base -vae_embed_dim=16 -vae_stride=16` with a modified `-blr=1e-6`, using a linear warmup schedule over the 7 Mi images seen.
- **Neon Evaluation:** Joint grid search over merge weight $w \in [0, 3.0]$ and CFG scale $\gamma \in [2.7, 5.0]$.

**xAR-L on ImageNet-256.**

- $\mathcal{S}$ **Generation:** Generated with `-cfg=2.3 -flow_steps=50 -num_iter=256`.
- **Fine tuning:** Default script of `-model=xar_large -vae_embed_dim=16 -vae_stride=16` with a modified `-blr=1e-6`, using a linear warmup schedule over the 7 Mi images seen.
- **Neon Evaluation:** Joint grid search over merge weight $w \in [0, 3.0]$ and CFG scale $\gamma \in [2.3, 5.0]$.

**VAR-d16 on ImageNet-256.**

- $\mathcal{S}$ **Generation:** Generated with `-cfg=1.25 -top_k=900 -top_p=0.95 -model_depth=16`.
- **Fine tuning:** Default script of `-depth=16 -bs=786 -fp16=1 -alng=1e-4`, modified to use a linear warmup to a target learning rate of `1e-5` over 7.5 Mi images seen.
- **Neon Evaluation:** Joint grid search over merge weight $w \in [0, 2.0]$ and CFG scale $\gamma \in [1.25, 4.0]$.

**VAR-d30 on ImageNet-512.**

- $\mathcal{S}$ **Generation:** Generated with `-cfg=2.0 -top_k=900 -top_p=0.95 -model_depth=16`.
- **Fine tuning:** Default script of `-depth=36 -bs=24 -fp16=1 -alng=5e-6 -saln=1 -pn=512`, modified to use a linear warmup to a target learning rate of `1e-5` over 3 Mi images seen.
- **Neon Evaluation:** Joint grid search over merge weight $w \in [0, 2.0]$ and CFG scale $\gamma \in [2.0, 4.5]$.

**IMM on ImageNet-256.**

- $\mathcal{S}$ **Generation:** Generated using the `imagenet256_ts_a2.pkl` model with `-T=8 -cfg_scale=1.5`.
- **Fine tuning:** Default training script with a modified learning rate of `-lr=1e-6`.
- **Neon Evaluation:** For each $T \in \{1, 2, 4, 8\}$, a joint grid search over $w \in [0, 5.0]$ and $\gamma \in [1.0, 3.0]$.

**Metric Calculation Details.** For the EDM and flow matching models, we used the official FID calculation script from the NVlabs/edm repository. The pre computed reference statistics were downloaded from the URL provided by the authors. For all autoregressive (xAR, VAR) and few step (IMM) models, we used the `torch-fidelity` library. The reference statistics for ImageNet were sourced from the openai/guided-diffusion repository. For Precision and Recall, we extracted InceptionV3 features and computed the metrics using the `prdc` library with $k = 5$.

**Practical Note on Normalization Layers.** The Neon merge, $\theta_{\text{Neon}} = (1 + w)\theta_r - w\theta_s$, is applied directly to model parameters. The architectures in our experiments use LayerNorm, GroupNorm, or RMSNorm; since these do not have running statistics, no special handling (e.g., recomputing statistics with a forward pass) is required.

**Practical Note on Mask Buffers.** The Neon merge applies only to the learned parameters ($\theta$) of a model. Architectures like xAR may use fixed buffers for attention masks containing infinity values. These buffers are not parameters and should be excluded from the merge. We follow the standard practice of copying all buffers directly from the base model $\theta_r$.

**Practical Note on Numerical Precision.** Some models use half precision (`fp16`). Performing the merge directly in `fp16` using $(1 + w)\theta_r - w\theta_s$ can cause numerical overflow. To ensure stability, we recommend one of two approaches:

1. Perform the merge in `fp16` using the more stable formula: $\theta_r - w(\theta_s - \theta_r)$.

2. Cast weights to a higher precision (e.g., `fp32`) before merging, then cast back to `fp16`.

We use the first approach in our implementation for its stability and efficiency.

**Practical Note on Efficient Hyperparameter Search.** While we performed a full grid search for thoroughness, a more efficient search is possible in practice. The relationship between the merge weight $w$ and FID is strongly unimodal and locally quadratic. For finding an optimal $w$, one can use standard 1D optimization algorithms like Brent's Method (Brent, 1973). For jointly optimizing $w$ and $\gamma$, this extends to fitting a 2D quadratic surface, which we found requires only six well-distributed points to find a near-optimal configuration.

# D    ADDITIONAL EXPERIMENTS FOR DIFFUSION AND FLOW MATCHING MODELS

We extend the precision-recall analysis from Section 4.1 to additional diffusion and flow matching experiments. Figure D.1 presents the complete FID, precision, and recall curves as a function of merge weight $w$ for EDM-VP on FFHQ-64 and Flow Matching on CIFAR-10.

For EDM-VP on FFHQ-64 (top row), we observe similar dynamics to those discussed in the main text. The FID curves exhibit the characteristic U-shape with optimal values around $w \approx 1.0$–$1.5$, achieving FID as low as 1.12 from a baseline of 2.39. The precision monotonically decreases with increasing $w$, dropping from approximately 0.78 to 0.40 as $w$ increases from 0 to 3. The recall shows the expected inverted-U pattern, peaking near the FID-optimal weight and demonstrating that Neon's improvement stems from recovering under-represented modes. As the fine-tuning budget increases from 1.5 Mi to 3 Mi, the effects become more pronounced: the FID improvement deepens, the precision drop steepens, and the recall peak sharpens.

For Flow Matching on CIFAR-10 (bottom row), the pattern is consistent but with model-specific characteristics. The baseline FID of 3.5 improves to 2.32 at optimal $w \approx 1.0$. The precision-recall trade-off is less extreme than for EDM-VP, with precision declining from approximately 0.73 to 0.55 and recall peaking around 0.72. This suggests that flow matching models may have a different mode coverage profile compared to diffusion models, but still benefit from Neon's redistribution mechanism. The optimal merge weight remains relatively stable across different fine-tuning budgets, indicating robust degradation directions for this architecture.

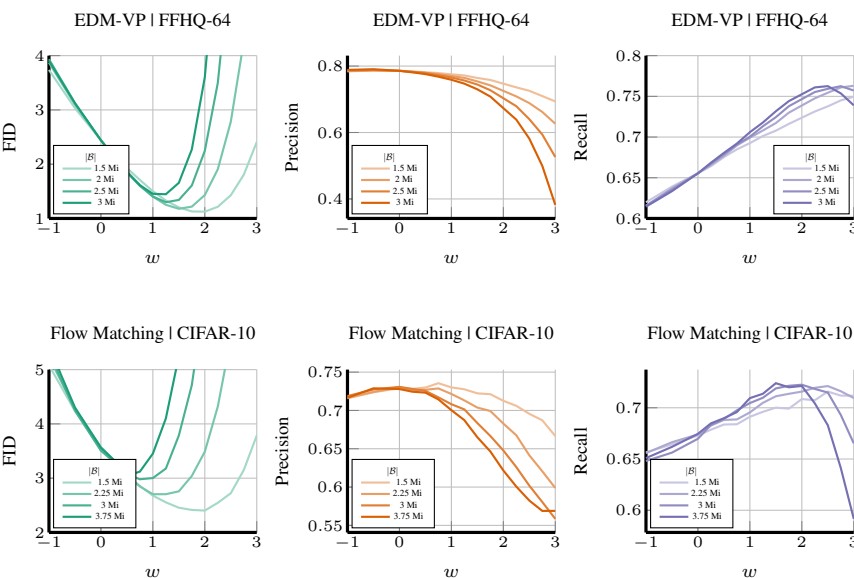

Figure D.1: **Neon's precision-recall trade-off across diffusion and flow matching architectures.** FID, precision, and recall as functions of merge weight $w$ for EDM-VP on FFHQ-64 with $|\mathcal{S}| = 18\text{k}$ (top row) and Flow Matching on CIFAR-10 with $|\mathcal{S}| = 25\text{k}$ (bottom row), shown across different fine-tuning budgets $\mathcal{B}$. Both architectures exhibit the characteristic pattern: FID reaches a minimum at intermediate $w$ values, precision monotonically decreases, and recall follows an inverted-U curve peaking near the FID optimum.

# E   xAR-B on Imagenet-256 synthesized images

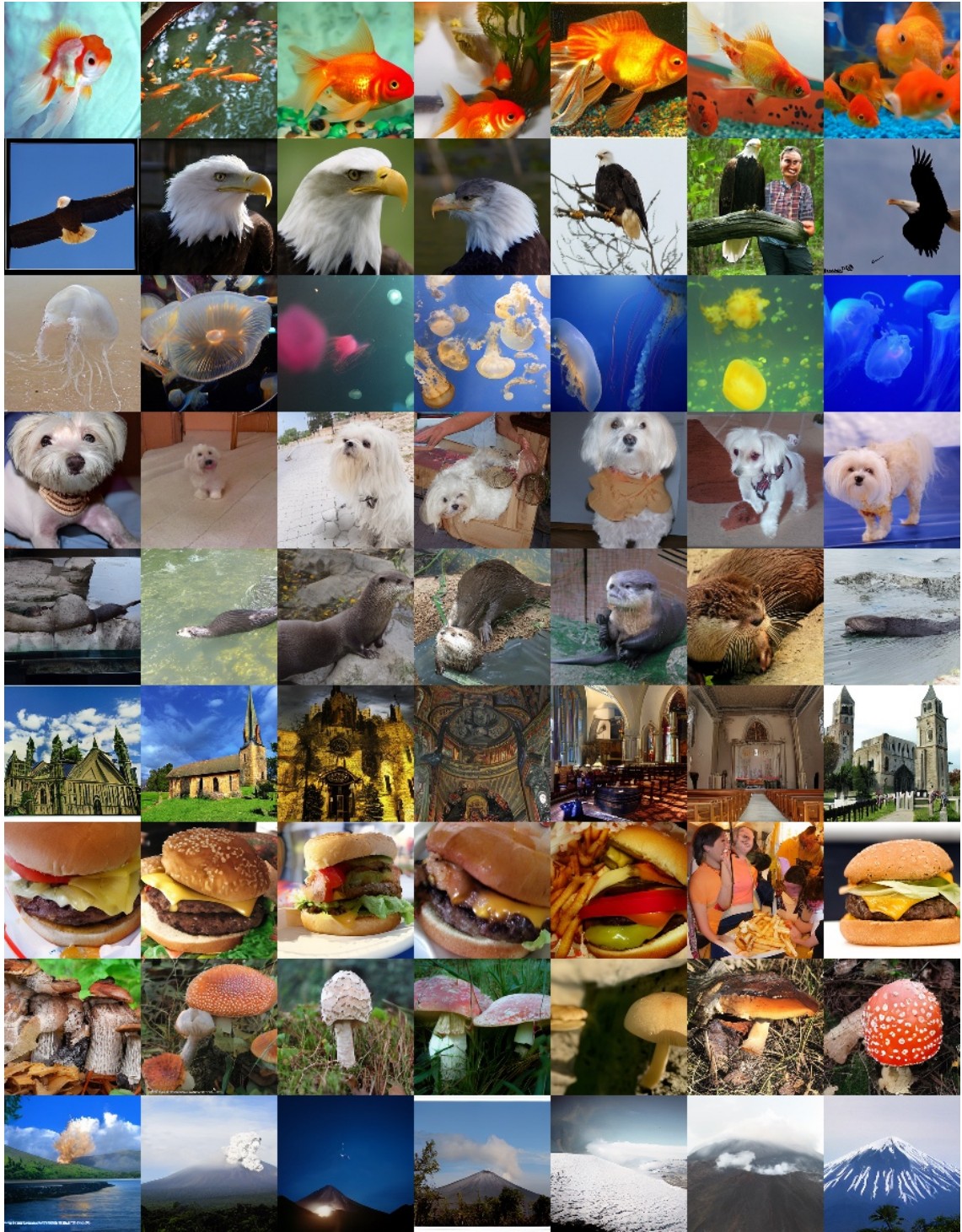

Figure E.1: Images generated from Neon model with $\mathcal{B} = 4.25$ (Mi), $w = 1.4$, $\gamma = 3.8, |\mathcal{S}| = 750k$, FID $= 1.31$

## F  XAR-L ON IMAGENET-256 SYNTHESIZED IMAGES

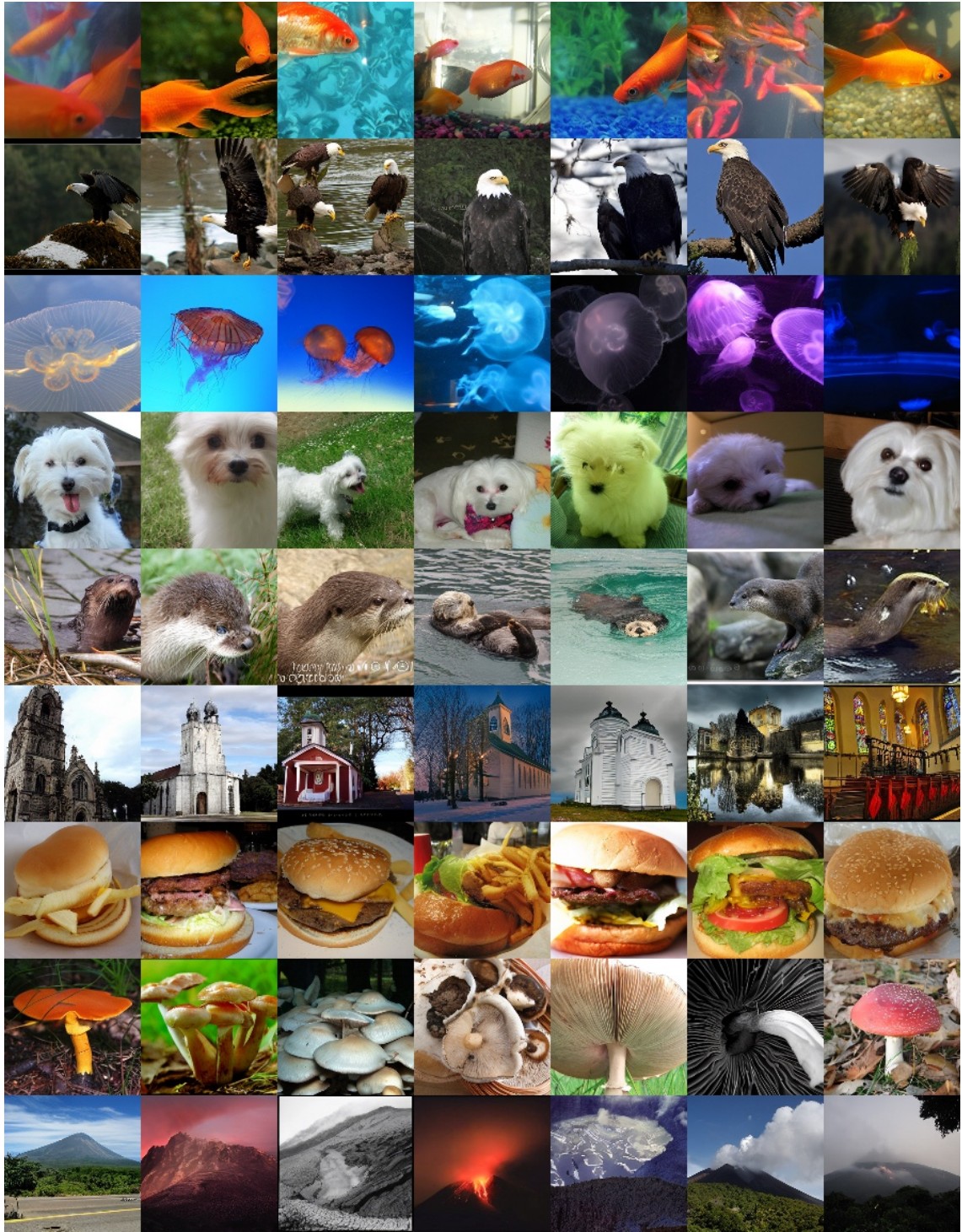

Figure F.1: Images generated from Neon model with $\mathcal{B} = 3.75$ (Mi), $w = 1.6$, $\gamma = 2.7$, $|\mathcal{S}| = 750k$, FID $= 1.02$

## G VAR-D16 ON IMAGENET-256 SYNTHESIZED IMAGES

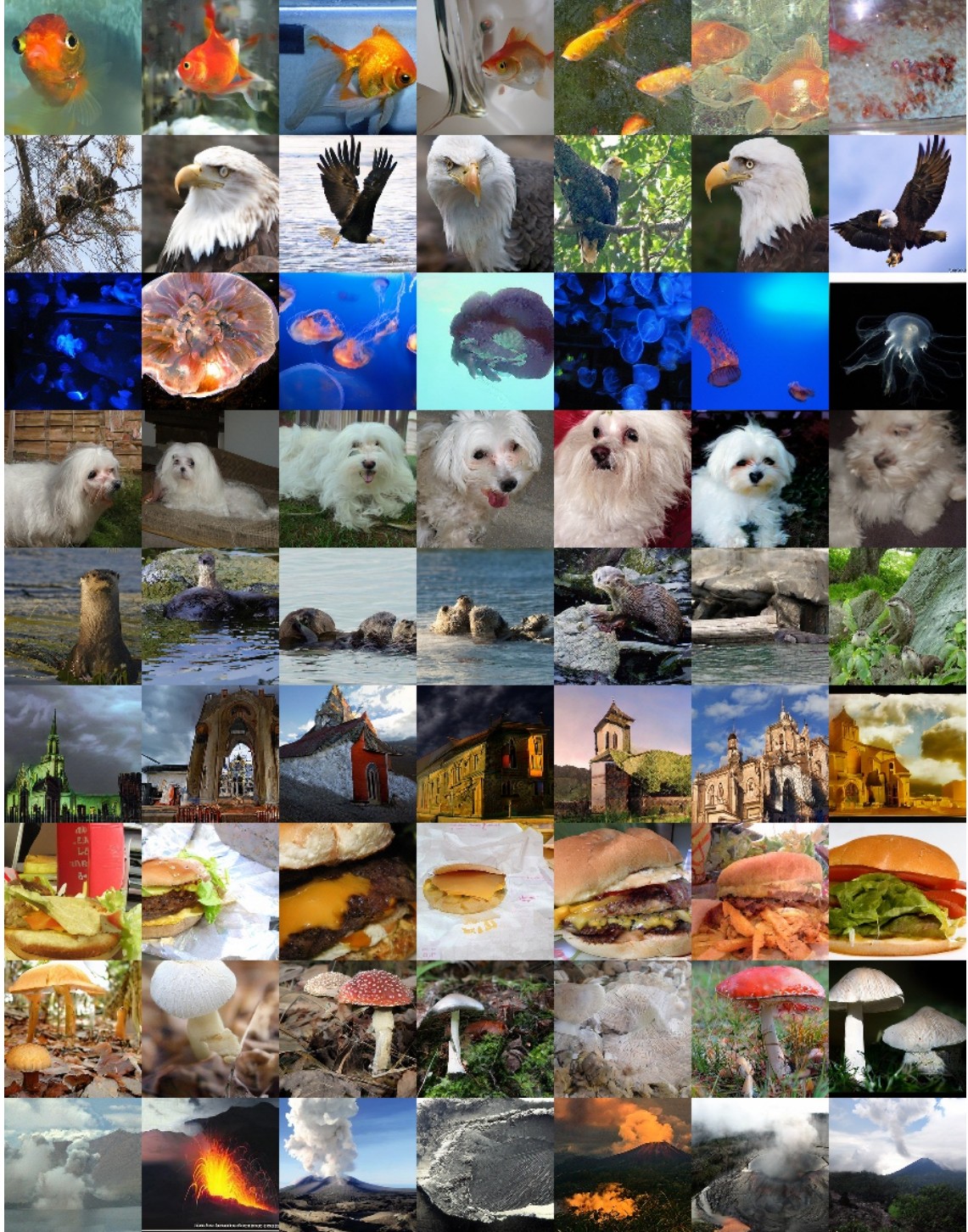

Figure G.1: Images generated from Neon model with $\mathcal{B} = 1.25$ (Mi), $w = 1$, $\gamma = 2.9$, $|\mathcal{S}| = 750$k, FID $= 2.01$

# H IMM ON IMAGENET-256 SYNTHESIZED IMAGES

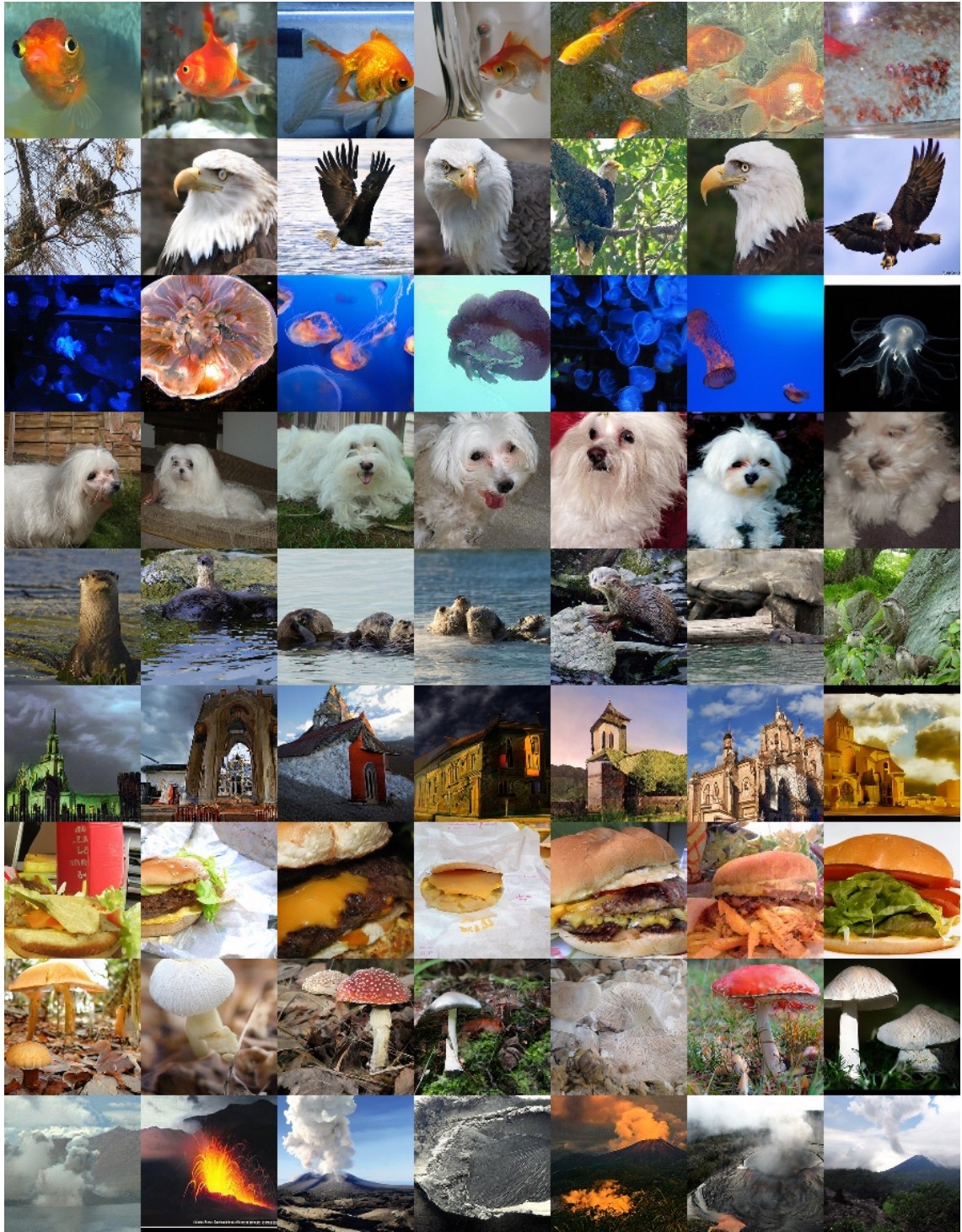

Figure H.1: Images generated from Neon model with $\mathcal{B} = 1.95(\text{Mi})$, $w = 3.6$, $\gamma = 1.8$, $|\mathcal{S}| = 30\text{k}$, FID $= 1.45$

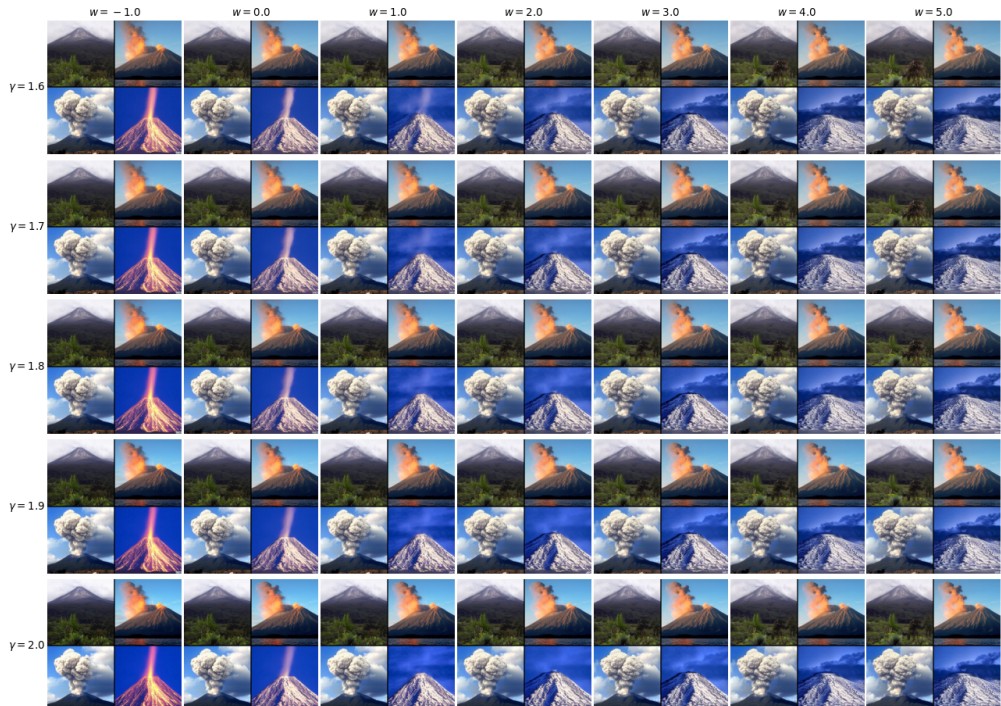

Figure H.2: Effect of negative extrapolation weight $w$ and CFG scale $\gamma$ on IMM generation quality for ImageNet class 980 (valley). Each 2×2 grid shows four random samples for a given $(w, \gamma)$ configuration for snapshot of model at $\mathcal{B} = 1.95$ (Mi) for $|\mathcal{S}| = 30$k.

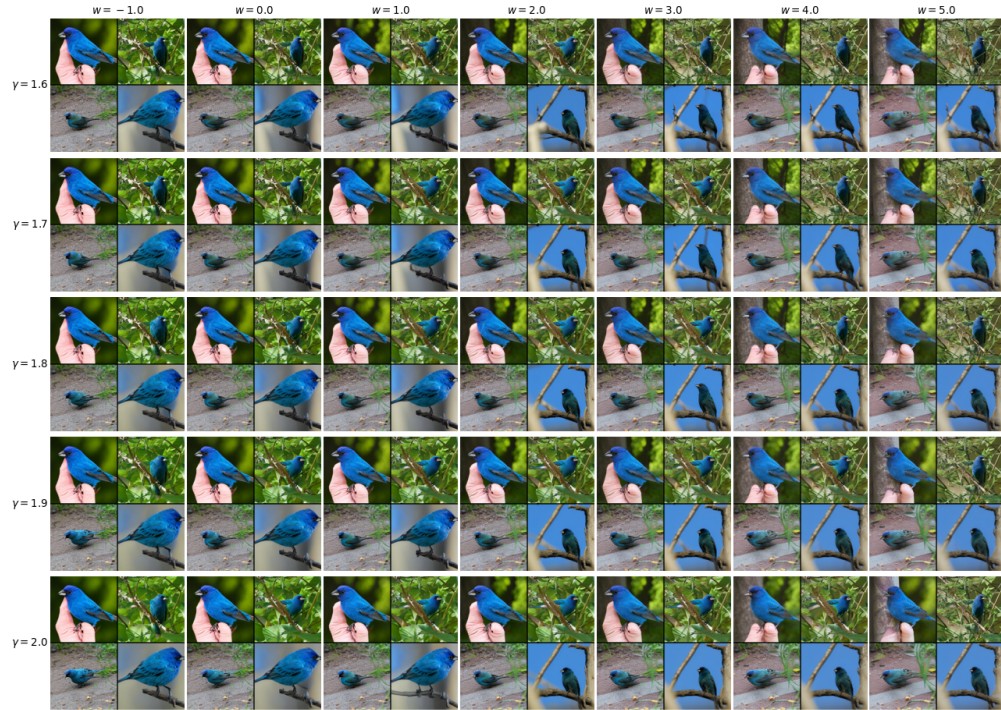

Figure H.3: Effect of negative extrapolation weight $w$ and CFG scale $\gamma$ on IMM generation quality for ImageNet class 14 (indigo bunting). Each 2×2 grid shows four random samples for a given $(w, \gamma)$ configuration for snapshot of model at $\mathcal{B} = 1.95$ (Mi) for $|\mathcal{S}| = 30$k.

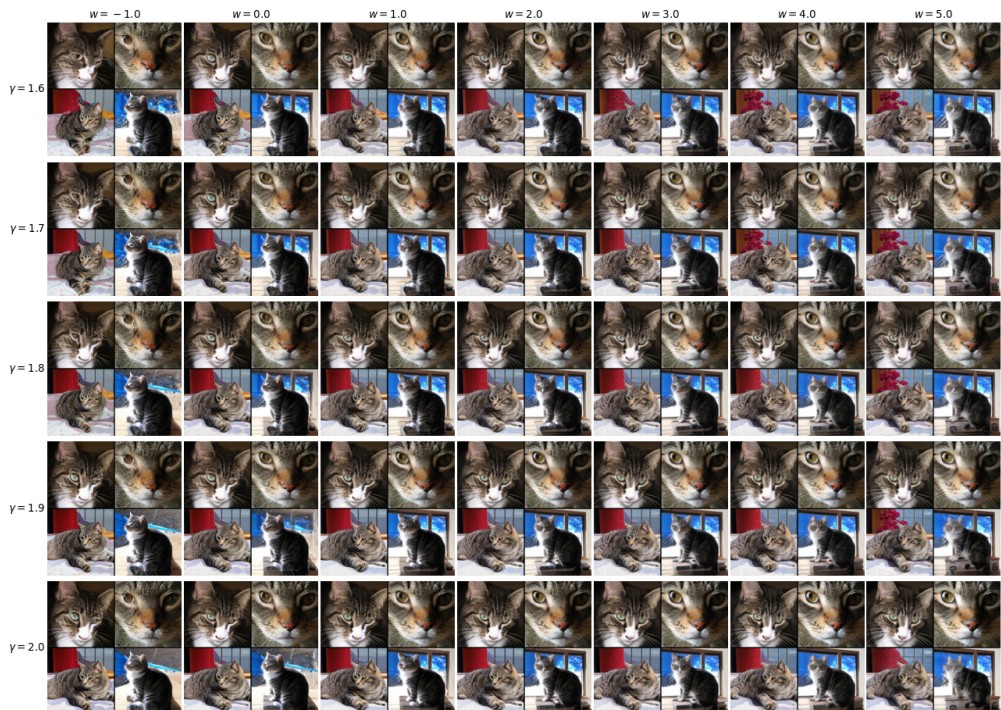

Figure H.4: Effect of negative extrapolation weight $w$ and CFG scale $\gamma$ on IMM generation quality for ImageNet class 281 (tabby cat). Each 2×2 grid shows four random samples for a given $(w, \gamma)$ configuration for snapshot of model at $\mathcal{B} = 1.95$ (Mi) for $|\mathcal{S}| = 30k$.

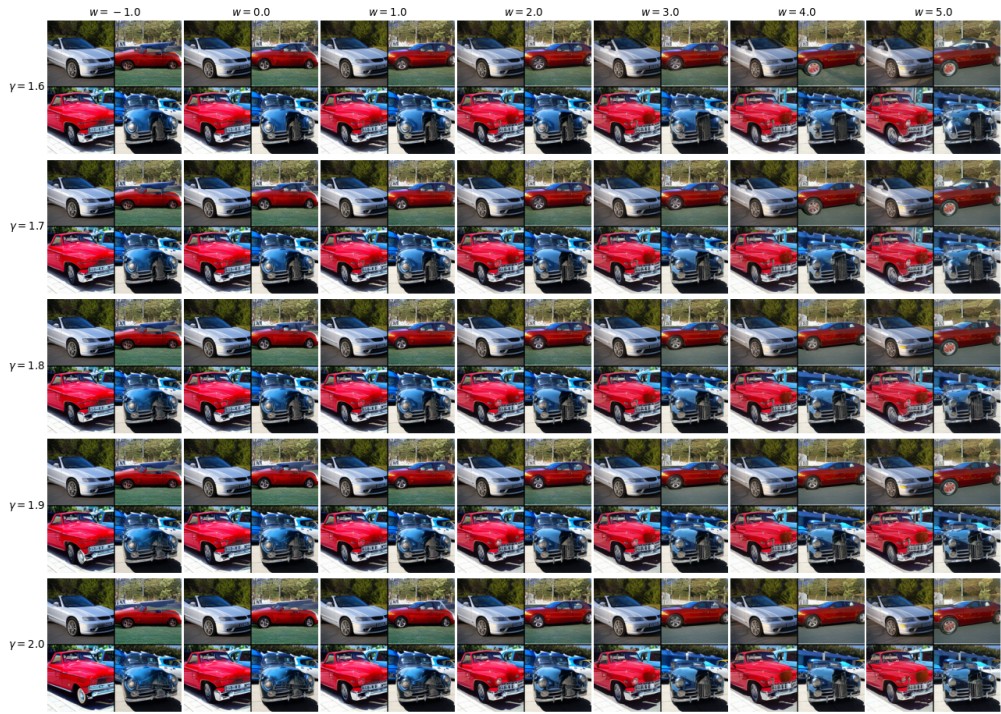

Figure H.5: Effect of negative extrapolation weight $w$ and CFG scale $\gamma$ on IMM generation quality for ImageNet class 511 (container ship). Each 2×2 grid shows four random samples for a given $(w, \gamma)$ configuration for snapshot of model at $\mathcal{B} = 1.95$ (Mi) for $|\mathcal{S}| = 30k$.

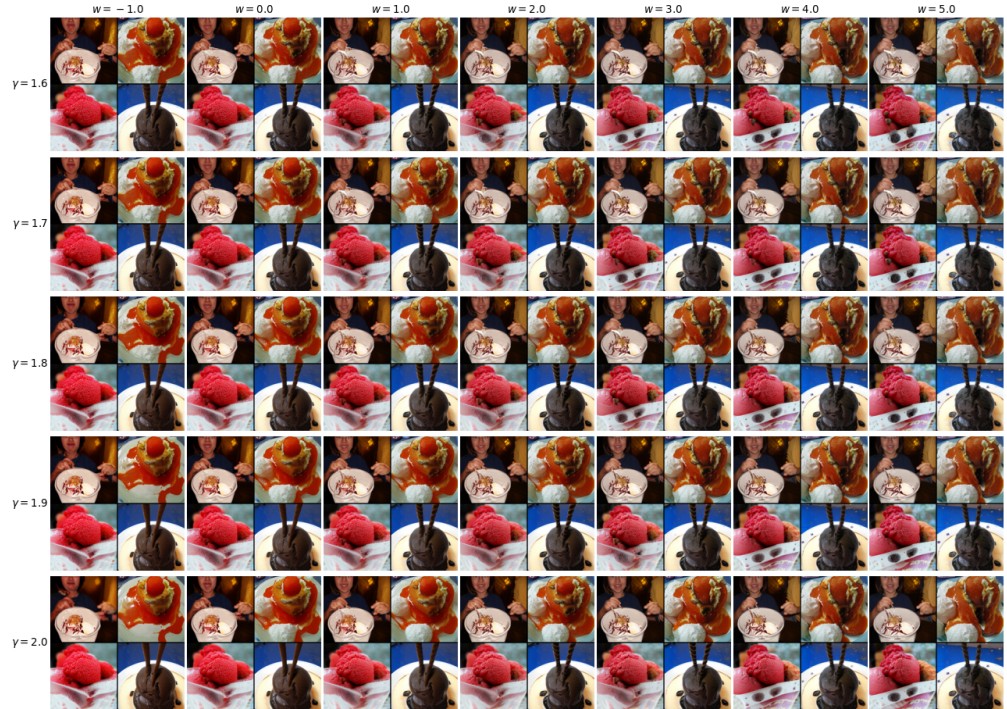

Figure H.6: Effect of negative extrapolation weight $w$ and CFG scale $\gamma$ on IMM generation quality for ImageNet class 928 (ice cream). Each 2×2 grid shows four random samples for a given $(w, \gamma)$ configuration for snapshot of model at $\mathcal{B} = 1.95$ (Mi) for $|\mathcal{S}| = 30$k.

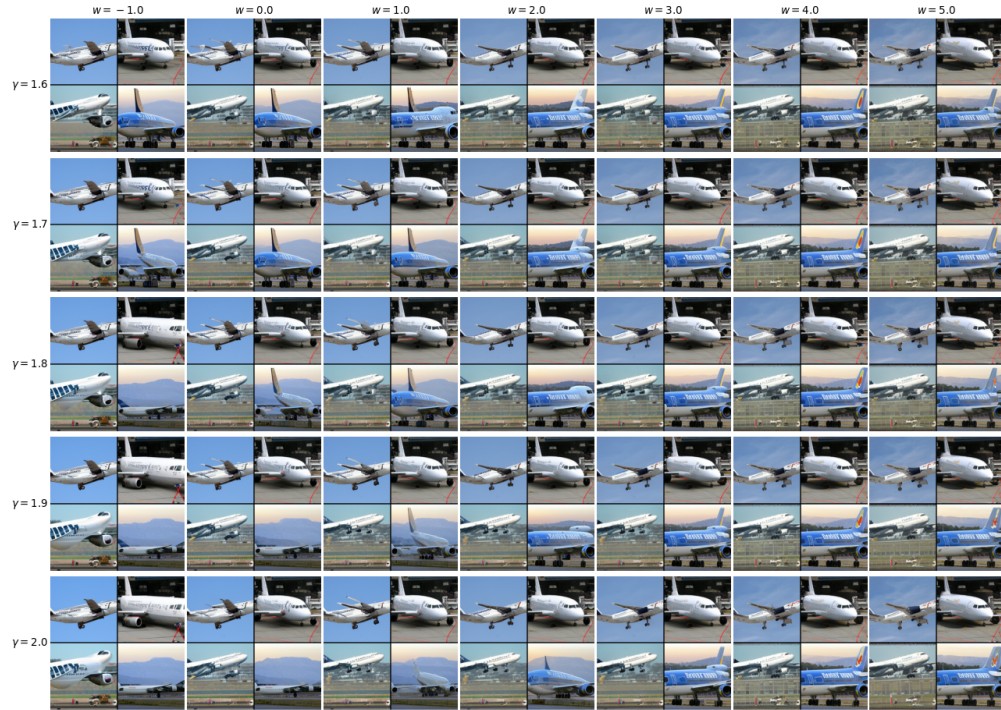

Figure H.7: Effect of negative extrapolation weight $w$ and CFG scale $\gamma$ on IMM generation quality for ImageNet class 404 (airliner). Each 2×2 grid shows four random samples for a given $(w, \gamma)$ configuration for snapshot of model at $\mathcal{B} = 1.95$ (Mi) for $|\mathcal{S}| = 30$k.

# I VAR-D36-S ON IMAGENET-512 SYNTHESIZED IMAGES

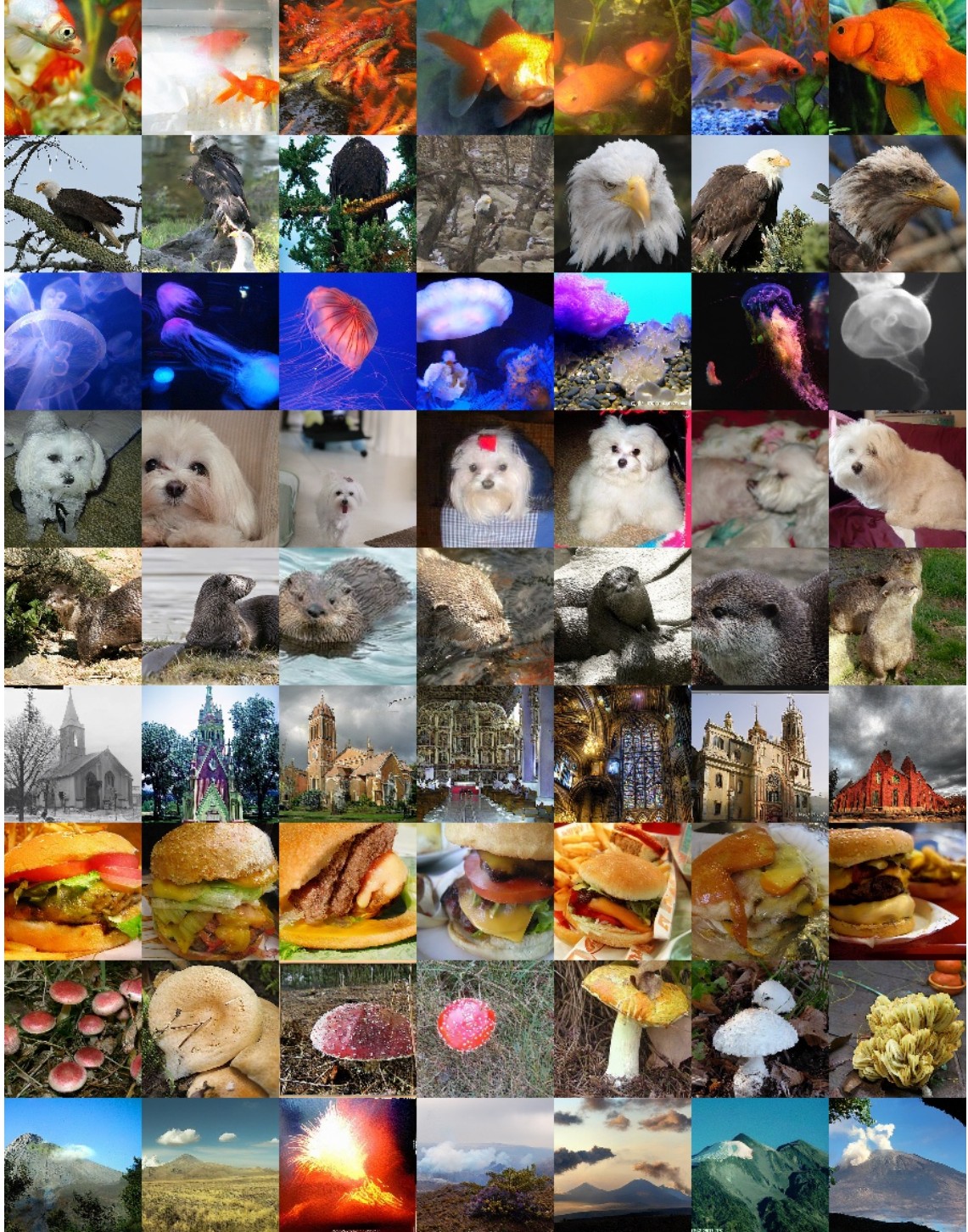

Figure I.1: Images generated from Neon model with $\mathcal{B} = 1.20$ (Mi), $w = 0.6$, $\gamma = 3.2$, $|\mathcal{S}| = 90$k, FID $= 1.69$

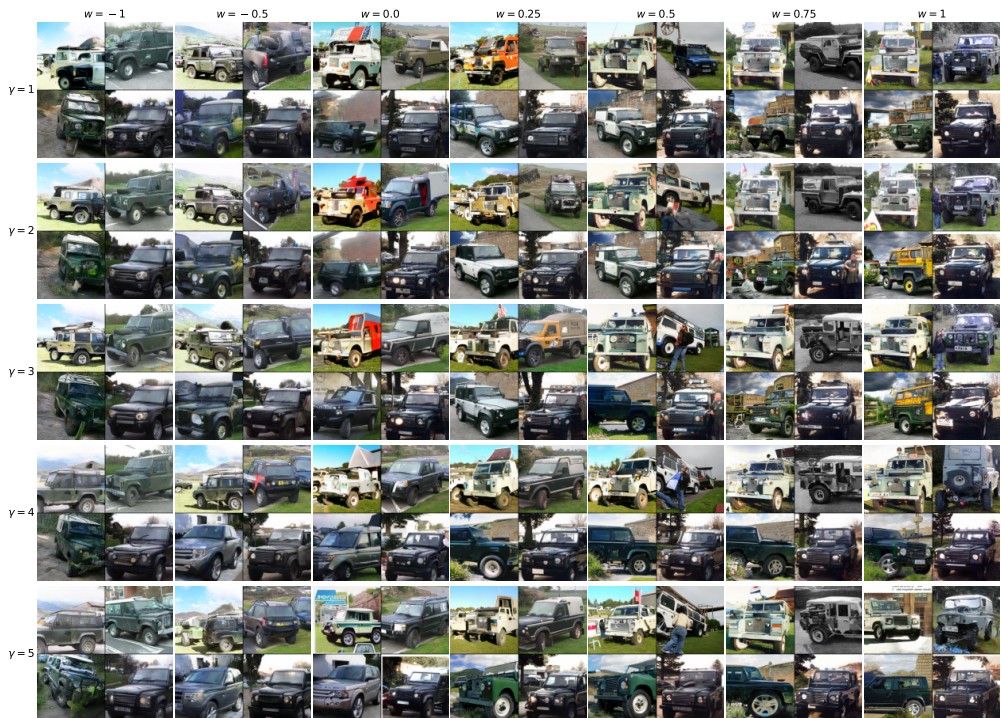

Figure I.2: Effect of negative extrapolation weight $w$ and CFG scale $\gamma$ on VAR-d36-s generation quality for ImageNet class 609 (jeep). Each 2×2 grid shows four random samples for a given $(w, \gamma)$ configuration for snapshot of model at $B = 1.20$ (Mi) for $|S| = 90$k.

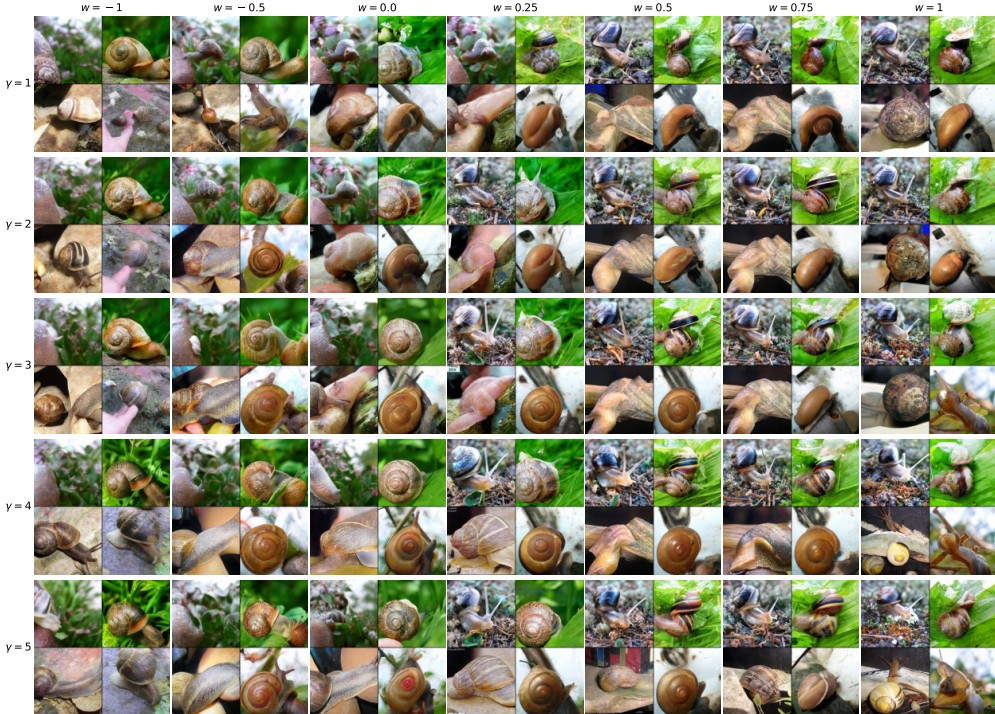

Figure I.3: Effect of negative extrapolation weight $w$ and CFG scale $\gamma$ on VAR-d36-s generation quality for ImageNet class 113 (snail). Each 2×2 grid shows four random samples for a given $(w, \gamma)$ configuration for snapshot of model at $B = 1.20$ (Mi) for $|S| = 90$k.

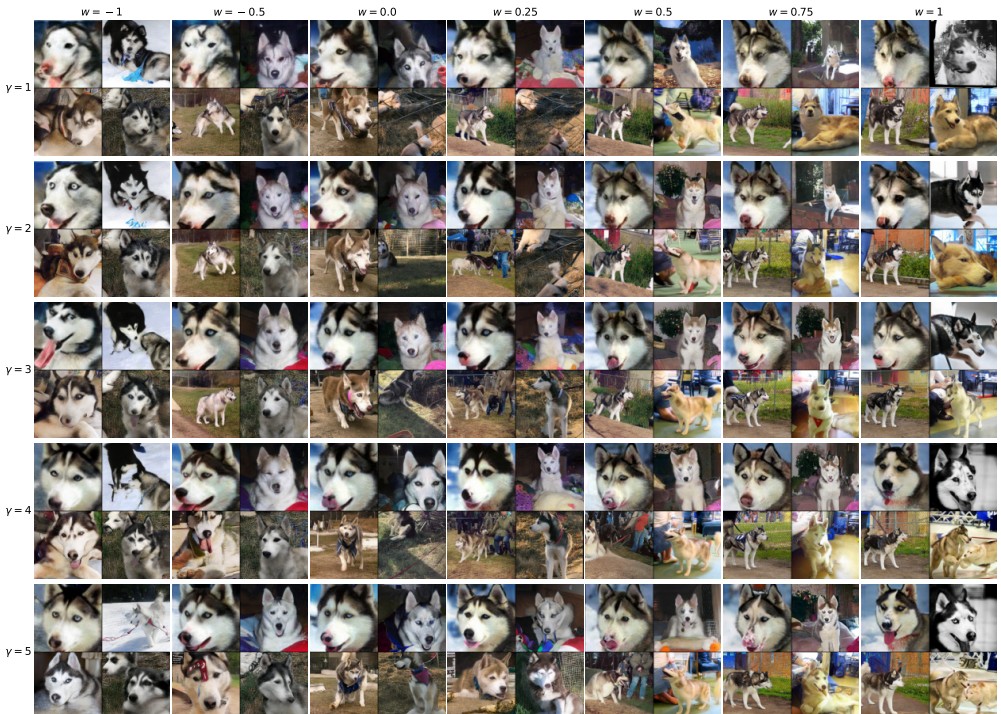

Figure I.4: Effect of negative extrapolation weight $w$ and CFG scale $\gamma$ on VAR-d36-s generation quality for ImageNet class 248 (husky). Each 2×2 grid shows four random samples for a given $(w, \gamma)$ configuration for snapshot of model at $B = 1.20$ (Mi) for $|S| = 90$k.

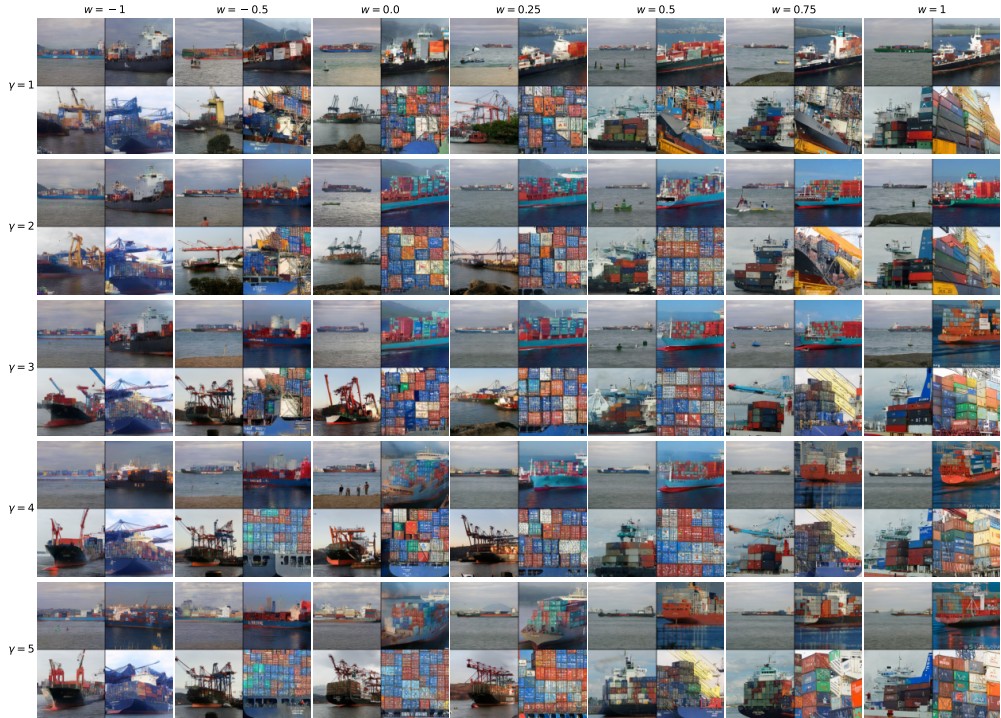

Figure I.5: Effect of negative extrapolation weight $w$ and CFG scale $\gamma$ on VAR-d36-s generation quality for ImageNet class 510 (container ship). Each 2×2 grid shows four random samples for a given $(w, \gamma)$ configuration for snapshot of model at $B = 1.20$ (Mi) for $|S| = 90$k.

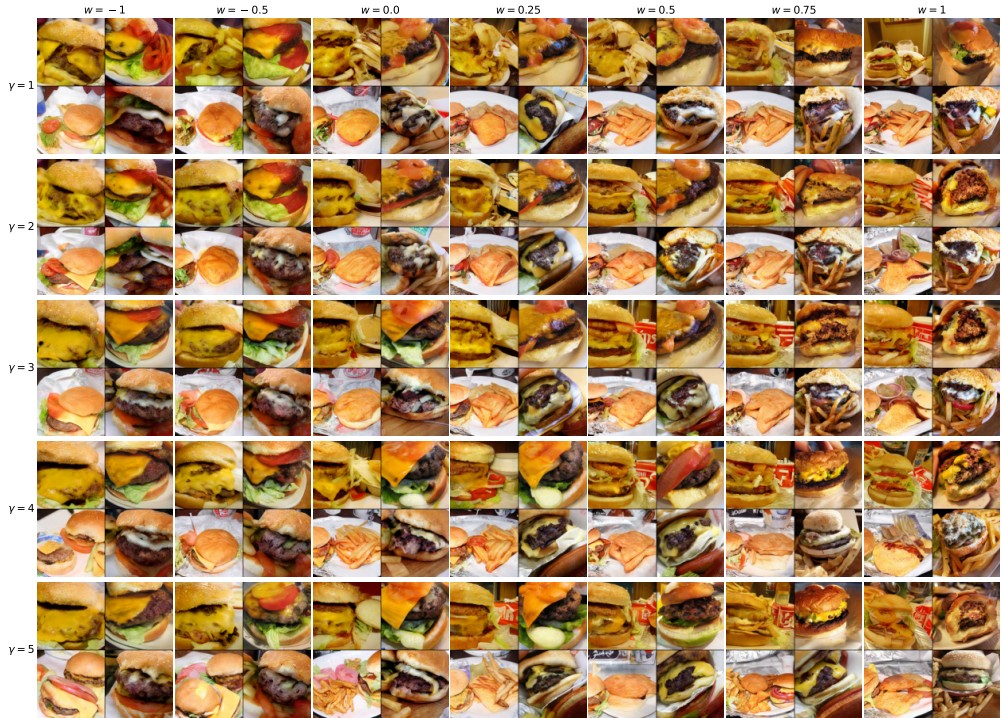

Figure I.6: Effect of negative extrapolation weight $w$ and CFG scale $\gamma$ on VAR-d36-s generation quality for ImageNet class 933 (cheeseburger). Each 2×2 grid shows four random samples for a given $(w, \gamma)$ configuration for snapshot of model at $B = 1.20$ (Mi) for $|S| = 90k$.

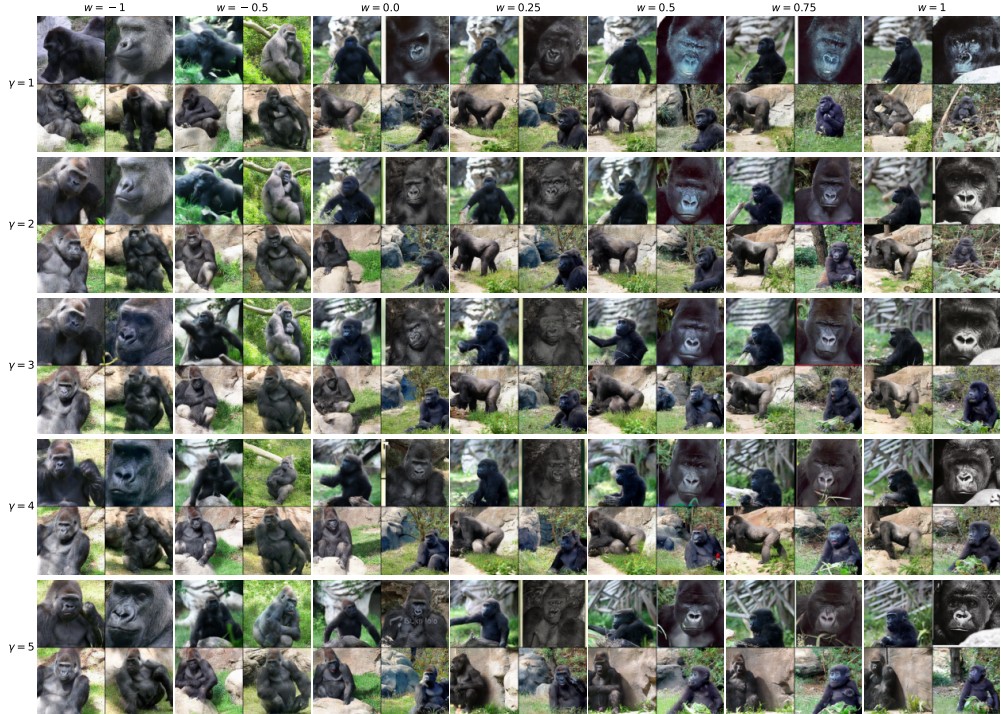

Figure I.7: Effect of negative extrapolation weight $w$ and CFG scale $\gamma$ on VAR-d36-s generation quality for ImageNet class 933 (cheeseburger). Each 2×2 grid shows four random samples for a given $(w, \gamma)$ configuration for snapshot of model at $B = 1.20$ (Mi) for $|S| = 90k$.

## J EDM-VP ON FFHQ-64 SYNTHESIZED IMAGES

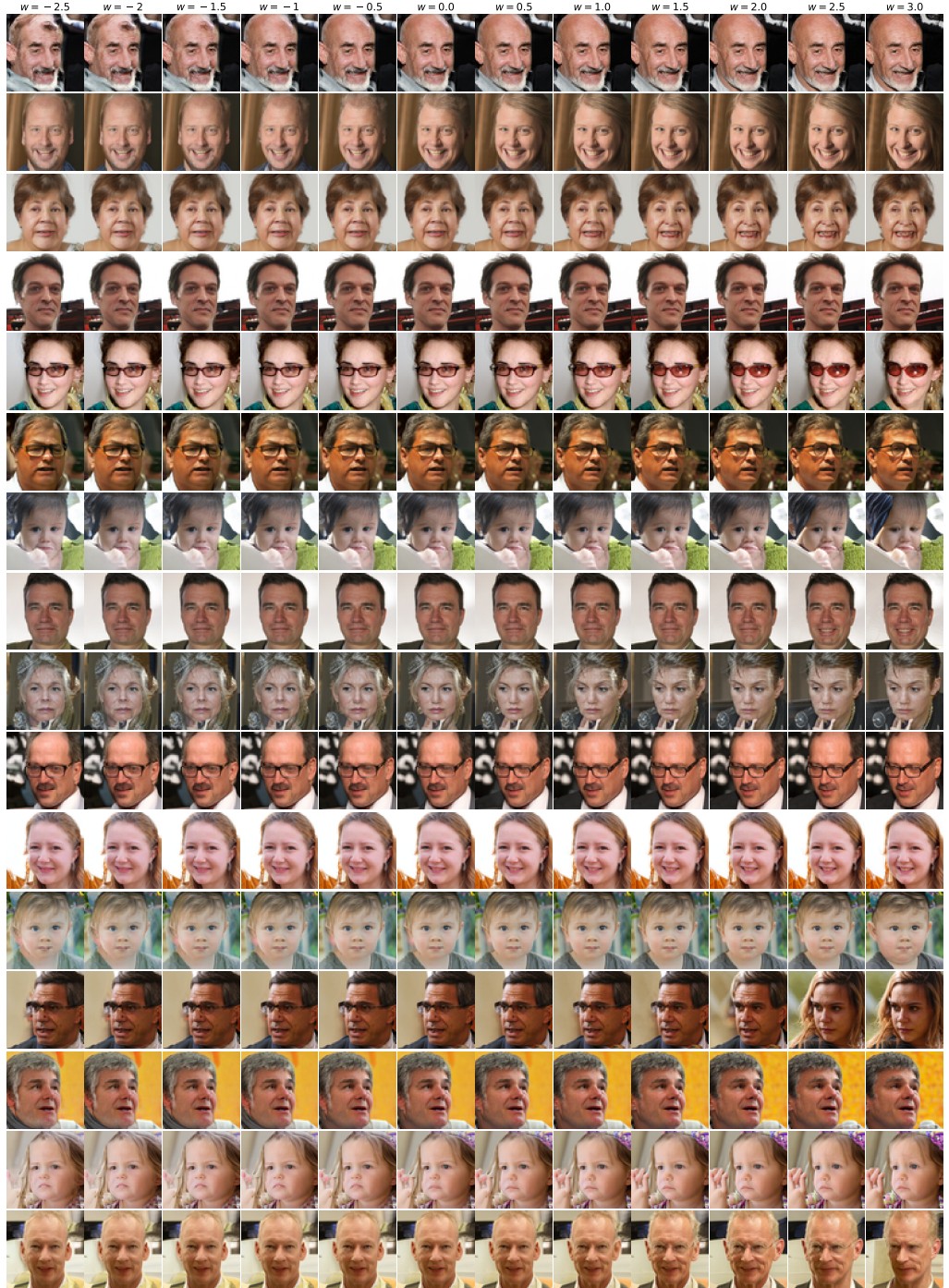

Figure J.1: Visual demonstration of Neon on EDM-VP (FFHQ-64) with merge formula $\theta_{\text{Neon}} = \theta_r - w(\theta_s - \theta_r)$, $\mathcal{B} = 1.5$ (Mi), $|S| = 18$k. Each column shows a different $w$ value: negative $w$ moves toward the degraded self-trained model ($w = -1$: FID = 3.73), $w = 0$ recovers the base model (FID = 2.39), and positive $w$ performs negative extrapolation with optimal results at $w = 2$ (FID = 1.12)

## K    VISUALIZING NEON'S EFFECT ON INDIVIDUAL SAMPLES

To provide intuition for how Neon operates at the level of individual training samples, we conduct a controlled experiment that isolates the effect of a single synthetic image on the model's output. By isolating the effect of a single synthetic sample, this visualization reveals how Neon's parameter-space extrapolation translates into coherent function-space changes, demonstrating that the method modifies semantically meaningful image features rather than introducing arbitrary noise.

**Experimental setup.**    We begin with a base model $\theta_r$ (VAR-d36) and generate a single synthetic sample $x_{\text{syn}}$ using a fixed random seed. This sample serves two roles: (i) as the sole training example for computing the synthetic gradient $r_s$, and (ii) as the reference image we will regenerate with different model weights to observe changes. Concretely, we:

1. Generate $x_{\text{syn}}$ from $\theta_r$ using seed $s = 999$ and CFG scale $\gamma = 1.5$

2. Compute the synthetic gradient $r_s(\theta_r) = \nabla_\theta \ell(x_{\text{syn}}; \theta)\big|_{\theta_r}$ averaged over 10 forward passes with different internal randomness (dropout, etc.)

3. Create Neon models: $\theta_{\text{Neon}}(w) = \theta_r + w \cdot \alpha \cdot r_s(\theta_r)$ for various merge weights $w \in \{0, 0.5, 1.0, 1.5, 2.0\}$, with step size $\alpha = 10^{-4}$

4. Regenerate the same image using each $\theta_{\text{Neon}}(w)$ with the identical seed $s = 999$

5. Compute the pixel-wise change: $\Delta f(w) = \|f(s; \theta_r) - f(s; \theta_{\text{Neon}}(w))\|_2$ where the norm is taken across RGB channels

Figure K.0 shows the results across diverse ImageNet classes. For each class, the top row displays the regenerated image $f(s; \theta_{\text{Neon}}(w))$ at each merge weight (at $w = 0$, this is identical to $x_{\text{syn}}$), while the bottom row shows the pixel-wise difference map $\|\Delta f(w)\|$ normalized to $[0, 1]$ (heatmap: black = no change, bright = large change). As $w$ increases, we observe semantically meaningful deviations from the original sample, confirming that Neon's extrapolation operates in a structured manner rather than introducing random perturbations, adding simple realistic details to the generated image.

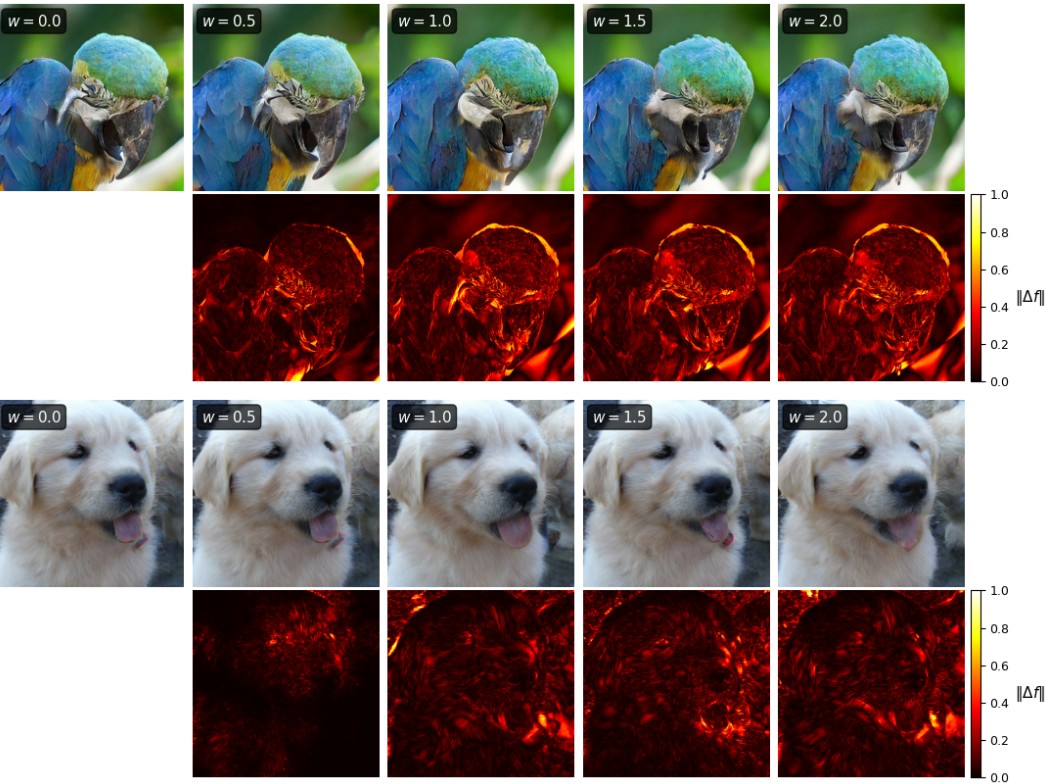

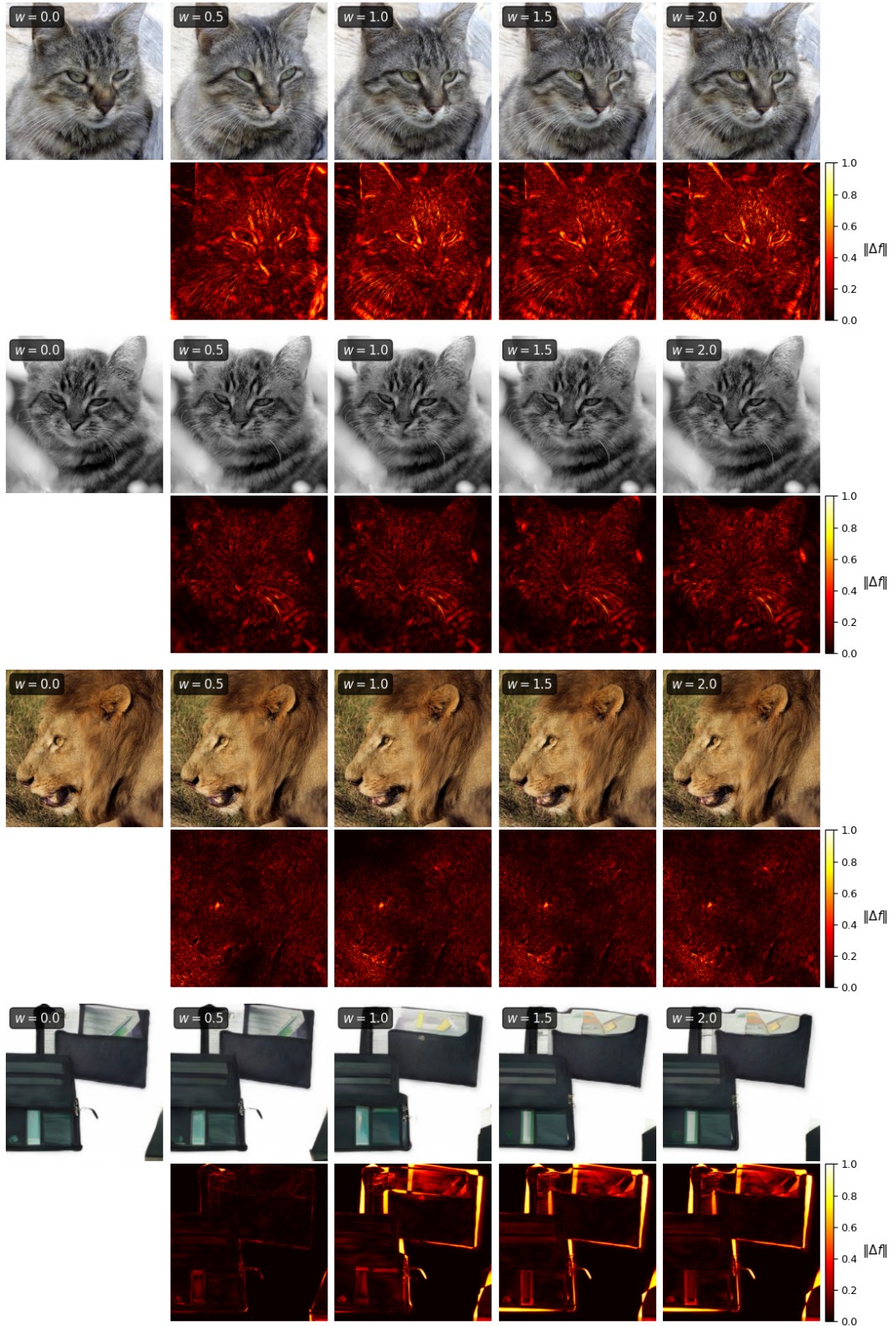

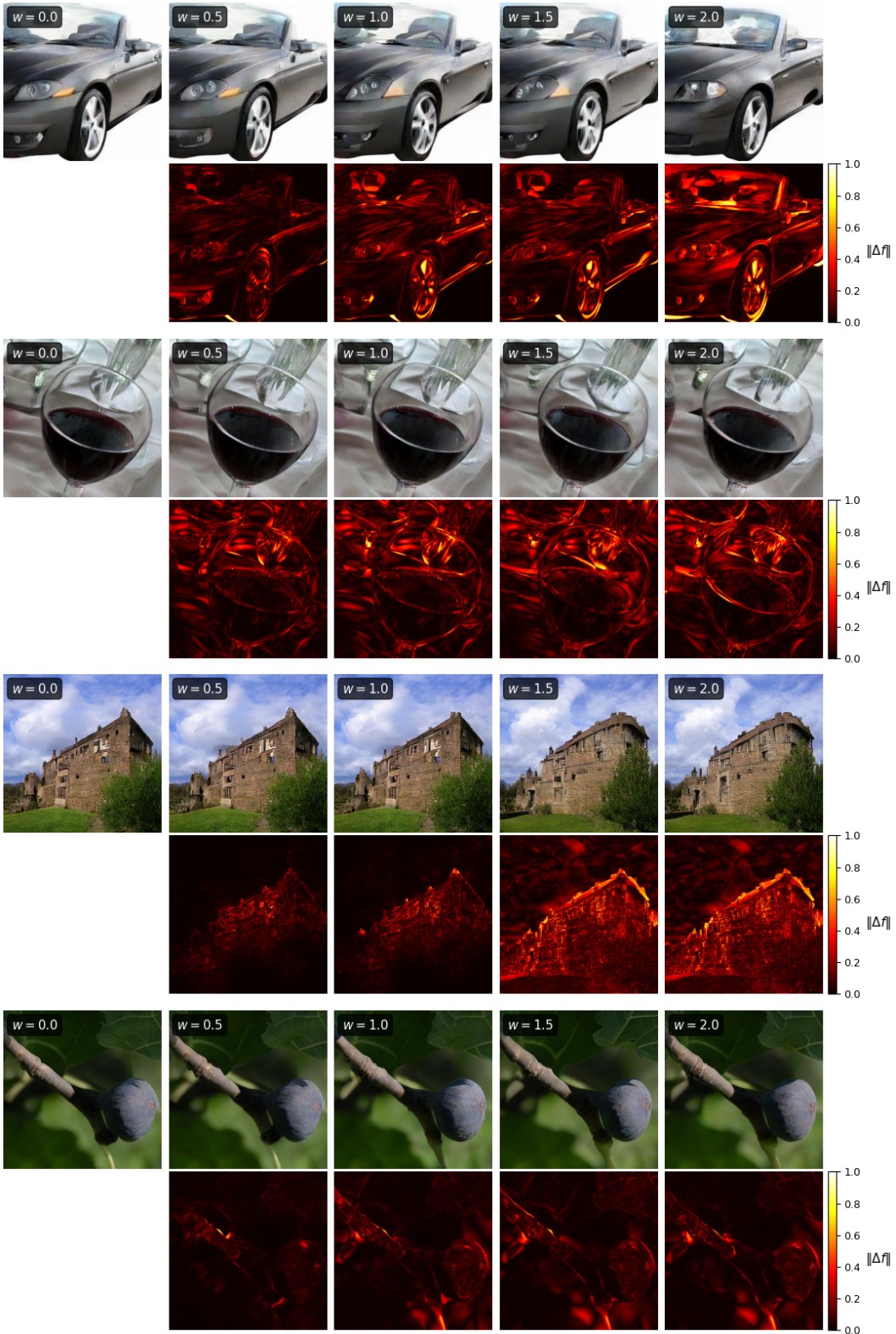

Figure K.0: Neon's effect on individual training samples across diverse ImageNet classes.

