# OpenReview forum: "Neon: Negative Extrapolation From Self-Training Improves Image Generation"
_ICLR.cc/2026/Conference — ICLR 2026 Oral_

### Official Review · Reviewer_BLA3 · 2025-10-21

**Soundness:** 3
**Presentation:** 3
**Contribution:** 3
**Rating:** 6
**Confidence:** 4

**Summary:**

The authors propose Neon, a self-training method that fine-tunes a base model using its own self-synthesized data with reversed gradient updates. Remarkably, Neon is effective even with as few as 1,000 synthetic samples. Its performance is validated across a wide range of architectures and datasets.

**Strengths:**

* Neon is simple to implement and effectively addresses common self-training challenges such as model autophagy disorder (MAD) and model collapse.

* The method is evaluated across a wide range of architectures—including diffusion, flow-matching, autoregressive, and momentum-matching models—and diverse datasets such as ImageNet, CIFAR, and FFHQ.

**Weaknesses:**

* The effectiveness of Neon hinges on the assumption that s < 0 (negative gradient alignment). However, this condition may not consistently hold in practice, raising concerns about the robustness of the approach.

* The hyperparameter configuration—particularly the training budget and negative extrapolation strength w —could introduce additional tuning complexity. Ablation studies in Figures 3 and 4 suggest that Neon is sensitive to these hyperparameters, which may impact its ease of deployment.

**Questions:**

* The key idea behind Neon—"synthetic degradation and real-data improvement point in opposite directions"—is intriguing. However, the current derivation of the supporting theorems appears loosely structured. Could you provide a clearer logical overview of how these derivations are organized? Additionally, how should this statement be interpreted intuitively in the context of model training?

* In Figure 3, the FID increases as the self-training budget B grows. Does this suggest that extended training with Neon may lead to performance degradation? If so, does it challenge the assumption that s < 0 (negative gradient alignment) consistently holds?

* In the images presented in Appendices E through I, are these generated samples from the Neon self-trained models, or are they the synthesized images used during the self-training process? Could you clarify this distinction and provide an analysis of the synthesized images used for self-training? Additionally, a visual interpretation explaining why these images are effective in contributing to the self-training process would be highly valuable.

---

> ### Author Response · Authors · 2025-11-16
> **Response to reviewer BLA3 - part 1**
>
> We thank the reviewer for their thoughtful feedback and for recognizing Neon's simplicity, effectiveness, and broad applicability.
>
> ---
>
> **Weakness 1: "The effectiveness of Neon hinges on the assumption that $s < 0$"**
>
> We respectfully clarify: **$s < 0$ is not an assumption. It is a proven consequence.** Our theory establishes that $s < 0$ emerges when (i) the sampler is mode-seeking (monotone reweighting), and (ii) the model error $\|\varepsilon\|_{H_d}$ is sufficiently small.
>
> Theorems 1 and 2, together with proofs in Appendices B.4–B.7, demonstrate that all standard inference procedures (such as temperature $\tau < 1$, top-$k$, top-$p$, and finite-step ODE solvers) satisfy the mode-seeking criterion, guaranteeing $\cos\varphi < 0$ and hence $s < 0$ near good models.
>
> The primary assumption is that $\theta_r = \theta^* + \varepsilon$ with small $\|\varepsilon\|_{H_d}$, where $\theta^*$ minimizes population risk. Practically, this requires the base model to achieve reasonable performance. Our ablation study "How good must the base model be?" (Section 4.4, Figure 9) shows Neon consistently improves performance even when base models are far from optimal, demonstrating robustness across quality levels.
>
> We validated Neon on nearly all available open-source generative models across diverse datasets (ImageNet, CIFAR-10, FFHQ) and architectures (diffusion, flow-matching, autoregressive, momentum-matching), providing strong empirical evidence that these theoretical conditions translate to practice.
>
> ---
>
> **Weakness 2: "Hyperparameter sensitivity may impact ease of deployment"**
>
> Neon exhibits **robust self-improvement across broad hyperparameter ranges** for both training budget $\mathcal{B}$ and merge weight $w$. Figures 3, 4, and 5 show consistent improvement over wide regions. Finding a configuration that improves over the base model requires only modest exploration.
>
> In practice, moderate training budgets (1–2% of base training) and merge weights ($w \in [0.5, 1.5]$) work well across settings. The U-shaped curves in Figure 3 are theoretically justified (Appendix B.10) and empirically predictable. The unimodal relationship between hyperparameters and FID enables efficient search using standard optimization methods.

---

> > ### Author Response · Authors · 2025-11-16
> > **Response to reviewer BLA3 - part 2**
> >
> > **Question 1: "Could you provide a clearer logical overview of how derivations are organized?"**
> >
> > We substantially revised the manuscript to improve theoretical presentation:
> >
> > 1.  **Restructured main paper (Section 3.1)**: Reorganized "Why Neon Works" with improved transitions and clearer connections between intuition and formalism.
> >
> > 2.  **Added Gaussian warmup (new Appendix B.1)**: We introduced a **fully worked example of Gaussian estimation** demonstrating Neon's mechanism with complete analytical calculations. We estimate mean $\mu$ and covariance $\Sigma$ of a $D$-dimensional Gaussian $\mathcal{N}(\mu, \Sigma)$ from $n$ finite samples, generate synthetic data from a mode-seeking sampler (shrunken covariance $\tau^2\Sigma$ with $\tau < 1$), perform an infinitesimal gradient step, and apply the Neon merge. Fine-tuning on mode-seeking synthetic samples contracts the distribution (moving $\propto \hat{\Sigma}^{-1}$); Neon's negative extrapolation reverses this contraction, correcting finite-sample MLE bias. We prove this reduces population negative log-likelihood using classical Wishart properties. This uses identical mathematical structure as our general theory (gradient alignment, Taylor expansion, optimal merge weight) but with closed-form Gaussian calculations. **This demonstrates Neon can improve MLE estimation through line search with no additional assumptions beyond mode-seeking sampling**.
> >
> > 3.  **Added proof roadmap (Appendix B.2)**: Created a detailed roadmap outlining the five-stage logical structure, explaining how theorems build toward establishing Neon's mechanism.
> >
> > 4.  **Improved proof flow**: Added linking explanations between proofs to enhance readability.
> >
> > Regarding "synthetic degradation and real-data improvement point in opposite directions": Neon is a **post-processing method** operating independently of training. We require only that $\theta_r$ lies near the population optimum $\theta^*$. The statement refers to the geometric relationship in parameter space: the synthetic gradient $r_s$ (capturing bias toward high-probability outputs) points opposite to the true data gradient $r_d$ (pointing toward the population optimum). This anti-alignment arises from mode-seeking inference, not the training algorithm's properties.
> >
> > ---
> >
> > **Question 2: "In Figure 3, does increasing FID at large $\mathcal{B}$ challenge the assumption $s < 0$?"**
> >
> > As noted, **$s < 0$ is proven, not assumed**. FID degradation at very large budgets reflects the breakdown of our **local Taylor expansion**, not a contradiction of $s < 0$.
> >
> > Our one-step guarantee (Theorem B.1) is valid only when the displacement $w\mathcal{B} P r_s$ remains within the local neighborhood where our linear approximation holds.
> >
> > When $\mathcal{B} = \alpha T$ becomes too large, the Neon extrapolation $\theta_{\text{Neon}} \approx \theta_r + w\mathcal{B} P r_s$ exits this local region, and higher-order terms $O((w\mathcal{B})^3)$ dominate, causing the observed degradation.
> >
> > The optimal region (Figure 3) occurs at a moderate $\mathcal{B}$ that balances two factors: the budget must be large enough to reliably estimate the degradation direction (low variance) but small enough to remain within the local validity region (low bias).
> >
> > A key benefit of **short** fine-tuning is this bias-variance trade-off. Reusing synthetic examples with fresh random draws reduces Monte Carlo variance as $1/T$, while the curvature bias scales with $\mathcal{O}(\alpha T)$. This is formalized in **Appendix B.10**, which provides the theoretical justification for the U-shaped curve and our practical hyperparameter guidance.

---

> > > ### Author Response · Authors · 2025-11-16
> > > **Response to reviewer BLA3 - part 3**
> > >
> > > **Question 3: "Are images in Appendices E–I from Neon models or synthetic samples?"**
> > >
> > > The images are **generated by the final Neon models** after applying our method, demonstrating output quality.
> > >
> > > Regarding individual image contributions: we cannot easily attribute effects to single images because (i) Neon operates on the **distribution** of synthetic data, and (ii) parameter updates depend on complex interactions between the image distribution, model architecture, and optimizer conditioning.
> > >
> > > However, we added **Appendix K: Visualizing Neon's effect on individual samples**. This controlled experiment isolates single-sample effects by computing the gradient $r_s(\theta_r) = \nabla_\theta \ell(x_{\text{syn}}; \theta)|_{\theta_r}$ from a single image, creating Neon models $\theta_N (w) = \theta_r + w\alpha r_s$ for various $w$, regenerating images with identical seeds, and visualizing the pixel-wise differences as heatmaps.
> > >
> > > Figure K.0 shows Neon's extrapolation produces **semantically meaningful changes** (such as modifications to texture, lighting, and details) rather than random noise. Changes concentrate where synthetic samples exhibit characteristic features, confirming principled operation on structured representations. This provides direct visual evidence that the method modifies semantically meaningful features.
> > >
> > > Additionally, we added quantitative grid visualizations in other appendix sections (e.g., App. H, I, J) showing outputs across different merge weights $w$, allowing direct visual comparison of how the method transforms model outputs.
> > >
> > > ---
> > >
> > > We hope these clarifications address the reviewer's concerns. We have significantly strengthened the theoretical presentation through reorganization, the addition of an accessible Gaussian warmup example with complete analytical derivations, and improved pedagogical flow. The empirical validation demonstrates that our theoretical conditions hold in practice across a wide range of model qualities and configurations. The new single-sample visualization and grid comparisons provide concrete intuition for Neon's mechanism at both individual and distributional levels.

---

> > > > ### Comment · Reviewer_BLA3 · 2025-11-17
> > > >
> > > > Thank you for the responses and updated paper writing. I will raise the score.

---

> > > > > ### Author Response · Authors · 2025-11-26
> > > > > **Thank you**
> > > > >
> > > > > Dear reviewer BLA3,
> > > > >
> > > > > Thank you for engaging with our response and for updating your score. We appreciate your time and valuable feedback.

---

### Official Review · Reviewer_Srik · 2025-10-28

**Soundness:** 3
**Presentation:** 4
**Contribution:** 3
**Rating:** 8
**Confidence:** 4

**Summary:**

The paper proposes NEON (Negative Extrapolation from self-traiNing): briefly self-train a generative model on its own samples to obtain degraded weights , then extrapolate backward from the degradation direction via a simple parameter merge . The authors give a theoretical account showing that common mode-seeking samplers create anti-alignment between synthetic and population gradients, so reversing the self-training direction reduces true data risk. Empirically, NEON improves diffusion, flow-matching, autoregressive, and few-step models on CIFAR-10, FFHQ, and ImageNet, often with <1% extra compute; notably it pushes xAR-L on ImageNet-256 from FID 1.28 → 1.02 with only 0.36% additional training compute.

**Strengths:**

Simple, general, post-hoc procedure that requires no extra real data, no auxiliary models, and no inference changes; just a short self-training run and a weight merge.

Clear theory: formalizes sampler-induced anti-alignment and gives conditions where negative extrapolation lowers population risk; also analyzes failure cases (diversity-seeking samplers).

Broad empirical coverage across diffusion, flow matching, AR, and few-step models with consistent FID gains; includes precision/recall analysis explaining NEON’s recall-boosting mechanism.

Strong results with tiny cost (often ≤2% of base training compute; sometimes as low as 0.36%), and improvements with as few as 1k synthetic samples.

SOTA highlight: ImageNet-256 xAR-L FID 1.02, plus useful studies on (w, γ) co-tuning and cross-architecture transfer of the degradation signal.

**Weaknesses:**

Positioning vs. simple weight merges: needs stronger comparisons to generic weight interpolation/extrapolation baselines (e.g., linear checkpoint merges/SWA-style extrapolation) to isolate NEON’s specific benefit beyond “negative LR step”.

Benchmark scope: focuses on standard class-conditional/unconditional image generation; lacks large-scale text-to-image or broader modalities, and relies mainly on FID + P/R without human eval.

Hyperparameter sensitivity: performance depends on w (and γ for AR/CFG); while grids are shown, guidance on automatic selection or stability across training checkpoints could be expanded.

**Questions:**

see weakness

---

> ### Author Response · Authors · 2025-11-16
> **Response to reviewer Srik**
>
> We sincerely thank the reviewer for the positive evaluation and thoughtful feedback. We are delighted that the reviewer found Neon simple, general, and empirically strong across diverse settings. We address each point below.
>
> ---
>
> ### Weakness 1: Positioning vs. simple weight merges
>
> We appreciate this important question about Neon's distinction from generic weight merging.
>
> **Comparison with model soups / weight averaging:** Model soups (Wortsman et al., NeurIPS 2022) interpolate between independently trained models: $\theta_{\mathrm{soup}} = (1-\lambda)\theta_1 + \lambda\theta_2$ with $\lambda \in [0,1]$. Neon performs *extrapolation*: $\theta_{\mathrm{Neon}} = (1+w)\theta_r - w\theta_s$ with $w > 0$, using a specific synthetic degradation signal where $\theta_s$ is deliberately degraded via self-training.
>
> **Comparison with SWA-style extrapolation:** SWA extrapolates along the optimization trajectory (SGD dynamics on real data). Neon is a **post-processing method** that works independently of how the base model was trained—it extrapolates along the *synthetic degradation direction* determined by mode-seeking sampler behavior.
>
> **What makes Neon's direction special:** Our theoretical contribution is proving that:
> 1.  Standard mode-seeking samplers induce anti-alignment ($s < 0$)
> 2.  This makes the synthetic gradient $r_s$ a reliable negative signal
> 3.  The optimal extrapolation weight $w$ can be predicted from geometry (Theorem 1)
>
> Generic checkpoint extrapolation lacks these guarantees.
>
> **Empirical evidence:** Cross-architecture transfer (Figure 9) shows synthetic data from one model improves a *different* architecture, impossible with standard checkpoint averaging or SWA-style methods.
>
> ---
>
> ### Weakness 2: Benchmark scope
>
> We agree that extending to large-scale text-to-image and broader modalities would strengthen the work. Our theoretical framework makes no assumptions specific to images. Experiments on autoregressive models (VAR, xAR) use the same next-token prediction objective as language models, suggesting natural extensibility. We have added extensive qualitative visualizations (Appendices H, I, J, K) to complement our quantitative metrics and provide visual evidence of the improvements.
>
> ---
>
> ### Weakness 3: Hyperparameter sensitivity
>
> We appreciate this practical concern. Hyperparameter selection is natural for any ML algorithm, and we find Neon is robust to wide ranges of hyperparameter choices—Figures 3-4 show broad regions of consistent improvement. Our VAR-d16 experiments demonstrate that optimal $w$ remains stable in $[0.8, 1.0]$ even as training budgets vary from 0.5M to 6M images. In practice, $w \in [0.5, 1.5]$ and budget = 1-2% of base training work well across diverse settings, making deployment straightforward.
>
> ---
>
> We are grateful for the positive assessment and hope these clarifications address the reviewer's questions. Please let us know if any additional details would be helpful.
>
> ---
>
> **References:**
>
> Izmailov P, Podoprikhin D, Garipov T, Vetrov D, Wilson AG. Averaging weights leads to wider optima and better generalization. *Proceedings of the 34th Conference on Uncertainty in Artificial Intelligence*; 2018.
>
> Wortsman M, Ilharco G, Gadre SY, Roelofs R, Gontijo-Lopes R, Morcos AS, Namkoong H, Farhadi A, Carmon Y, Kornblith S, Schmidt L. Model soups: averaging weights of multiple fine-tuned models improves accuracy without increasing inference time. *Proceedings of the 39th International Conference on Machine Learning*; 2022.
>
> ---

---

> > ### Author Response · Authors · 2025-11-26
> > **Follow-up on Rebuttal**
> >
> > Dear reviewer Srik,
> >
> > We wanted to kindly follow up on our rebuttal. If you have any feedback or further questions, we would be happy to address them.

---

### Official Review · Reviewer_YZmD · 2025-10-31

**Soundness:** 2
**Presentation:** 3
**Contribution:** 2
**Rating:** 6
**Confidence:** 5

**Summary:**

Usually training/retraining generative models on synthetic data can degrade performance metrics. On the opposite, this paper proposed a technique to leverage (bad) synthetic data from generative models, and improve generation. This work is in the direct line of work following Karras 2024 and Alemohammad 2024.

**Strengths:**

The paper is well-written, and the idea is clearly explained. Experiments are overall rather well presented. Figure 2 explains well the approach.

**Weaknesses:**

- I do not understand the novelty with respect to previous works.
Could authors explain what is the difference between their work and [1] and mostly [2]? Especially, I really would like to better understand better the difference with [2]: the idea of using negative guidance from synthetic data is already in [2], in particular, Equation (1) of the proposed manuscript resembles line 4 in algorithm 1 of [2]. Are you doing SIMS, but in the parameter space? Could authors comment on that?
- How significant do you consider the empirical results? Can you show the same plots on test set, with the Dinov2 embedding?
- in particular I would be interested to see if the minimum value on the training set correlates with the minimum value on the test set


[1] Tero Karras, Miika Aittala, Tuomas Kynkäänniemi, Jaakko Lehtinen, Timo Aila, and Samuli Laine.
Guiding a diffusion model with a bad version of itself.

[2] Sina Alemohammad, Ahmed Imtiaz Humayun, Shruti Agarwal, John Collomosse, and Richard Baraniuk. Self-improving diffusion models with synthetic data


Non-scientific comment: I would remove the following quote, "In the words of Martin Luther King, Jr., “Sometimes to move forward, we have to go backward.”, used to motivate negative parameter guidance. It does not feel appropriate

**Questions:**

see weaknesses

---

> ### Author Response · Authors · 2025-11-16
> **Response to reviewer YZmD - part 1**
>
> We sincerely thank the reviewer for the thorough evaluation and valuable feedback. We address each concern below.
>
> ---
>
> **Novelty with respect to SIMS [2] and Karras et al. [1]**
>
> Both Karras et al. [1] and SIMS [2] can be written as the same inference-time mixing rule on the score/velocity:
> $$
> \tilde{s}(x_t,t) = (1+w)s_{\theta_r}(x_t,t) - ws_{\theta_s}(x_t,t),\qquad w\ge 0,
> $$
> where $s_{\theta_r}$ is the base model and $s_{\theta_s}$ is a deliberately worse model. Karras et al. instantiate $s_{\theta_s}$ by under-training (a less-capable checkpoint), while SIMS instantiates $s_{\theta_s}$ by self-training on synthetic data, which is known to degrade generators via model collapse/MAD. Both report empirical gains from subtracting a worse model, but neither provides a theory predicting when the sign and magnitude help or hurt, nor do they analyze how sampler behavior (mode- vs. diversity-seeking) governs the outcome.
>
> **What Neon changes.** First, Neon provides a principled analysis distinguishing mode-seeking from diversity-seeking samplers and proves the sign must flip between these regimes (Theorem 1). This explains why subtracting a worse model can help—and when it must fail. For example, if one constructs a "worse" guidance model by fine-tuning on diversity-seeking synthetic data, it is still worse overall—yet negative guidance would not help; our sign analysis predicts this, whereas Karras/SIMS do not.
>
> Second, Karras/SIMS operate at sampling time, require sampler modifications, and roughly double NFEs. **Neon performs a one-shot parameter-space merge**, $\theta_{\mathrm{Neon}}=(1+w)\theta_r - w\theta_s$, yielding a single improved model with zero sampling overhead and no sampler changes. In practice this is more data/compute efficient (e.g., thousands vs. hundreds of thousands of synthetic samples; ~0.36% extra compute) and aligns with deployment constraints where added inference cost is undesirable. The simplicity of the final update does not diminish the depth of the theory that determines its correct sign and guarantees.
>
> **Why SIMS "worked," and when it wouldn't.** Our analysis shows that widely used finite-step samplers are mode-seeking. SIMS happened to pair its negative signal with such samplers; thus, subtracting the self-trained (degraded) model improved fidelity—precisely for the reason our theory predicts. Conversely, if one instead made the negative model "worse" via diversity-seeking synthetic fine-tuning, the same negative guidance would fail—the sign is wrong for mode-seeking samplers. This separates "being worse" from "being useful as negative guidance," a distinction Neon formalizes and prior work lacks.
>
> To summarize, Neon is not just SIMS in parameter space. A local Taylor expansion can relate the forms, but that view ignores Neon's core contributions: a sign-correct, sampler-aware theory that predicts success and failure, and an offline, data/compute-efficient update with no inference overhead. SIMS is best seen as a heuristic, diffusion-specific approximation to the principled effect Neon attains. As for Karras et al., both Karras and SIMS share the intuition "use a bad model as negative guidance," but remain empirical heuristics. Neon predicts when negative guidance helps (mode-seeking) versus hurts (diversity-seeking), and achieves the benefit without inference-time overhead. Any link to Karras is indirect via the SIMS-style form; conceptually, Neon stands on its own theoretical foundation.
>
> **References:**
>
>
> [1] Karras et al., "Guiding a diffusion model with a bad version of itself," 2024.
>
> [2] Alemohammad et al., "Self-improving diffusion models with synthetic data," 2024.

---

> > ### Author Response · Authors · 2025-11-16
> > **Response to reviewer YZmD - part 2**
> >
> > **Test set validation with DINOv2 embeddings**
> >
> > We conducted additional experiments on ImageNet 256×256 using VAR-d16 to validate generalization. Critically, we selected all hyperparameters $(w,\gamma)$ using only the training-set FID ($\gamma$ denotes the classifier-free guidance scale), then evaluated on a held-out test set using both Inception and DINOv2 feature spaces, ensuring no test leakage.
> >
> > | Budget (Mi) |  w  |  γ  | FID Train | FID Test (Inception) | FID Test (DINOv2) |
> > | :---------: | :-: | :-: | :-------: | :------------------: | :---------------: |
> > |     0.00    | 0.0 | 1.5 |    3.30   |         4.48         |       69.60       |
> > |     0.50    | 1.0 | 2.6 |    2.28   |         3.07         |       63.32       |
> > |     1.00    | 0.9 | 2.5 |    2.13   |         2.67         |       62.99       |
> > |     1.50    | 0.8 | 2.4 |    2.12   |         2.72         |     **61.79** |
> > |     2.00    | 0.9 | 2.7 |  **2.07** |         2.60         |       62.28       |
> > |     3.00    | 0.9 | 2.8 |    2.08   |       **2.56** |       64.83       |
> > |     4.00    | 0.8 | 2.7 |    2.13   |         2.75         |       63.12       |
> > |     5.00    | 0.8 | 2.8 |    2.16   |         2.73         |       64.87       |
> > |     6.00    | 0.8 | 2.9 |    2.18   |         2.77         |       66.85       |
> >
> > *Table:* Test-set validation on ImageNet 256×256 using VAR-d16. Row 1 shows the baseline ($w=0$, no Neon). Budget $B$ is in millions of training images. $(w,\gamma)$ were selected on the training set only.
> >
> > **Train–test correlation.** The training-set minimum occurs at $B=2.00$M (FID 2.07), while the test-set minima occur at $B=3.00$M (Inception: 2.56) and $B=1.50$M (DINOv2: 61.79)—all within a narrow 1.5–3.0M band. Across budgets, **Spearman $\rho(\mathrm{train}, \mathrm{test}-\mathrm{Inception}) = 0.929$** and **$\rho(\mathrm{train}, \mathrm{test}-\mathrm{DINOv2}) = 0.711$**, confirming that training-set optimization reliably predicts test performance.
> >
> > **Consistent improvements without overfitting.** Every Neon configuration outperforms the baseline on the test set for both metrics. Test Inception-FID improves from 4.48 to **2.56** at $B=3.0$M (−42.9%), with early gains at small budgets (3.07 at $B=0.5$M). Test DINOv2-FID improves from 69.60 to **61.79** at $B=1.5$M (−11.2%). Hyperparameters are stable across budgets ($w\in[0.8,1.0]$, $\gamma\in[2.4,2.9]$), making deployment practical without extensive re-tuning.
> >
> > ---
> >
> > **Regarding the quotation**
> >
> > We appreciate the comment. To keep the manuscript focused on technical content, we have removed the quotation from the introduction. We retain a single, non-quotational sentence at the end of the conclusion to summarize the core insight.
> >
> > ---
> >
> > We hope these responses address the reviewer's concerns. Regarding the test set validation table, please let us know if the reviewer would like us to include it in the appendix of the revised manuscript, or if this analysis was primarily for confirming results during the review process. We are happy to provide additional clarifications or experiments as needed.

---

> > > ### Author Response · Authors · 2025-11-26
> > > **Follow-up on Rebuttal**
> > >
> > > Dear reviewer YZmD
> > >
> > > We wanted to kindly follow up on our rebuttal. If you have any feedback or would like us to conduct additional experiments or make any changes, we would be happy to do so. We simply want to ensure we have sufficient time to address any remaining concerns and update the manuscript accordingly.
> > > Thank you for your consideration.

---

### Official Review · Reviewer_dZE2 · 2025-11-01

**Soundness:** 3
**Presentation:** 4
**Contribution:** 3
**Rating:** 8
**Confidence:** 3

**Summary:**

In this paper, the author proposed a new self-training algorithm for image generative models NEON. Specifically, NEON uses synthetic images generated by the generative model to finetune the model, and use the performance degredation as a learning signal. The key insight is that most inference samplers are mode-seeking, biasing samples towards high-density regions of the model distribution, resulting in model collapasing, and worsens FID. Consequently, NEON uses negative extrapolation between the reference model and briefly self-trained model, avoids mode collapsing. The author tested their NEON algorithm across different generative models and datasets and shows consistent FID improvement with little computational overhead.

**Strengths:**

Overall, the paper introduces Neon, a simple but effective method that interprets self-training degradation as a useful signal for improvement. The approach is theoretically grounded and empirically validated across diffusion, flow, and autoregressive models.

- The core idea of reversing the direction of self-training degradation is both simple and effective. The implementation requires only a single weight extrapolation step and no architectural or loss modifications, making the method easy to implement and apply.
- Neon adds little computational overhead and scales well across model families. Its simplicity makes it readily to deploy in large-scale training pipelines.
- The experiments are thorough, spanning multiple datasets (CIFAR-10, FFHQ, ImageNet-256/512) and model architectures. Across all cases, Neon consistently improves FID and recall. The recall and precision analysis provided nice support to the theory of Neon. It treats precision for recall, encouraging diveristy by negative extropolation.
- Ablations show that Neon’s benefits generalize across architectures and even compensate for reduced real data availability.
- The controlled toy experiment from the appendix effectively illustrate the theoretical predictions and help readers grasp the distinction between mode-seeking and diversity-seeking regimes.

I did not verify all derivations in detail, but at an intuitive level the theoretical claims are consistent and well-motivated. I defer to other reviewers for a closer assessment of the mathematical rigor and proofs.

**Weaknesses:**

- The paper does not include quantitative comparisons against other self-training baselinesin the precision/recall/FID analyses. Even showing results for the simplest baseline of direct self-finetuning would help clarify how much of Neon’s gain comes from the negative extrapolation itself versus the self-training process. Moreover, comparisons in terms of data and compute efficiency would contextualize Neon’s benefits relative to prior self-improvement algorithms.
- While the quantitative results are compelling, the paper would benefit from a few qualitative visual examples to illustrate how Neon changes the generative behavior. It remains a bit unclear whether the performance gains are purely distributional (i.e., improved diversity and recall) or also reflect higher fidelity and perceptual realism compared to naive self-training.

**Questions:**

- Can the authors provide qualitative visual examples to illustrate how Neon changes the generative distribution? Are the observed gains primarily due to improved diversity (recall), or do they also enhance perceptual fidelity compared to naïve self-training—or possibly hurt fidelity due to the diversity–precision trade-off?
- Can the author also include some baselines for the precsion/recall/FID analysis?
- Do the authors see Neon as applicable to other domains such as NLP generation?
- For figure 9, what is the synhetic dataset used samped from for the 30k model?

---

> ### Author Response · Authors · 2025-11-16
> **Response to reviewer dZE2 - part 1**
>
> We sincerely thank the reviewer for the enthusiastic support and thoughtful questions. We are delighted that the reviewer found Neon simple, effective, and well-validated. We address each point below.
>
> ---
>
> **Quantitative comparisons against self-training baselines**
>
> In the revised manuscript, we have added quantitative comparisons between direct self-training (the simplest baseline) and Neon across multiple settings:
>
> **Main paper:** Figure 4 plots FID, precision, and recall as functions of merge weight $w$ for EDM-VP on CIFAR-10. Crucially, we extended the $w$ range to include $w=-1$, which corresponds to direct self-training (i.e., $\theta_{\mathrm{Neon}} = \theta_s$, using the self-trained model directly without correction). The results show that self-training ($w=-1$) consistently degrades performance compared to the base model ($w=0$), with substantially worse FID, reduced recall and precision. As $w$ increases from $-1$ to positive values, Neon progressively improves the model, achieving optimal FID at moderate positive $w$ while trading precision for recall—consistent with our theoretical predictions.
>
> **Appendix:** Figure D.1 provides the same analysis for EDM-VP on FFHQ-64, again showing that $w=-1$ (direct self-training) degrades performance, while positive $w$ (Neon's negative extrapolation) produces substantial improvements.
>
> Self-training has been extensively analyzed in recent work, with consistent findings that training directly on synthetic data causes quality (precision) or diversity (recall) to progressively decrease [Shumailov et al., Nature 2024; Alemohammad et al., ICLR 2024]. Our results are fully consistent with this line of research, confirming that the $w=-1$ case reproduces known model autophagy disorder (MAD) and distributional collapse patterns.
>
> ---
>
> **Data and compute efficiency comparisons**
>
> We appreciate the reviewer's interest in understanding Neon's efficiency advantages. We have conducted detailed comparisons with the two most relevant self-improvement works on the same model (EDM) and dataset (CIFAR-10):
>
> **Comparison with SIMS (Alemohammad et al., 2024):**
>
> 1.  Test-time inference compute: SIMS requires 2× number of function evaluations (NFE) compared to Neon. Specifically, SIMS uses 70 NFE while Neon maintains the base model's 35 NFE, making Neon significantly more sampling-efficient.
>
> 2.  Training compute: SIMS requires approximately 20% of the original pretraining compute to achieve optimal results, while Neon needs only 1.75% of pretraining compute—a ~11× reduction in training overhead.
>
> 3.  Synthetic dataset size: SIMS uses |S| = 100k synthetic samples, while Neon achieves comparable or better results with only 6k datapoints—a ~17× reduction in data requirements.
>
> **Comparison with Direct Discriminative Optimization (DDO) (Zheng et al., 2025):**
>
> For the class-conditional EDM on CIFAR-10:
> * DDO requires 0.75% of pretraining compute per round × 16 rounds = 12% total pretraining compute
> * DDO uses 50k synthetic samples per round × 16 rounds = 800k total synthetic samples
> * Neon requires only 1.75% of pretraining compute in total—a ~7× reduction in training overhead
> * Neon uses only 6k synthetic samples in total—a ~133× reduction in data requirements
>
> These comparisons on identical experimental settings (same model architecture and dataset) demonstrate that Neon is significantly more efficient than prior self-improvement approaches in terms of training compute, synthetic data requirements, and inference cost, while achieving competitive or superior FID scores (Neon: 1.38 vs SIMS: 1.33 vs DDO: 1.30 on conditional CIFAR-10).
>
> If the reviewer has other specific self-improvement algorithms in mind for comparison, we would be happy to include additional analyses.

---

> > ### Author Response · Authors · 2025-11-16
> > **Response to reviewer dZE2 - part 2**
> >
> > **Qualitative visual examples**
> >
> > We have added extensive qualitative visualizations throughout the revised appendices to illustrate how Neon changes generative behavior at both individual and distributional levels:
> >
> > 1.  **Appendix J (EDM-VP on FFHQ-64):** Figure J.1 shows a grid of images generated across different merge weights $w \in [-1, -0.5, 0, 0.5, 1, 1.5, 2, 2.5, 3]$ using identical random seeds. Each column corresponds to a specific $w$ value, with $w=-1$ showing direct self-training outputs and positive $w$ showing Neon-corrected samples. Visual inspection reveals that self-training ($w=-1$) produces mode-collapsed outputs with reduced diversity, while Neon progressively recovers and enhances sample variety and realism.
> >
> > 2.  **Appendix H (IMM on ImageNet-256):** Figures H.2–H.7 provide systematic 2D grids exploring the joint effect of merge weight $w$ and classifier-free guidance scale $\gamma$ across diverse ImageNet classes. Each grid shows 2×2 random samples for each $(w, \gamma)$ configuration. These visualizations demonstrate that:
> >     * Self-training regions ($w < 0$) produce degraded outputs with artifacts
> >     * Neon regions ($w > 0$) generate perceptually realistic, diverse samples
> >     * The improvements are consistent across semantically distinct classes
> >
> > 3.  **Appendix I (VAR-d36-s on ImageNet-512):** Figures I.2–I.7 provide similar 2×2 grids across different classes at higher resolution, confirming that Neon's benefits scale to larger models and resolutions.
> >
> > 4.  **Appendix K (Single-sample visualization):** Figure K.1 isolates the effect of Neon on individual samples by regenerating the same image (fixed seed) across different merge weights $w \in [0, 0.5, 1.0, 1.5, 2.0]$. For each of 10 ImageNet classes, we show the regenerated image (top row) and pixel-wise difference heatmap (bottom row), revealing how Neon modifies individual outputs.
> >
> > ---
> >
> > **Are gains distributional (diversity) or perceptual (fidelity)?**
> >
> > The reviewer raises an excellent question. Our analysis shows that Neon improves both diversity and perceptual fidelity, but through a specific mechanism. We added a self-contained Gaussian estimation example in Appendix B.1 to clarify this point. Consider estimating the mean $\mu$ and covariance $\Sigma$ of a $D$-dimensional Gaussian $p_*(x) = \mathcal{N}(\mu_*, \Sigma_*)$ from $n$ finite samples. The maximum likelihood estimates are $\hat{\mu}_r$ and $\hat{\Sigma}_r$. When we apply Neon with mode-seeking synthetic data (generated from $q(x) = \mathcal{N}(\hat{\mu}_r, \tau^2 \hat{\Sigma}_r)$ with $\tau < 1$), the Neon update that reduces the population loss becomes:
> >
> > $\mu^{\rm Neon} = \hat{\mu}_r$ and $\Sigma^{\rm Neon} = \hat{\Sigma}_r + w \hat{\Sigma}_r^{-1}$
> >
> >
> > Neon corrects the covariance, not the mean. This means the distribution does not change where the highest-probability data (highest perceived quality) lies according to the model. Instead, Neon improves sampling in regions away from the mode—i.e., less probable regions under the model now generate higher-quality outputs. Neon does not increase the maximum quality a model can generate, but increases the fraction of outputs that are high-quality above a perceptual threshold. This theoretical prediction aligns with our qualitative observations in **Appendices H, I, and J**. These grids show that as we increase the strength of $w$, images that have lower quality or fewer details undergo more significant, corrective changes, while images that already have high quality remain largely stable. This effect is also visualized at the pixel level in Appendix K.
> >
> > ---
> >
> > **Applicability to NLP generation**
> >
> > Yes, we believe Neon is applicable to NLP and other generative domains. Our theory is general and makes no assumptions specific to images:
> > * The loss function is the standard negative log-likelihood (NLL) for next-token prediction in language models
> > * Mode-seeking sampling (temperature $\tau < 1$, top-$k$, top-$p$) is standard in NLP
> > * Visual autoregressive models (xAR, VAR) already share identical loss and sampling structures with NLP tasks, differing only in tokenization (visual vs. text tokens)
> >
> > Our experiments on autoregressive image models (Section 4.2) demonstrate that Neon works seamlessly with next-token prediction objectives. Extending to language models is conceptually straightforward and requires no modifications to the method. We leave this as future work on this line of research.

---

> > > ### Author Response · Authors · 2025-11-16
> > > **Response to reviewer dZE2 - part 3**
> > >
> > > **Figure 9 clarification**
> > >
> > > We thank the reviewer for bringing this to our attention. In all experiments shown in Figure 9, we used $|\mathcal{S}| = 6$k synthetic samples. We have added this information in the main manuscript.
> > >
> > > ---
> > >
> > > We hope these clarifications and additions address the reviewer's questions. We are grateful for the positive evaluation and have substantially strengthened the manuscript with extensive qualitative visualizations, explicit self-training baselines, the Gaussian warmup example clarifying the mechanism of improvement, and detailed efficiency comparisons. Please let us know if any further details would be helpful.
> > >
> > > ---
> > >
> > > **References:**
> > >
> > > Shumailov I, Shumaylov Z, Zhao Y, Papernot N, Anderson R, Gal Y. AI models collapse when trained on recursively generated data. *Nature*. 2024;631(8022):755-759.
> > >
> > > Alemohammad S, Casco-Rodriguez J, Luzi L, Humayun AI, Babaei H, LeJeune D, Siahkoohi A, Baraniuk R. Self-consuming generative models go mad. *The Twelfth International Conference on Learning Representations*; 2024.
> > >
> > > Alemohammad S, Humayun AI, Agarwal S, Collomosse J, Baraniuk R. Self-improving diffusion models with synthetic data. *arXiv preprint arXiv:2408.16333*; 2024.
> > >
> > > Zheng K, Chen Y, Chen H, He G, Liu M-Y, Zhu J, Zhang Q. Direct discriminative optimization: Your likelihood-based visual generative model is secretly a GAN discriminator. *Proceedings of the 42nd International Conference on Machine Learning*; 2025.

---

> > > > ### Comment · Reviewer_dZE2 · 2025-11-25
> > > > **Reply to rebuttal**
> > > >
> > > > I sincerely thank the author for responding to my questions and made a lot more contribution to the paper. I find the qualitative examples and quantative comparison to self-training reall enhance the soundness and further improve the readability of the paper. I believe this is a really solid work and will maintain my strongly positive score. Thanks.

---

> > > > > ### Author Response · Authors · 2025-11-26
> > > > > **Thank you**
> > > > >
> > > > > Dear Reviewer dZE2,
> > > > >
> > > > > Thank you for engaging with our response and for the positive feedback. We are glad the additional qualitative examples and quantitative comparisons were helpful. We appreciate your time and support.

---

### Author Response · Authors · 2025-11-16
**Summary of Changes to the Revised Manuscript**

We sincerely thank all reviewers for their thoughtful feedback. We have made the following major changes to the manuscript:

**Note:** Changes in the main paper are highlighted in blue for easy identification.

---

### **1. Strengthened Theoretical Presentation**

- **Added Gaussian Warmup Example (new Appendix B.1):** A fully worked analytical example demonstrating Neon's mechanism on Gaussian estimation with closed-form calculations.

- **Added Proof Roadmap (new Appendix B.2):** A detailed roadmap outlining the five-stage logical structure of our proofs.

- **Reorganized Section 3.1:** Improved transitions and clearer connections between geometric intuition and formal guarantees.

- **Improved Proof Flow:** Added linking explanations between theorems throughout the appendix for better readability of proofs.

---

### **2. Extensive Qualitative Visualizations**

- **Single-Sample Effect Visualization (new Appendix K):** Demonstrates how Neon affects individual samples through pixel-wise heatmaps, showing semantically meaningful changes rather than random noise.

- **Systematic Grid Visualizations (Appendices H, I, J):** 2D grids showing joint effects of merge weight w and CFG scale γ across diverse ImageNet classes for IMM, VAR-d36-s, and EDM-VP models.

- **Extended Baseline Comparisons (Figures 4, D.1):** Added w = −1 (direct self-training) to demonstrate that self-training degrades performance while Neon improves it.

---

We believe these revisions substantially improve the manuscript's clarity and empirical support. We are grateful for the constructive feedback.

---

---

### Meta-Review · Area_Chair_UXmX · 2026-01-06

**Summary:**

Generative models trained on their own generations are known to degenerate and lose performance; this paper proposes to address this issue by reversing the gradient direction provided by this training procedure. Reviewers unanimously praised the paper for its thorough and convincing experiments and all recommend acceptance. The proposed method is cheap and efficient, and it could be impactful if it ends up being adopted. This paper is a clear accept.

**Reviewer Concerns:**

I don't believe there are significant outstanding concerns from the reviews. The authors added examples and visualizations which I believe addressed the main concerns raised by the reviewers.

**Reviewer Scores:**

I believe reviewer BLA3 would have increased their score as indicated in the discussion. I don't think other reviewers would have increased their score: reviewers dZE2 and Srik had already provided strong scores and their concerns were minor, whereas reviewer YZmD's main concern was about differences with existing work, and I find it hard to judge how convinced the reviewer would have been.

---

### Decision · Program_Chairs · 2026-01-26

Accept (Oral)